# Contribution of the coupled atmosphere–ocean–sea ice–vegetation model COSMOS to the PlioMIP2

Christian Stepanek[1], Eric Samakinwa[1,2,3], Gregor Knorr[1], and Gerrit Lohmann[1,4]

[1]Alfred Wegener Institute – Helmholtz-Centre for Polar and Marine Research, Bremerhaven, Germany
[2]Institute of Geography, University of Bern, Bern, Switzerland
[3]Oeschger Centre for Climate Change Research, University of Bern, Bern, Switzerland
[4]Institute for Environmental Physics, University of Bremen, Bremen, Germany

**Correspondence:** Christian Stepanek (Christian.Stepanek@awi.de)

**Abstract.** We present the Alfred Wegener Institute's contribution to the Pliocene Model Intercomparison Project, Phase 2 (PlioMIP2) where we employ the Community Earth System Models (COSMOS) that include a dynamic vegetation scheme. This work builds on our contribution to Phase 1 of the Pliocene Model Intercomparison Project (PlioMIP1) where we employed the same model without dynamic vegetation. Our input to the PlioMIP2 special issue of Climate of the Past is twofold.

In an accompanying manuscript we compare results derived with COSMOS in the framework of PlioMIP2 and PlioMIP1. With this manuscript we present details of our contribution with COSMOS to PlioMIP2. We provide a description of the model and of methods employed to transfer reconstructed mid-Pliocene geography, as provided by the Pliocene Reconstruction and Synoptic Mapping Initiative, Phase 4 (PRISM4), to model boundary conditions. We describe the spin-up procedure for creating the COSMOS PlioMIP2 simulation ensemble and present large scale climate patterns of the COSMOS PlioMIP2

mid-Pliocene core simulation. Furthermore, we quantify the contribution of individual components of PRISM4 boundary conditions to characteristics of simulated mid-Pliocene climate and discuss implications for anthropogenic warming. When exposed to PRISM4 boundary conditions, COSMOS provides insight into a mid-Pliocene climate that is characterised by increased rainfall (+0.17 mm d$^{-1}$) and elevated surface temperature (+3.37 °C) in comparison to the Pre-Industrial (PI). About two-thirds of the mid-Pliocene core temperature anomaly can be directly attributed to carbon dioxide that is elevated w.r.t. PI.

The contribution of topography and ice sheets to mid-Pliocene warmth is in contrast much smaller – about one-quarter and one-eighth, respectively, and nonlinearities are negligible. The simulated mid-Pliocene climate comprises pronounced polar amplification, a reduced meridional temperature gradient, a northwards shifted tropical rain belt, an Arctic Ocean that is nearly free of sea ice during boreal summer, as well as muted seasonality at Northern Hemisphere high latitudes. Simulated mid-Pliocene precipitation patterns are defined by both carbon dioxide and PRISM4 paleogeography. Our COSMOS simulations

confirm longstanding characteristics of the mid-Pliocene Earth system, among these increased meridional volume transport in the Atlantic Ocean, an extended and intensified equatorial warm pool, as well as pronounced poleward expansion of vegetation cover. By means of a comparison of our results to a reconstruction of sea surface temperature (SST) of the mid-Pliocene we find that COSMOS reproduces reconstructed SST best if exposed to a carbon dioxide concentration of 400 ppmv. In the Atlantic to Arctic Ocean the simulated mid-Pliocene core climate state is too cold in comparison to the SST reconstruction.

The discord can be mitigated to some extent by increasing carbon dioxide that causes increased mismatch between model and reconstruction in other regions.

# 1   Introduction

Climate projections provide policymakers with a range of possible future climates (Collins et al., 2013). They deliver knowledge that is key to preparing humankind for future environmental conditions under the impact of elevated carbon dioxide of anthropogenic origin – or, in other terms, to climate change. This issue is urgent and not academic in nature. Climate change is accelerating as inferred from observations of ocean heat content (Cheng et al., 2020). Furthermore, its fingerprints can already be traced in short-term weather data (Sippel et al., 2020). Over the observational period climate simulations are a tool to separate the climatic impact of human activities, that are related to emission of radiative active trace gases, from natural climate forcing (Bindoff et al., 2013). Yet, when directly studying future climate by means of modelling, we have to rely on precision and accuracy of climate models. These are continuously improved (Flato et al., 2013), but per definition of a *model* will always only provide an idealized representation of processes and mechanisms that drive the Earth's climate system. Consequently, model-dependency of simulated climate is to be expected and has been shown (e.g. Haywood et al., 2009a).

Fortunately, Earth's geologic history provides us with a unique laboratory in which we can study the climate of warmer worlds. There, we find complementary information that is independent from climate models. Performing climate simulations in the context of paleoclimatology furthermore enables us to test our model against climate states that are warmer than the current one, for which the models have been developed. Successful reproduction of past climates increases confidence in a climate model as a tool for climate projections (Flato et al., 2013). In the mid-Pliocene (about 3.3–3.0 Ma BP; Haywood et al., 2016), a time period within the Piacenzian Age (3.60–2.59 Ma BP; Gradstein et al., 2012) that is a subdivision of the Pliocene Epoch (5.33–2.59 Ma BP; Gradstein et al., 2012), a warmer–than–present climate state has been found. Global average temperatures of the mid-Pliocene were elevated by about 2–3°C above recent conditions (Jansen et al., 2007; Haywood et al., 2013a). Note that the approximate time period 3.3–3.0 Ma BP has been referred to as mid-Pliocene warm period (Haywood et al., 2010, 2011, 2013a), mid-Pliocene (Haywood et al., 2016), and mid-Piacenzian (Dowsett et al., 2016). For simplicity and convenience, we refer to it exclusively as *mid-Pliocene*.

For the Pliocene Model Intercomparison Project, Phase 1 (PlioMIP1) Haywood et al. (2010, 2011) have proposed mid-Pliocene climate simulations for investigating agreement between different climate models. An additional aim was testing the degree of congruence between climate model output and climate reconstructions from the geologic archive (for brevity, we will use the term *data* to distinguish a climate reconstruction from model output, the latter being simply referred to as *model*). The intercomparison has been performed in an internationally coordinated effort based on an extensive climate model ensemble (Haywood et al., 2013a). One inference of PlioMIP1 has been that climate models agree with each other, and with data, with regard to specific characteristics of simulated mid-Pliocene climate, while there is disagreement for others. The mid-Pliocene surface temperature anomaly in the tropics as simulated in PlioMIP1 is an example of model–model agreement, whereas mid-Pliocene temperatures at high latitudes reveal both model–model and model–data discords (Haywood et al., 2016,  and

references therein). With these results in mind, Haywood et al. (2016) have introduced the second phase of PlioMIP (PlioMIP2). Differences to the PlioMIP1 modelling protocol include use of updated paleogeography provided by Dowsett et al. (2016) and an extended simulation ensemble (Haywood et al., 2016). The aim of PlioMIP2 is not only to further explore reasons for model–model and model–data discord that has been identified in PlioMIP1, but also to understand, from the viewpoint of the past, potential aspects of future climate (Haywood et al., 2016).

With the work presented here we carry on our contribution from PlioMIP1 to PlioMIP2. Following the PlioMIP2 modelling guidelines by Haywood et al. (2016) we expose the Community Earth System Models (COSMOS) to mid-Pliocene geography provided by Dowsett et al. (2016) in the framework of the Pliocene Reconstruction and Synoptic Mapping Initiative, Phase 4 (PRISM4). As per PlioMIP2 modelling protocol our contribution evolves from presenting one single mid-Pliocene climate state in PlioMIP1 (Stepanek and Lohmann, 2012), that was based on a best knowledge model setup (Haywood et al., 2010, 2011), to provision of an extensive and diverse simulation ensemble in PlioMIP2. This simulation ensemble allows us to test sensitivity of COSMOS to mid-Pliocene trace gas forcing and to different components of reconstructed geography. Note that our contribution to PlioMIP2 is twofold:

a) In order to bridge the gap between PlioMIP1 and PlioMIP2, Samakinwa et al. (2020) explore mid-Pliocene climate, as simulated with COSMOS in PlioMIP1 (Stepanek and Lohmann, 2012) and PlioMIP2 (this study). Their focus is on identification of differences and of their origin. Furthermore, Samakinwa et al. (2020) sample the impact of plausible variations of orbital forcing during the mid-Pliocene on the simulated climate – an aspect that is beyond the PlioMIP2 simulation ensemble as defined by Haywood et al. (2016), but that has been proposed as a potential contributor to model–data disagreement (Prescott et al., 2014).

b) With the present publication we provide an extensive PlioMIP2 simulation ensemble based on COSMOS. Our aims are to sample mid-Pliocene climate and its drivers and to interpret paleoclimate patterns in the context of the future. Beyond the simulation ensemble proposed by Haywood et al. (2016) we present additional simulations that aim at quantifying the effects of ocean gateways and high carbon dioxide levels. By employing the same model as for PlioMIP1 (Stepanek and Lohmann, 2012) we provide continuity from PlioMIP1 to PlioMIP2. This avoids overprint of specific characteristics of a new model on our results towards enhancing the climate signal created by updates in boundary conditions from PlioMIP1 to PlioMIP2. Important differences to our modelling methodology in PlioMIP1 are improvements of the paleogeographic model setup, in particular a more detailed representation of mid-Pliocene coast lines. Furthermore, while per requirement in PlioMIP1 (Haywood et al., 2010, 2011) a static global vegetation distribution of the mid-Pliocene, based on a reconstruction by Salzmann et al. (2008a, b), has been employed (Stepanek and Lohmann, 2012), we now apply vegetation dynamics in the model. Consequently, vegetation patterns are consistent with the model climate, and climate–vegetation–feedbacks are dynamically represented in our simulation results.

This manuscript is organised as follows. In Section 2 we outline our methodology of modelling various realisations of Pre-Industrial (PI), quasi-modern, quasi-future, and mid-Pliocene climate states. In this context we also provide the motivation for our model choice. We describe our model, illustrate methods of implementation of reconstructed mid-Pliocene boundary conditions, and document the model spin-up procedure. The section concludes with a survey of COSMOS PlioMIP2 climate

simulations. Section 3 presents selected results from the COSMOS PlioMIP2 simulation ensemble. In Section 4 we discuss our model results in the context of Pliocene4Pliocene and Pliocene4Future foci as suggested by Haywood et al. (2016). Section 5 concludes and summarises our publication.

## 2 Methodology

We describe the methodology towards generating model output for the PlioMIP2 mid-Pliocene core simulation and the PI reference state. Our documentation also covers sensitivity simulations that explore the impact of various potential contributors to mid-Pliocene warmth. We give justification for using COSMOS in PlioMIP2 and provide a description of the model. As per request in PlioMIP, we document in detail the transfer functions applied to reconstructed paleogeography towards generation of a mid-Pliocene model setup, and outline our methodology of model spin-up. This section concludes with a survey of the
COSMOS simulation ensemble produced for PlioMIP2.

### 2.1 Choice of the model

For our contribution to PlioMIP2, we employ the same model as in the framework of PlioMIP1 (Stepanek and Lohmann, 2012). One difference is the utilisation of the dynamic vegetation module that was disabled for the purpose of PlioMIP1. Surveys of models represented in the PlioMIP1 (Haywood et al., 2013a) and PlioMIP2 (Haywood et al., 2020) simulation
ensembles reveal that COSMOS is one of seven climate models in the 16 member PlioMIP2 model ensemble that has already been employed for PlioMIP1. The COSMOS model employed here is of comparably low spatial resolution and there is only one other PlioMIP2 climate model that employs a similarly low resolution in the atmosphere. Furthermore, COSMOS is also an older model in the PlioMIP2 model ensemble, in particular in comparison to six PlioMIP2 models that were published since 2017. Hence, the model is not state–of–the–art anymore. Yet, COSMOS has characteristics that are advantageous for our
contribution to PlioMIP2. Currently, there is no other PlioMIP2 model that employs dynamic vegetation (Haywood et al., 2020) and fully resolves climate–vegetation–feedbacks. In the framework of PlioMIP2, availability of a dynamic vegetation module is a relevant model feature: during the mid-Pliocene vegetation distribution differed significantly from modern – in particular at high latitudes of the Northern Hemisphere (Salzmann et al., 2013), up to the high Arctic (Tedford and Harington, 2003), but also at lower latitudes in Europe, where shifts in vegetation during the Pliocene have been recorded (Viera et al., 2018; Viera
and Zetter, 2020). Furthermore, in PlioMIP1 COSMOS was shown to predict a mid-Pliocene climate that is among the warmer end members (Haywood et al., 2013a). This behaviour is related to the model's large climate sensitivity (CS), more precisely equilibrium climate sensitivity (ECS), of more than 4 °C in both PlioMIP1 (4.1 °C; Haywood et al., 2013a) and PlioMIP2 (4.7 °C for modern geography, this publication). These values are at the upper end of likely values of ECS (1.5 °C to 4.5 °C per doubling of carbon dioxide from PI conditions; Bindoff et al., 2013; Collins et al., 2013) and higher than the ECS of the
successor models MPI-ESM (3.0 °C) and AWI-ESM (3.6 °C). High CS is linked to pronounced greenhouse gas emissivity in the model that is the strongest across the PlioMIP1 model ensemble at low latitudes and Northern Hemisphere mid latitudes, and among the upper end members at all other latitude bands (Hill et al., 2014). In PlioMIP2 there are only six models with an

ECS larger than 4.0 °C, and COSMOS is among the three models with the highest ECS (Haywood et al., 2020). By employing COSMOS we ensure that the possibility of a high CS of the Earth system is represented in the PlioMIP2 model ensemble. Last but not least, what COSMOS lacks in spatial detail it makes up for by means of execution speed and modest computational cost. This is a necessary asset towards spin-up of the extensive PlioMIP2 simulation ensemble into quasi-equilibrium. We

have exploited the computational efficiency of our climate model to an extent that allows us to create additional sensitivity simulations not considered in the official PlioMIP2 modelling protocol.

On the other hand, with this publication we provide results that are not based on a climate model employed in the current iteration of model-intercomparisons in the framework of the World Climate Research Programme – in particular the Paleoclimate Model Intercomparison Project, Phase 4 (PMIP4; Kageyama et al., 2018) and the Climate Model Intercomparison

Project, Phase 6 (CMIP6; Eyring et al., 2016). As COSMOS does not implement all model characteristics that are required for a CMIP6 model, it does not take part in PMIP4 and CMIP6. Consequently, the so-called DECK simulations (Eyring et al., 2016), devised in CMIP6 towards providing a test environment where key model characteristics can be easily compared, will not be published via the Earth System Grid Federation. In order to mitigate limited comparability of COSMOS to other CMIP6 models in PlioMIP2, we will present results of three CMIP6 simulations conducted with COSMOS, bearing in mind that full

comparability of COSMOS to other CMIP6 models is in principle not possible. In addition to the PlioMIP2 PI simulation for non-CMIP6 models with 280 ppmv of carbon dioxide (cf. Haywood et al., 2016) we prepare a similar simulation where we employ the CMIP6 *piControl* radiative and orbital forcing as defined by Otto-Bliesner et al. (2017a) for comparison. Furthermore, starting from the CMIP6 PI control simulation, we test short- and long-term climate response in COSMOS to abrupt quadrupling of PI carbon dioxide and to a transient one-percent per year increase of carbon dioxide as outlined by Eyring et al.

(2016). Note that we will provide further simulations with official CMIP6/PMIP4 models at a later stage of PlioMIP2.

## 2.2   The COSMOS modelling toolbox

Stepanek and Lohmann (2012) have provided an extensive description of the model and of its PI configuration. Based on their work, here we present a general overview of model characteristics and kindly refer the interested reader to the previous publication for further details. COSMOS is a fully-coupled atmosphere-ocean-sea ice-vegetation general circulation model

without the need for flux correction. It contains model representations of the climate system components atmosphere, ocean, and land surface. Further components, not used in this study, are a coupled ice sheet and a representation of the marine branch of the carbon cycle. Below we give a description of the various model components employed in PlioMIP2.

For the simulation of atmosphere dynamics COSMOS relies on the ECHAM5 spectral atmosphere general circulation model (Roeckner et al., 2003). We employ the model at a spatial resolution of T31 (3.75°×3.75°) with 19 hybrid sigma-pressure levels

and at a time step of 2,400 s. As an internally coupled extension ECHAM5 includes the land surface and carbon cycle model JSBACH (Raddatz et al., 2007). Hence, COSMOS is able to resolve the land-based part of the carbon cycle. It is able to adapt global vegetation distribution and related albedo- and evapotranspiration-feedbacks in the presence of changes in ambient climate by means of a dynamic vegetation model (Brovkin et al., 2009). This model characteristic ensures that the state of vegetation in the model is always consistent with prescribed soil type and modelled climate, and that climate-vegetation-

feedbacks, related to vegetation's albedo and evapotranspiration, are considered in the model. In the paleo-mode of COSMOS, that is employed here, JSBACH simplifies the complexity of natural flora by considering eight different plant functional types (PFTs). Four of the PFTs represent characteristics of different tree species. There are two types of tropical (evergreen, deciduous broad-leaved) and two types of temperate (evergreen and deciduous) forest. Furthermore, the model setup resolves raingreen

and cold shrubs (tundra) as well as two types of perennial grass (C3/C4). The hydrological cycle is closed by COSMOS via a river routing scheme that returns excess precipitation over the continents along the topographic gradient back to the ocean (Hagemann and Dümenil, 1998a, b; Hagemann and Gates, 2003). In simulations where topography and/or land sea mask differ from modern conditions, this scheme is adjusted to reflect reconstructed changes in paleogeography, towards computation of river flow that is consistent with both topography and land sea mask in the model.

A numeric solution of ocean circulation is provided by MPIOM (Marsland et al., 2003) that is integrated with a time step of 8,640 s without any flux correction. We employ MPIOM at a curvilinear bipolar GR30 grid configuration with 40 unequally spaced depth-levels. This model setup is characterised by a nominal lateral resolution of $1.8° \times 3.0°$ with strong regional dependency on the actual grid cell area. Overall, resolution of the grid varies between 29–391 km of lateral extent per grid cell. The two grid poles are located over land masses of Greenland and Antarctica and facilitate increased spatial resolution

in particular for the high latitude North Atlantic Ocean. Beyond ocean circulation, MPIOM performs dynamic-thermodynamic simulation of sea ice based on the model described by Hibler (1979). As an important process for breaking stratification, MPIOM parameterises overturning by convection by means of increased vertical diffusivity (Jungclaus et al., 2006). Towards a proper representation of details of global bathymetry on the rather coarse vertical model resolution, that reduces from 10 m layer thickness near the ocean surface to 600 m layer thickness in the deep ocean, we employ the model's ability to represent

topography on "partial grid cells" (Marsland et al., 2003) where the thickness of the lowermost wet grid cell is adjusted to local bathymetry. Important dynamic model characteristics, that improve the simulation at the relatively coarse model resolution, include eddy-induced mixing (Gent et al., 1995), subgrid-scale mixing based on an isopycnal diffusion scheme (Marsland et al., 2003), as well as a bottom boundary layer scheme (Beckmann and Döscher, 1997; Lohmann, 1998; Legutke and Maier-Reimer, 2002). The model ocean is coupled to the atmosphere/land model by means of the OASIS3 ocean-atmosphere, sea ice,

and soil coupler (Valcke et al., 2003). Fluxes of mass, energy, and momentum are exchanged between atmosphere and ocean once a day. The overall structure of COSMOS is illustrated schematically in Fig. 1.

## 2.3 Transfer of reconstructed mid-Pliocene paleogeography to COSMOS

All simulations that are based on a modern geography employ the standard modern model setup of COSMOS that is described in detail by Stepanek and Lohmann (2012). For simulations that equip a full or partial set of mid-Pliocene geography we adjust

modern COSMOS boundary conditions towards mid-Pliocene Earth system characteristics as reconstructed in the framework of PRISM4 (Dowsett et al., 2016). We employ the enhanced data set of boundary conditions and follow implementation guidelines by Haywood et al. (2016). With the aim of ensuring comparability of results derived by us with the model in PlioMIP2 and PlioMIP1, we employ similar transfer methods as Stepanek and Lohmann (2012) did for their PlioMIP1 simulations. Respective updates to our implementation strategy are outlined below.

### 2.3.1 Mid-Pliocene core simulation setup

We aim at faithful reproduction of the reconstructed PRISM4 mid-Pliocene land sea mask and optimal accommodation of paleogeographic characteristics, including changes in the state of specific ocean gateways. These differ in various details from the suggested PlioMIP1 setup (cf. Haywood et al., 2010, 2011, 2016; Dowsett et al., 2016). For PlioMIP2 we adapt soil characteristics in the model as defined by the reconstruction, and compute vegetation dynamics rather than prescribing a vegetation reconstruction as done for PlioMIP1. Furthermore, there are slight changes in mid-Pliocene carbon dioxide ($400\,\mathrm{ppmv}$ in PlioMIP2 instead of $405\,\mathrm{ppmv}$ in PlioMIP1; cf. Haywood et al., 2011, 2016) and, specifically to our model, minor adaptations of orbital parameters and updated concentrations of nitrous oxide and methane due to adjustment of the COSMOS PI model setup towards PMIP4 forcing (Otto-Bliesner et al., 2017a). These updates to the modelling methodology imply that not all differences in climate, as simulated by COSMOS in PlioMIP1 and PlioMIP2, can be fully attributed to updates of the reconstruction of paleogeography from PlioMIP1 (PRISM3D; Dowsett et al., 2010) to PlioMIP2 (PRISM4; Dowsett et al., 2016). Studying details of the impact of differences in modelling methodology on the PlioMIP1/PlioMIP2 mid-Pliocene climate states simulated with COSMOS is beyond the framework of this study and pursued by Samakinwa et al. (2020).

Probably the most obvious difference in our PlioMIP2 model setup with respect to the PlioMIP1 setup by Stepanek and Lohmann (2012) is that the mid-Pliocene land sea mask in the model is now substantially modified from modern conditions. Paleogeographic boundary conditions, that are to be implemented as an anomaly (i.e. ocean bathymetry and land topography), are transferred to COSMOS by computing their anomaly from PRISM4 modern conditions following equations in section 2.3.2 by Haywood et al. (2016). Generation of anomalies is performed at PRISM4 resolution ($1° \times 1°$). The resulting anomalies are then interpolated to model resolution and added to the respective boundary condition of the modern COSMOS setup. In some aspects the methodology suggested by Haywood et al. (2016) is adjusted in order to meet model-specific characteristics of COSMOS, as will be outlined below.

We employ a three step approach to set up mid-Pliocene bathymetry, topography, and land sea mask. First, the mid-Pliocene ocean bathymetry is generated based on the above outlined anomaly method. In various regions of land and ocean realms the approach does not reproduce the reconstructed mid-Pliocene distribution of land and ocean as provided by PRISM4. Where details of reconstructed land sea mask, coastlines, and ocean gateway state are lost in the combined approach of interpolation and anomaly method, MPIOM mid-Pliocene bathymetry and land sea mask are manually corrected: to restore reconstructed mid-Pliocene ocean regions, model bathymetry is set to the depth of the nearest available grid cell where bathymetry, computed from the anomaly method, is consistent with the reconstructed land sea mask; reconstructed land regions on the other hand are restored in MPIOM by explicitly setting ocean bathymetry to zero.

Second, based on the corrected mid-Pliocene bathymetry and the implied land sea mask of the ocean model, we create the mid-Pliocene land sea mask for the atmosphere model. Consistency between ocean and atmosphere land sea masks is achieved by first interpolating the ocean bathymetry to the atmosphere model grid and subsequently correcting the result where interpolation caused discord between atmosphere and ocean land sea masks. The derived atmosphere model land sea

mask serves as the foundation for setting up all other atmosphere- and land-model mid-Pliocene paleogeographic boundary conditions in the third, following, step.

We generate the geopotential of the earth surface, that acts as the lower boundary condition for the atmosphere model, via the anomaly method, similarly as done for ocean bathymetry. The reconstructed anomaly of mid-Pliocene topography, that already

incorporates the reconstructed height of mid-Pliocene ice sheets, is interpolated to the atmosphere model grid and added to the modern model topography. The result is converted to surface geopotential and smoothed in the spectral domain. Modifications of topography are applied, where necessary, in order to ensure consistency with the atmosphere model's (and consequently, the ocean model's) land sea mask. The PRISM4 reconstruction of mid-Pliocene ice sheet extent is interpolated to the atmosphere model grid and replaces the modern ice mask towards realistic representation of mid-Pliocene land surface albedo. As a result

of these steps we derive a representation of geography in the mid-Pliocene setup, where ice sheets are substantially reduced in comparison to the modern reference setup, and where the Hudson Bay, Canadian Arctic Archipelago, as well as the Bering Strait are replaced by land (Fig. 2).

In order to further complete the COSMOS mid-Pliocene model setup we adapt ECHAM5's parameterisation of sub-gridscale topography characteristics based on the reconstructed PRISM4 mid-Pliocene topography data set. Furthermore, we update the

river routing scheme to ensure that runoff is consistent with the topographic gradient. Both methods are outlined in detail by Stepanek and Lohmann (2012). From the PRISM4 reconstruction of soil characteristics (Pound et al., 2014) we compute global distributions of soil-related parameters: near-infrared, visible-range, and background surface albedo, maximum soil field capacity, and FAO soil classes. To this end we employ a regression approach that bears similarity to the method devised by Stepanek and Lohmann (2012) for implementing the vegetation reconstruction by Salzmann et al. (2008a, b) into model

boundary conditions and that produced satisfactory results. The starting points are the PRISM4 modern soil map and its mid-Pliocene counterpart, interpolated to the atmosphere- and land-surface model grid. Gaps over land in the interpolated data sets, that arise from interpolation and small inconsistencies between modern land sea masks of PRISM4 and COSMOS, are closed by considering the dominant soil type among the neighbouring grid cells. This approach yields global masks of both modern and mid-Pliocene PRISM4 soil type distributions on the ECHAM5 model grid. Of these, the modern masks are used to compute

average characteristics of the following soil-related parameters in the modern model setup of ECHAM5/JSBACH: maximum soil field capacity, three types of surface albedo, and FAO soil classes. For each of these quantities, we average over all grid cells of the modern, namely PI, COSMOS setup, that overlap with the respective modern PRISM4 mask of a specific soil type. Based on the necessary assumption that a PRISM4 soil type has similar characteristics today and in the mid-Pliocene, we generate the mid-Pliocene boundary condition via a superposition of the derived average modern model characteristics

of PRISM4 soil types by guidance of the reconstructed mid-Pliocene soil distribution on the model grid. Where land ice is reconstructed in the mid-Pliocene, the respective land region in COSMOS is set to soil characteristics that are consistent with those of land ice regions in the modern model setup.

Some modern characteristics of the COSMOS model setup are not modified towards reconstructed PRISM4 conditions in our mid-Pliocene model setup. Surface roughness by elevation is kept unchanged from the PI setup as no respective information

is available beyond modern conditions. Snow depth and soil wetness are initial conditions that are updated by the model during

the course of any model integration based on simulated climate. These are therefore not modified from modern conditions. All simulations are conducted employing dynamic vegetation. Hence, any vegetation-related boundary condition of the modern COSMOS setup is not employed and hence also not manually updated to reconstructed mid-Pliocene characteristics. As mid-Pliocene topography and hydrological cycle differed from modern conditions (Haywood et al., 2013a, 2016; Dowsett et al.,
2016), a full consideration of paleoclimatic boundary conditions in PlioMIP2 in principle also entails adaptation of the lake distribution to the data set provided by Pound et al. (2014). Yet, since our modern, respectively PI, COSMOS model setup does not explicitly consider lakes, we refrain from implementing these into mid-Pliocene and derived setups for PlioMIP2 simulations.

### 2.3.2   Hybrid mid-Pliocene/PI model setups

For two types of model simulations we generate the setup by means of an adapted methodology. A subset of simulations described in the modelling protocol (Haywood et al., 2016) are based on a superposition of modern and mid-Pliocene characteristics inside and outside modern ice sheet regions, respectively. We assemble the hybrid model setup by superposing the characteristics of the modern (PI) and, described above, mid-Pliocene core simulation setup. Due to the spectral core of the atmosphere model surface geopotential is subject to special treatment. The boundary condition is not derived directly from
a superposition of modern and mid-Pliocene components, but is rather recomputed from a superposition of modern and mid-Pliocene topography and subsequently smoothed to the spectral domain. This approach is necessary in order to correctly resolve far-field effects of elevation gradients in the spectral domain. These cannot be properly represented by a simple superposition of modern and mid-Pliocene surface geopotential.

    In extending the PlioMIP2 modelling protocol we also provide a model setup derived from the mid-Pliocene core setup,
where modern states of the Northern Hemisphere gateways, the Bering Strait and Canadian Arctic Archipelago, as well as the Hudson Bay (for brevity, below we will refer to these three as *Northern Hemisphere gateways*) are retained. In this case the mid-Pliocene ocean model setup is derived from the full paleogeographic reconstruction, but ocean bathymetry and land sea mask are set to modern conditions in respective gateway regions. The atmosphere's land sea mask is modified in a similar manner, and all mid-Pliocene boundary conditions of atmosphere and land-surface are projected onto it based on the methodology
described for the mid-Pliocene core model setup. In particular, where the land sea mask differs in this model setup from the mid-Pliocene core setup, the river routing scheme is updated to ensure that there are no volume sinks within the routing of overland water flow to the ocean.

### 2.4   Exposure of COSMOS to non-geographic boundary conditions

Beyond the representation of land surface conditions we apply non-geographic boundary conditions and model forcing to
complete the mid-Pliocene COSMOS model setup. For PlioMIP2-related simulations we prescribe concentrations of radiative active trace gases as defined by Haywood et al. (2016). There, a preference is given to a PI climate forcing of 280 ppmv of carbon dioxide. As COSMOS is not an official CMIP6 model, we take the freedom to follow the suggestion by Haywood et al. (2016) rather than employing CMIP6/PMIP4-conforming concentrations of carbon dioxide. This is towards simplification

of the analysis of climate anomalies for simulations that are based on a doubling of PI carbon dioxide to 560 ppmv. For both mid-Pliocene and PI, concentrations of methane and nitrous oxide are set to PI values suggested by Otto-Bliesner et al. (2016) in their discussion paper, that provided the most up-to-date information when we set up and integrated our model simulations. Hence, there is a small discrepancy between concentrations of these two trace gases as used in our model setup and as officially defined for PMIP4 in the final version of the manuscript by Otto-Bliesner et al. (2017a). Yet, we would like to highlight that deviations are vanishingly small. We employ 808.000 ppbv of methane in our simulations (808.249 ppbv are suggested by Otto-Bliesner et al., 2017a) and 273.0 ppbv of nitrous oxide (273.021 ppbv are suggested for PMIP4).

As outlined in Section 2.1 we also provide a subset of CMIP6-related simulations (see Section 2.6 below). This is in order to improve the comparison of COSMOS model output to that produced by models that are based on the official CMIP6 protocol and the PI climate forcing defined for PMIP4 (Otto-Bliesner et al., 2017a). As a result, in the framework of PlioMIP2 a grading of the COSMOS' sensitivity to carbon dioxide forcing in comparison to other CMIP6 models is enabled. Rationale is that within our contribution to PlioMIP2 there is no possibility to bring COSMOS, that is a legacy model suite, into a state that fully satisfies demands by CMIP6 with regard to model characteristics, parameters, and climate forcing. We provide two CMIP6 simulations suggested by Eyring et al. (2016) that quantify the climatic impact of a steadily increasing carbon dioxide forcing (CMIP6-simulation *1pctCO2*) and of an abrupt increase to four times the PI carbon dioxide forcing (CMIP6-simulation *abrupt4xCO2*). The transient carbon dioxide forcing used in 'ramp-up' simulation 1pctCO2 spans calendar years 1850 CE to 2100 CE and is taken from Table 9 of Meinshausen et al. (2017). For simulation abrupt4xCO2, carbon dioxide concentration is computed by quadrupling the PI volume mixing ratio defined by Otto-Bliesner et al. (2017a). Furthermore, we provide the CMIP6/PMIP4 PI control simulation, *piControl*, employing trace gas concentrations for carbon dioxide, methane, and nitrous oxide, as well as orbital forcing, that are in agreement with PMIP4 specifications (284.317 ppmv of carbon dioxide, 808.249 ppbv of methane, and 273.021 ppbv of nitrous oxide). This simulation is created to provide an equilibrium PI climate state, conforming to CMIP6 standards, as a starting point for simulations 1pctCO2 and abrupt4xCO2, but is not analysed in detail here. A deviation of the model setups of these simulations from CMIP6/PMIP4 specifications is the value of the solar constant, that is 1367 W m$^{-2}$ in the COSMOS reference setup. This setting has already been employed in PlioMIP1, and the solar constant hence differs from the value suggested for CMIP6 and PMIP4 (1360.747 W m$^{-2}$). We refrain from modifying the solar constant in our PlioMIP2 model setup towards the official CMIP6/PMIP4 condition. On the one hand such a step would have required a re-tuning of the coupled model. On the other hand we would have hampered comparability of PlioMIP2 COSMOS simulations to a vast amount of previous modelling work that has been done by other authors, both inside PlioMIP1 (Stepanek and Lohmann, 2012) and beyond.

## 2.5 Model initialisation and spin-up

At the start of all PlioMIP2 equilibrium simulations model components are instantaneously exposed to the full set of boundary conditions and model forcing. The same holds for CMIP6 equilibrium simulations piControl and abrupt4xCO2. To avoid initial numerical instability in simulations where land surface conditions differ significantly from modern, the atmosphere model is integrated with half the time step (1,200 s instead of 2,400 s) during the first model year only. For all simulations we initialise

the ocean from the three-dimensional climatology of Levitus et al. (1998), where necessary adapted to modifications in ocean bathymetry. The choice of initialisation data is independent of the context of a simulation – whether it is representative for PI (respectively modern), mid-Pliocene, future, or the model setup is a hybrid of these. In contrast to spinning up the model from arbitrary conditions, this approach has the advantage of shortening spin-up time for the deep ocean to a near-equilibrium state.

This time gain is critical for enabling us to prepare the large COSMOS simulation ensemble, despite the relatively modest computational demand of the model. In contrast, the two CMIP6-related simulations with abruptly and steadily increased concentrations of carbon dioxide (see Table 1), are branched off from the equilibrated model state of the CMIP6 piControl simulation. This approach is motivated by the desire to facilitate a continuous progression of the modelled climate (Eyring et al., 2016) from PI conditions to a state with increased carbon dioxide. Integration of simulations towards equilibrium (Table 1)

is longer than requested by Haywood et al. (2016) and Eyring et al. (2016) in order to ensure that the analysed model states are not affected by imperfect model equilibration, and that drifts in the atmosphere and at least in upper ocean layers are as low as practically feasible. Yet, we note that the coupled atmosphere-ocean system of COSMOS needs in the order of 5,000 years to fully adapt to substantial change in forcing (Li et al., 2013). In principle, we would have aimed to integrate the model even further into equilibrium. Yet, for high carbon dioxide simulations we find multi-centennial variability of the AMOC, that is

linked to large scale modifications of surface air temperature (SAT) and sea surface temperature (SST). A similar effect has been reported for a different PlioMIP2 climate model (Hunter et al., 2019). Presence of AMOC variability during the analysis period would complicate interpretation of COSMOS PlioMIP2 equilibrium model output due to the need to establish and maintain a stable background climate state. Hence, we analyse our model results for the latest time period where centennial-scale AMOC fluctuations are not yet present in high carbon dioxide simulations.

**2.6   Survey of model simulations**

With this work we aim to understand the climate of the mid-Pliocene by means of employing COSMOS. Furthermore, we assess potential relevance of our findings in the framework of anthropogenic climate change. To this end, we follow the modelling protocol by Haywood et al. (2016), that is extended in comparison to PlioMIP1 (Haywood et al., 2010, 2011) and aims at providing an extensive set of climate simulations towards enabling a clearer differentiation between various contributors to

mid-Pliocene warmth in the framework of Pliocene4Pliocene and Pliocene4Future (Haywood et al., 2016). Among these are components of paleogeography, in particular ice sheet extent and elevation, both inside and outside modern ice sheet regions, and radiative forcing via the contribution of trace gases that lead to increased global average surface temperatures – the latter lumped into a fixed concentration of carbon dioxide prescribed to model simulations (Haywood et al., 2016).

To simplify identification of model setup characteristics of a particular simulation, we follow the terminology for simulation

names as defined by the PlioMIP2 standard (Haywood et al., 2016): a specific simulation $E$ is labelled with an $o$ if topography differs from modern conditions; it is labelled with an $i$ if ice sheets differ from modern conditions; presence of both an $i$ and an $o$ in the simulation name signifies that the model setup is modified towards mid-Pliocene conditions both with respect to ice sheets and topography. Concentrations of carbon dioxide in units of ppmv are identified by their numerical value in the simulation name. For the mid-Pliocene core simulation Eoi400, for example, both land and ice are adjusted to mid-Pliocene

conditions, and 400 ppmv of carbon dioxide are prescribed throughout the simulation. For simulation Ei280 on the other hand, topography and albedo are specified to mid-Pliocene conditions only for modern ice sheet regions of Greenland and Antarctica, while other regions are kept at conditions as in the PlioMIP2 core PI control simulation (E280). For the simulation with modern Northern Hemisphere gateways in a model setup equivalent to the PlioMIP2 mid-Pliocene core simulation Eoi400, difference in gateway configuration is highlighted by the letters *GW*, that follow the carbon dioxide concentration after an underscore. We provide details of setups of the various simulations in Table 1.

We exploit the comparably modest computational expense of COSMOS and present the full set of PlioMIP2 model simulations as proposed by Haywood et al. (2016) in their Table 3. Beyond the protocol we provide two sensitivity simulations based on mid-Pliocene paleogeography and one based on modern geography. Of the simulations based on mid-Pliocene geography, one produces a climate state with high carbon dioxide (560 ppmv, i.e. a doubling of PI concentrations) towards derivation of ECS for mid-Pliocene paleogeography. Another simulation tests how a mid-Pliocene climate state would be characterised if the configuration of Northern Hemisphere gateways was as for modern conditions. The rationale is to derive the effect of the Bering Strait, Hudson Bay, and Canadian Arctic Archipelago on mid-Pliocene climate towards isolating this effect from that of other aspects of paleogeography. The sensitivity simulation with modern geography tests the impact of an even higher level of carbon dioxide (600 ppmv). It allows to test which regions are sensitive to relatively small differences of carbon dioxide in a future warm state, and it enables us to identify synergies in the mid-Pliocene that intensify the effect of carbon dioxide in comparison to the modern world. For reference, and towards analysis of the transient behaviour of COSMOS to carbon dioxide forcing, we provide CMIP6-related simulations 1pctCO2 and abrupt4xCO2. The CMIP6 piControl simulation (here referred to as E280C) is generated to initialise the two aforementioned simulations. It is not analysed in this publication beyond global and hemispheric scale climate indices (Table 2).

In summary, our COSMOS simulation ensemble provides a comprehensive data resource that is suitable for a number of purposes. First of all, we present data that enables the study of mid-Pliocene climate anomalies and a comparison to PRISM4 reconstructions of SST via COSMOS core simulations E280 and Eoi400. Furthermore, we sample ECS and the impact of variations in atmospheric concentrations of carbon dioxide on climate for modern (simulations E280, E400, E560) and mid-Pliocene (simulations Eoi280, Eoi350, Eoi400, Eoi450) geography. Quantification of the contributions of ice sheet and topography to mid-Pliocene climate conditions is enabled via simulations Eo280, Eo400, Ei280, and Ei400, that are suggested by Haywood et al. (2016) for forcing factorisation. As an additional contribution, we present simulations that sample the impact of even higher carbon dioxide for modern (E600) and mid-Pliocene (Eoi560) geography. The impact of the state of three Northern Hemisphere ocean gateways on mid-Pliocene climate is quantified by means of simulation Eoi400_GW.

# 3 Results

In this section we present selected results from the extensive COSMOS PlioMIP2 simulation ensemble in the context of Pliocene4Pliocene and Pliocene4Future (cf. Haywood et al., 2016). First, we describe the state of the mid-Pliocene core simulation Eoi400. Thereafter, we analyse various aspects of mid-Pliocene and potential future climate. Beyond the presentation

of the simulations proposed by Haywood et al. (2016) we include results of additional sensitivity simulations (see Section 2.6) that shed further light on mid-Pliocene climate patterns. In order to illustrate the response of COSMOS to CMIP6 model forcing, we also show results of selected CMIP6-related simulations towards documentation of COSMOS as a non-CMIP6 model toolbox (see Section 2.6). Where no other indication is given, results presented below are based on an averaging period of 100 years.

## 3.1 Characterisation the PlioMIP2 mid-Pliocene core simulation Eoi400

### 3.1.1 Global and hemispheric average key climate indices

Modifying geography and carbon dioxide in COSMOS from PI (simulation E280) to the PRISM4 mid-Pliocene reconstruction by Dowsett et al. (2016, simulation Eoi400) causes profound differences in key global and hemispheric average climate indices (Table 2). We find a pronounced global average increase of near-surface air temperature by 3.37 °C and of sea surface temperature by 2.13 °C. Related to the warming from E280 to Eoi400 we find strongly decreased extent of sea ice across seasons and hemispheres. The highest sea ice decline is evident in the Northern Hemisphere during boreal summer to autumn (August, September, October – ASO; -80.3%), while Arctic winter to spring (February, March, April – FMA) sea ice drops by a comparably mild -48.8%. Respective changes in the Southern Ocean are in comparison weaker, with a simulated loss of austral summer to autumn (FMA) sea ice by -69.1% and of austral winter to spring (ASO) sea ice by -30.3%. Noteworthy is that Arctic sea ice is confined to an average area below $2 \cdot 10^6$ km$^2$ during ASO. Furthermore, there are globally increased levels of precipitation (+0.17 mm d$^{-1}$) and evaporation (+0.18 mm d$^{-1}$). Overall, the mid-Pliocene state is characterised by less cloud cover (-2.9%) and reduced planetary surface albedo, the albedo change being slightly biased towards the total earth surface, land and ocean (-16.6%), in comparison to albedo changes over the land surface alone (-15.6%). The simulated mid-Pliocene climate state Eoi400 is, over the analysis period, slightly less equilibrated (by 0.16 W m$^2$) than the reference climate state E280, as evidenced by top of the atmosphere (TOA) radiative imbalance. The generally high TOA radiative imbalance across the simulation ensemble (1.7–2.0 W m$^2$) is comparable to imbalances in the model that are present in the framework of PlioMIP1 (Stepanek and Lohmann, 2012). Radiative imbalance is related to the slow response of the ocean to changes in carbon dioxide forcing as shown in a previous publication (Li et al., 2013). In our case, a combination of changes in carbon dioxide and geographic boundary conditions causes a slow equilibration process that is not fully finished at the end of the model spinup (Fig. S1 in the supplement to this article). In particular, simulation Eoi400 still exhibits a temperature trend of about $7 \cdot 10^{-4}$ °C yr$^{-1}$ at 3000 m depth over the analysis period. On the other hand, the ocean surface, on which PlioMIP2 analyses focus heavily, is in quasi-equilibrium in Eoi400. During the analysis period the simulation is subject to an ocean surface temperature trend that is actually below the respective trend in the PI control state E280. Furthermore, the ocean surface cools slightly in the analysed portion of mid-Pliocene core simulation Eoi400. This suggests that the diagnosed surface temperature trend is, from the view point of the ocean surface, largely overprinted by internal variability. Similarity of TOA radiative imbalance and small residual ocean surface temperature trends across the PlioMIP2 COSMOS simulation ensemble demonstrate the expediency of

simulation Eoi400 and other COSMOS PlioMIP2 simulations for the study of climate anomalies, despite incomplete model equilibration.

### 3.1.2 Large scale climate patterns

We analyse the effect of mid-Pliocene paleogeography at 400 ppmv of carbon dioxide on selected characteristics of the climate
as simulated by COSMOS. Seasonal anomalies of SAT with respect to PI (simulation E280, cf. Fig. 3) are significant in most regions. Largest anomalies, beyond +14 °C, are apparent in modern ice sheet regions, where the PRISM4 reconstruction (Dowsett et al., 2016) suggests lower elevation and the absence of ice sheets in the mid-Pliocene. There are only a few regions where temperature anomalies are slightly negative. Most of these regions are found over, or close to, land. They are related to a) differences in the land sea mask, b) localised elevation change, as well as c) regions of increased precipitation in the
mid-Pliocene (cf. Fig. 3 to 4). Examples for a) are boreal autumn and winter cooling across Hudson Bay and the Japanese archipelago. An example of c) is cooling of the Sahel in summer, autumn, and winter. Regions of extreme warming include the whole of Eurasia north of 40°N in boreal winter, the Arctic Ocean in all seasons except for boreal summer, the Weddell Sea in austral autumn to spring, and parts of the Indian Ocean sector of the Southern Ocean. Noteworthy is also pronounced year-round warming across the Mediterranean and adjacent land regions, the Sahara, the Andes, the southernmost part of South
Africa, and parts of South America. In comparison to PI we find in the mid-Pliocene pronounced reduction of seasonality at high latitudes of the Northern Hemisphere, as warming in autumn and winter is stronger than in spring and summer.

In contrast to SAT, seasonal precipitation anomalies with respect to PI (simulation E280) show diverse spatial patterns and are less often beyond internal variability in the model (Fig. 4). The predominant precipitation pattern for mid to high latitudes from 40° poleward is slightly more precipitation across the year. Particularly pronounced are increased boreal autumn
to winter precipitation in North to Northeastern Europe, across the central North Atlantic, and over the Northeast Pacific. Greenland locally receives more precipitation in the mid-Pliocene from boreal summer to winter. There are few exceptions at these latitudes where the precipitation change is negative. Among these are regions in proximity of the Gulf Stream system, including North Atlantic Drift, North Atlantic Current, and Norwegian Current, as well as a north-eastward bending band of drying across Northeast America (year-round). Reduced precipitation is also present over the ocean near the region of the
strongly reduced West Antarctic Ice Sheet (year-round), while Antarctica as a whole receives more precipitation. Consequently, ice sheet regions of the mid-Pliocene receive more precipitation than for PI, although the gain is mostly below 0.5 mm yr$^{-1}$. Over Patagonia we find year-round a distinct precipitation dipole with more rain towards the Atlantic Ocean basin, and reduced precipitation towards the Pacific Ocean basin. In general, precipitation change poleward of 40° is much weaker in the Southern Hemisphere.

If we turn our attention towards the regions of low to mid latitudes, equatorward of 40°, we find a distinctly different pattern. Most obvious is an apparent northward shift of the Intertropical Convergence Zone (ITCZ) related to a generally increased amount of rainfall north of the equator, and consequently reduced rainfall south of the equator. This pattern is particularly pronounced in boreal summer to autumn. Related are increased boreal summer to autumn rainfall in the Sahel and over the Arabian Peninsula. We find hotspots of rainfall in the low-latitude western Pacific Ocean. There is a distinct east-west dipole

over the Indian Ocean with reduced rainfall towards Indonesia and increased rainfall towards the Arabian Peninsula and the Horn of Africa. Predominant drying is apparent over Central America, south to southeast of Asia, and generally across land masses of the Southern Hemisphere, with the exception of Antarctica and the tip of South America. Pronounced changes in low-latitude precipitation, mid-Pliocene versus PI, are related to the asymmetric warming between hemispheres. It has been shown by various authors that the ITCZ shifts towards the warmer hemisphere via links between tropical and extratropical climate (Haug et al., 2001; Broccoli et al., 2006; Kang et al., 2008; Deplazes et al., 2013; Schneider et al., 2014). In our mid-Pliocene simulations, the warmer hemisphere is the northern one, where warmth is more widespread than in the Southern Hemisphere across all seasons (Fig. 3).

The state of mid-Pliocene sea ice in the Northern Hemisphere is strongly altered from PI conditions (cf. Fig. 5a,b to Fig. 5c,d). For boreal spring differences in sea ice extent are small, including a retreat towards the western Labrador Sea, ice free conditions off the coast of both southern Greenland and Svalbard, and loss of most of the sea ice of the North Pacific Ocean. In contrast, changes in the boreal autumn mid-Pliocene marine cryosphere are much more pronounced. Sea ice retreats towards the centre of the Arctic Ocean, leading to nearly ice-free Labrador, Barents, and Kara Seas. The maximum of autumn ice distribution is in the central Arctic Ocean towards the Laptev Sea, but barely reaches 50% of compactness.

Low latitudes of the oceans also have different characteristics in the mid-Pliocene core simulation. We demonstrate this with the example of the Equatorial Warm Pool (EqWP), following the definition by Watanabe (2008), who characterises the EqWP as those ocean regions where SST exceeds 28 °C. The PI state of simulation E280 reproduces the average EqWP temperature derived from recent observations (Huang et al., 2017), but underestimates its extent (Table 3). A perfect match is not expected as observations relate to the time period from 1989 CE to 2018 CE, rather than to PI (1850 CE). Implementing mid-Pliocene paleogeography with 400 ppmv of carbon dioxide (Eoi400) increases the average EqWP SST by 1.1 °C, while nearly doubling its spatial extent (Table 3). This confirms relative warmth of the low-latitude ocean in the COSMOS PlioMIP2 mid-Pliocene core simulation, and highlights the disagreement between model simulation and various low-latitude reconstructions outside upwelling areas (e.g. Dowsett et al., 2013). In contrast, we find that the maximum stream function of the Atlantic Meridional Overturning Circulation (AMOC) is increased in simulation Eoi400 (Table 4) with respect to simulation E280. Hence, our model confirms, as suggested by Raymo et al. (1996) and Dowsett et al. (2009), that mid-Pliocene thermohaline circulation was stronger than today. Yet, we note that at least for PlioMIP1 this finding has not been consistently represented by the model ensemble (Zhang et al., 2013a), and that the overall structure of the upper AMOC cell of the mid-Pliocene in COSMOS is largely unchanged from modern patterns (Fig. 6). Beyond the increase of the maximum overturning, between 20°N and 40°N and 500 m to 2000 m depth, the AMOC shallows. This is shown by the tendency towards less intense clockwise circulation, in other words, negative AMOC anomaly, between 2000 m and 3000 m depth (Fig. 7c).

## 3.2 Steepness of the meridional temperature gradient in PlioMIP2 simulations

One characteristic inherent to mid-Pliocene reconstructions is the reduced meridional temperature range (e.g. Fedorov et al., 2013, 2015; Dowsett et al., 2011, and references therein). Increased equality between low and high latitude temperatures is caused by the tendency that warming in high latitudes is much stronger than in low latitudes as shown for the mid-Pliocene e.g.

by Dowsett et al. (2013) and Haywood et al. (2013a). This climate characteristic has also been found in PlioMIP1, explicitly shown for example by Stepanek and Lohmann (2012). Yet, a detailed study of contributions to reduced meridional temperature gradients in the mid-Pliocene was elusive in PlioMIP1 due to a lack of suitable sensitivity simulations. Here, based on the PlioMIP2 COSMOS simulation ensemble, we present an attempt to identify contributors to increased similarity of high and

low latitude temperature in the mid-Pliocene. In Fig. 8 we show zonal mean temperatures and the meridional temperature gradient for simulations with modern and mid-Pliocene geography and for various concentrations of carbon dioxide. Both for mid-Pliocene and modern geography the temperature gradient of the Northern Hemisphere is more sensitive to changes in carbon dioxide than that of the south. For simulation E400 there is a reduction of the meridional temperature range by -0.6 °C with respect to PI (E280) in the south, and in the north the reduction is -2.5 °C. For mid-Pliocene geography the respective

change from Eoi280 to Eoi400 is -1.7 °C (south) and -2.9 °C (north). The combined effect of mid-Pliocene geography and carbon dioxide (from simulation E280 to Eoi400) creates a reduction of the meridional temperature gradient by -4.2 °C (south) and -5.1 °C (north). A modern state of Northern Hemisphere gateways (simulation Eoi400_GW) causes a slight decrease in the mid-Pliocene meridional temperature gradient by -0.1 °C (south) and -0.2 °C (north). If we assume a high versus low carbon dioxide forcing in the mid-Pliocene (Eoi560-Eoi280), COSMOS suggests a drop in meridional temperature range by -2.5 °C

(south) and by -5.2 °C (north). For modern geography (E280-E560) the respective change is -1.8 °C (south) and -5.4 °C (north).

We interpret these findings in the light of Arctic or polar amplification. This term highlights the fact that for climate projections (e.g. Collins et al., 2013), as well as for paleoclimatological studies (e.g. Haywood et al., 2013a), changes of climate are most pronounced in high latitudes. Despite increasing carbon dioxide from 280 ppmv to 560 ppmv causing warming by about 6 °C (modern topography, Fig. 9) and 10 °C (mid-Pliocene topography, Fig. 10) near the South Pole, there is a relatively

modest change of the Southern Hemisphere meridional temperature gradient in the PlioMIP2 COSMOS simulation ensemble. Temperature changes in the Northern Hemisphere are on the other hand much more sensitive to carbon dioxide. This confirms findings for projected future climate that polar amplification in the Southern Hemisphere is muted in comparison to that of the Northern Hemisphere (Collins et al., 2013).

Also the applied geography has a pronounced impact on meridional temperature gradients. In particular, there is a synergy

between mid-Pliocene paleogeography and carbon dioxide in comparison to the effect if carbon dioxide is increased using modern geography. As illustrated above, for specific given variations of carbon dioxide the change in meridional temperature gradient is larger in the mid-Pliocene model setup. This is confirmed for an increase of carbon dioxide from 280 ppmv to 400 ppmv. The increase from 280 ppmv to 560 ppmv also leads to larger changes of the meridional temperature gradient for the mid-Pliocene Southern Hemisphere, but not for the Northern Hemisphere. There, the change in the gradient is very similar for

both geographic settings, with the modern one experiencing a slightly higher reduction. We furthermore find that the largest impact of carbon dioxide occurs in the mid-Pliocene setup for a change from 280 ppmv to 400 ppmv. Increasing carbon dioxide further to 560 ppmv only generates an additional reduction of the meridional temperature gradient by -2.3 °C (-2.9 °C from 280 ppmv to 400 ppmv), despite the carbon dioxide increase being higher (160 ppmv versus 120 ppmv). The impact of carbon dioxide on the meridional temperature gradient is hence dependent on the background climate and reduces towards higher

levels of carbon dioxide in our simulations. An interesting aspect of the impact of mid-Pliocene geography on the meridional

temperature gradient is that the reconstructed mid-Pliocene state of Northern Hemisphere gateways increases warmth in the North Atlantic Ocean in comparison to a modern gateway configuration (shown in Section 3.10), but has a very small impact on the meridional temperature range. In fact, the reconstructed state of Northern Hemisphere gateways opposes the general trend of reduced meridional temperature range in the mid-Pliocene, in that the modern gateway state causes a smaller meridional

temperature gradient than the reconstructed configuration.

Above derived characteristics of changes in the meridional temperature gradient are also seen in the annual mean of global SAT anomalies under changes in carbon dioxide. These are shown for modern (Fig. 9) and mid-Pliocene geography (Fig. 10). Related to the lack of topographic differences in high latitudes in the PI setup, simulation E560 provides less polar amplification of the simulated temperature signal than simulation Eoi560 (cf. Fig. 9b and Fig. 10d). Changes in high latitudes in simulation

Eoi560 actually exceed those for a modern setup with 600 ppmv carbon dioxide (cf. Fig. 9c and Fig. 10d). Even when carbon dioxide is at PI levels in simulation Eoi280 we find a pronounced contribution of mid-Pliocene geography at high latitudes, that locally exceeds the impact of carbon dioxide for all simulations with modern geography (cf. Fig. 9a,b,c and Fig. 10a). Yet, only in those mid-Pliocene simulations, where carbon dioxide is increased above modern conditions, we find a pronounced zonal expansion of large anomalies in SAT beyond the regions where ice sheets have been modified (cf. Fig.10a,b,c,d).

### 3.3   Sensitivity of COSMOS to carbon dioxide in CMIP6 simulations

The PlioMIP2 simulation ensemble exclusively considers equilibrium climate states. Yet, for an interpretation of mid-Pliocene modelling results in terms of near–term future climate, a consideration of the transient model response to increased carbon dioxide is necessary. To this end we present results from a subset of CMIP6-related simulations following the design by Eyring et al. (2016). Here, we focus on those CMIP6-simulations that test the model sensitivity to changes in carbon dioxide only, i.e.

simulations 1pctCO2 and abrupt4xCO2. Figure 11 illustrates the time evolution of various climate indices under the impact of carbon dioxide. We show response to transient carbon dioxide forcing, applied in simulation 1pctCO2 on a CMIP6 PI reference state (simulation E280C) for 250 years until year 2100 CE (Fig. 11a), and to instantaneously quadrupled PI carbon dioxide forcing, applied in simulation abrupt4xCO2 that is based on the same initial state, but where the forcing is subsequently maintained for 1,000 years (Fig. 11b). The results of both simulations illustrate time-dependency and sensitivity of COSMOS

to variations in carbon dioxide, that is an important climate driver in the PlioMIP2 simulation ensemble as will be presented further down.

A one percent annual increase in carbon dioxide levels increases global average SST and SAT over the course of 250 years by about 10 °C and 13.5 °C, respectively. The response of SAT to increased radiative forcing outpaces that of SST, although global mean values of both reach similar values by the end of the simulation. In contrast, abruptly quadrupling carbon dioxide

and maintaining the forcing for 1,000 model years leads to ocean surface warming of 8 °C, and to SAT increase of 10 °C. While the strong carbon dioxide forcing applied in simulation 1pctCO2 over the relatively short time period of 250 years has only a moderate impact on deeper parts of the global ocean, quadrupling of PI carbon dioxide also warms the ocean at 3,000 m depth by about 2 °C over the course of 1,000 model years, and increases ocean temperatures by 8 °C at 1,000 m depth over that time. In simulation 1pctCO2 we find a large impact on the hydrological cycle that manifests via increase of global average

precipitation by 24.2% (not shown). Increased rainfall coincides with progressive reduction of global average sea surface salinity (SSS) by about $0.8\,\mathrm{PSU}$. Over the course of 1,000 years the hydrological cycle adapts to a quadrupling of carbon dioxide, and the global average SSS reaches a new equilibrium of about $33.7\,\mathrm{PSU}$. For both simulations, reduction of SSS is related to increased salinity in deeper parts of the ocean. Initially, the ocean becomes more saline around $1,000\,\mathrm{m}$ and $2,000\,\mathrm{m}$.

Over time, in simulation abrupt4xCO2 with quadrupled carbon dioxide, deeper ocean regions catch up, where salinification at $1,000\,\mathrm{m}$ depth is an initial transient effect that reverts to desalinification after about 250 years. The AMOC, diagnosed as the maximum of the meridional stream function in the Atlantic Ocean basin north of $20°\mathrm{N}$ at depths between $500\,\mathrm{m}$ and $1,500\,\mathrm{m}$, reacts strongly to carbon dioxide. After 50 years of increasing radiative forcing by one percent per year, meridional volume transport via the AMOC decays by 63% to below $6\,\mathrm{Sv}$ ($1\,\mathrm{Sv} \equiv 10^6\,\mathrm{m^3\,s^{-1}}$). Reduction of the AMOC for quadrupled carbon

dioxide is at similar magnitude, but meridional volume transport starts to relax after about 250 model years and from then on steadily increases, again reaching $10\,\mathrm{Sv}$ (or about 60% of the initial strength) after 1,000 model years. In simulation 1pctCO2, radiative forcing is steadily increasing. Similarly, radiative imbalance increases to nearly $8\,\mathrm{W\,m^2}$ at the end of the simulation. With increasing carbon dioxide there is an obvious reduction of interannual variability of the energy imbalance of the Earth system. At around year 2060 CE there is a discontinuity, after which the Earth system's ability to adapt to increased radiative

forcing reduces, as evidenced by the increased slope of TOA imbalance. In contrast, for a steady carbon dioxide forcing of quadrupled PI concentration, energy imbalance decays to $2\text{–}3\,\mathrm{W\,m^2}$, but relaxation has not completed after 1,000 years.

Beyond inferences on the impact of carbon dioxide on major climate indices and on time-characteristics of excess energy transfer to the deep ocean, simulations abrupt4xCO2 and 1pctCO2 allow us to estimate CS characteristics of COSMOS. Based on the method by Gregory et al. (2004) we provide a regression of TOA radiative imbalance in simulation abrupt4xCO2 versus

SAT change (Fig. 12). The model exhibits a clear change in the relationship between annual mean TOA radiative imbalance and SAT temperature change over the course of the simulation as evidenced by providing two different regressions – one based on the data sample of 10% of the total amount of annual mean data-pairs taken at the start of the simulation, and the other based on the remainder of the data sample. This highlights the importance of sufficient equilibration of the model towards an estimate of ECS. Based on the SAT-intercept at the extrapolated point of vanishing radiative imbalance, we estimate an ECS

(i.e. the global average SAT change for a doubling of carbon dioxide after vanishing radiative imbalance) of $4.9\,°\mathrm{C}$ for the 10% data-sample, and $6.0\,°\mathrm{C}$ if we ignore the data points of the previous analysis. There are some caveats to this analysis: the estimates are of course based on an extrapolation of the relationship TOA imbalance versus SAT change, and full equilibration of the climate state has not yet been reached at time of analysis. Furthermore, vegetation (but not land ice) adapts to and impacts on differences in the simulated climate. Hence, ECS, as derived here, considers shifts in vegetation patterns, and ignores the

response of ice sheets. A rough estimate of transient CS ($2.1\,°\mathrm{C}$, Table 5) is derived by averaging the temperature change of simulation 1pctCO2 over the time period of $\pm14$ model years around the year when doubling of carbon dioxide is reached in the transient forcing. This value is much lower than our estimate for ECS due to the incomplete response of the various model components to carbon dioxide forcing.

## 3.4 Climate- and Earth-System-Sensitivity based on PlioMIP2 simulations

In addition to inferences that we have drawn from CMIP6-related simulations on the sensitivity of climate to variations in carbon dioxide forcing, the PlioMIP2 simulation ensemble provides the opportunity to go beyond CS and ECS for both modern and mid-Pliocene geography and derive earth system sensitivity (ESS) (Table 5 and Fig. 13a,b). Here, we follow the common

definition of CS and derived metrics ECS and ESS as the global average warming of SAT that arises from a doubling of carbon dioxide from 280 ppmv to 560 ppmv. If we estimate ECS from the quasi-equilibrium state of simulations that are forced with the respective concentrations of carbon dioxide, we find that for modern geography (computed from simulations E560 and E280) there is an ECS of $4.7\pm0.4$ °C. For mid-Pliocene geography (computed from simulations Eoi560 and Eoi280) the value is slightly higher ($4.9\pm0.4$ °C). The values of ECS do not reflect differences in paleogeography between modern and mid-

Pliocene model setups. We consider these in our analysis by computing ESS from simulations Eoi560 and E280, deriving a value of $5.9\pm0.5$ °C. There is a difference between these metrics and those derived in Section 3.3 via the regression method, based on simulation abrupt4xCO2. This illustrates that prolonged simulation run time towards equilibrium, beyond what is feasible for the relatively large PlioMIP2 simulation ensemble, would likely increase ECS derived for COSMOS in this study. Consequently, ECS and ESS derived from COSMOS PlioMIP2 simulations represent conservative estimates.

Furthermore, we note that the ESS value derived here for COSMOS by means of comparing simulations Eoi560 and E280 leads to a lower value than that derived by Haywood et al. (2020) for the model. The reason is a difference in methodology. Haywood et al. (2020) derive ESS in their PlioMIP2 model intercomparison from simulations Eoi400 and E280, which involves a scaling for carbon dioxide from 400 ppmv to 560 ppmv. This is justified as the PlioMIP2 modelling protocol (Haywood et al., 2016) does not suggest a simulation Eoi560. Yet, the fact that the value of ESS depends on the employed greenhouse

forcing shows that nonlinearity between equilibrium SAT and carbon dioxide causes methodological uncertainty in derivation of ESS. To overcome such uncertainty, future iterations of PlioMIP may explicitly propose a simulation Eoi560 to consider all nonlinearities in the climate system that impact on SAT when increasing carbon dioxide from 400 ppmv to 560 ppmv.

Beyond comparison of two specific simulations with a carbon dioxide forcing of 280 ppmv and 560 ppmv, we can also attempt to analyse ECS for mid-Pliocene and modern geography based on a regression approach, considering simulations with

various levels of carbon dioxide between 280 ppmv and 600 ppmv. There is a complex relation between the change of global average SAT and carbon dioxide forcing. Simulations with intermediate carbon dioxide forcing (350, 400, and 450 ppmv) are below the regression in Fig. 13b, while simulations Eoi280 and Eoi560 are above that line. While error bars of all simulations overlap with the regression line, this hints at negative feedbacks in the model that reduce (albeit only slightly and within internal variability) the impact of intermediate levels of carbon dioxide in the mid-Pliocene. Analysis of global average albedo

in a similar manner (Fig. 13c,d) suggests that these feedbacks are albedo–related. Results derived in the course of this section's analysis should be considered in the light of limitations of the model in comparison to the real climate system. As in the case of the real world, simulated SAT and surface albedo include contributions of vegetation changes as a response to differences in ambient climate – the impact of dynamic vegetation of the model's sensitivity to carbon dioxide will be discussed in Section 4.1. The ice sheet response on the other hand is considered only indirectly via the mid-Pliocene paleogeography that reflects

reconstructions of reduced ice sheets in Greenland and Antarctica (Dowsett et al., 2016; Haywood et al., 2016). We explicitly highlight here uncertainties in mid-Pliocene ice sheets (e.g. Dolan et al., 2015, 2018). In particular, the ice-albedo effect is not considered for different carbon dioxide forcings within the mid-Pliocene and modern geography components of the PlioMIP2 simulation ensemble. Hence, the ECS and ESS, that is evident in the modelled data, is beyond the Charney Sensitivity (Charney

et al., 1979), that only considers fast feedbacks. Yet, it does not fully satisfy the definition of ESS by Lunt et al. (2010): in contrast to the Charney Sensitivity, the ESS also considers slow feedbacks related to vegetation and ocean circulation (both of which are explicitly modelled here), and (dynamic) changes of ice sheets, dust and aerosols (none of which are implemented in our model setup).

## 3.5   Impact of mid-Pliocene geography and carbon dioxide forcing on large scale climate patterns

Carbon dioxide and mid-Pliocene paleogeography have a strong impact on Arctic sea ice (Fig. 14 and Fig. 15). In comparison to mid-Pliocene geography the sensitivity of boreal spring sea ice is relatively small for modern geography. For a strong carbon dioxide forcing of 600 ppmv in simulation E600 there is still widespread sea ice in the Arctic Ocean, with the 95% isoline reaching the Davis Strait. In contrast, for simulation Eoi560 the 95%-isoline of boreal spring sea ice coverage is completely confined to the basin of the central Arctic Ocean, and it does not even intercept adjacent land regions. A similar finding of higher

sensitivity of sea ice to carbon dioxide for mid-Pliocene geography is evident from comparing boreal autumn sea ice extent for modern and mid-Pliocene geography (cf. Fig. 14 to Fig. 15). For a modern setup, boreal autumn sea ice progressively reduces, but completely ice free conditions are not reached even for 600 ppmv of carbon dioxide. For mid-Pliocene geography, on the other hand, a sea ice free Arctic is nearly present at 450 ppmv, and reached with 560 ppmv in simulation Eoi560. This is related to the impact of carbon dioxide on SST being enhanced for mid-Pliocene geography (cf. Fig. 16a,c to Fig. 16b,d). Only for high

concentrations of 600 ppmv of carbon dioxide, changes of SST in a modern geographic framework exceed those of simulation Eoi560 regionally in the Barents Sea (cf. Fig. 16d,e). For the Northern Hemisphere we find that increased SST is also present in absence of increased carbon dioxide, and hence there is a contribution of mid-Pliocene geography (cf. Fig. 16f). In the low latitude ocean we find strong modification of climate patterns in response to imposure of mid-Pliocene paleogeography and varying levels of carbon dioxide in the model. Increasing carbon dioxide has a monotonic tendency towards increased average

temperature and extent of the EqWP in the model; this is evident both for modern and mid-Pliocene geography (Table 3). In simulation Eoi560, the average EqWP temperature is increased by 2.4 °C and its extent is nearly tripled in reference to the PI state of simulation E280. A comparison of simulations with identical carbon dioxide forcing, but different geography (mid-Pliocene versus modern), reveals that there is a gain of average SST (by 0.3–0.4 °C) and extent (by 4–9·$10^6$ km$^2$) of the EqWP for mid-Pliocene geography. There is also a trend towards reduced interannual variability of the EqWP's extent when carbon

dioxide concentrations are increased. Note, however, that there is no such clear trend for average SST. Hence, increased carbon dioxide reduces the spatial variability of SST in low latitudes in the region of the EqWP.

Similar to the trends of EqWP characteristics we find a dependency of the maximum strength of the AMOC on carbon dioxide on the one hand, and to prescribed geography on the other hand (Table 4). Yet, high temporal variability of the AMOC renders many of these changes between individual simulations insignificant. Only for large changes in carbon dioxide does

a clear signal of increased AMOC emerge from internal variability in the model. In comparison to the modern setup, for simulations with the same carbon dioxide forcing, mid-Pliocene geography causes an increase of between 0.48 Sv and 1.85 Sv in the maximum strength of the AMOC. The strongest gain is present for simulations with 400 ppmv of carbon dioxide (E400, Eoi400). When increasing carbon dioxide for a given geographical setup, we find a similar trend towards increased maximum

strength of the AMOC. There is a dependency on geography, with a gain of between 0.16 Sv and 1.57 Sv for individual simulations with modern geography, and between 0.48 Sv and 2.37 Sv for simulations with a mid-Pliocene geography. We find an obvious exception to the rule "more carbon dioxide leads to higher AMOC" in that for simulation Eoi450 maximum AMOC is slightly lower than for simulation Eoi400, but internal variability ranges of AMOC overlap in both simulations. Furthermore, there is no linear relationship between AMOC strength and carbon dioxide. Strongest increase of AMOC in

the mid-Pliocene setup (2.37 Sv from simulation Eoi280 to Eoi350) occurs for a carbon dioxide increase of 70 ppmv, while an increase by 160 ppmv, from Eoi400 to Eoi560, only causes an increase in AMOC by 0.36 Sv that is within internal variability in the simulation. Overall, mid-Pliocene geography has a large impact on the simulated AMOC. While carbon dioxide increases the AMOC both for modern and mid-Pliocene geography, even at a very high level of carbon dioxide simulation E600 cannot reproduce the AMOC strength of simulation Eoi400 (cf. Fig. 17a to Fig. 6d), although their ranges of values, including internal

variability, overlap one another (Table 4). When analysing the structure of the AMOC cells (Fig. 6 and Fig. 7) we find that beyond changes of the maximum transport and the volume export across the equator there is little change in the overall structure of the streamfunction. By increasing carbon dioxide we find, for modern geography, a slight deepening of the upper cell from 2,750 m to about 3,000 m at 30°S. No similar deepening is found for mid-Pliocene geography, in which case the AMOC becomes rather shallow. This effect increases with carbon dioxide (Fig. 7). On the other hand, for both modern and

mid-Pliocene geography, carbon dioxide induced strengthening of the AMOC in the upper cell is at the expense of the strength of the lower cell that imports Antarctic Bottom Water (AABW) into the North Atlantic Ocean. Reduced AABW in the North Atlantic Ocean is linked to a decrease of mixed layer depth in the Weddell Sea with increasing concentrations of carbon dioxide (Fig. 19). Note also that the mixed layer depth in the North Atlantic to Arctic Ocean reduces with carbon dioxide, and the formation region generally migrates south-westwards. This implies that characteristics of the North Atlantic Deep Water

(NADW) also change with carbon dioxide, although the impact is not evidently represented in the structure of the upper NADW AMOC cell (Fig. 6). For mid-Pliocene geography the robustness of AABW formation in the Atlantic Ocean under the impact of increasing carbon dioxide appears to be higher than for modern geography (cf. Fig. 19 and Fig. 6).

Global patterns of precipitation are strongly influenced by carbon dioxide both for modern (Fig. 20) and mid-Pliocene geography (Fig. 21). Increasing carbon dioxide for modern geography from PI conditions to 400 ppmv causes modest changes

(Fig. 20a). In particular, there is a slight shift of the ITCZ and large scale reduction of precipitation over the equatorial Atlantic Ocean. As reduced precipitation in the low latitude Atlantic Ocean is accompanied by increased precipitation in mid to high latitudes, and in particular in the low latitude Pacific Ocean, we infer that increased carbon dioxide may lead to a net-export of freshwater by precipitation from low to high latitudes, and from the Atlantic Ocean Basin to the Pacific Ocean basin. This inference is confirmed by SSS anomalies showing that, with the exception of highest latitudes, the Atlantic/Arctic ocean basin

loses freshwater at the surface, while the Atlantic and Indian Ocean predominantly gain freshwater – the effect is amplified by

increasing the prescribed level of carbon dioxide (Fig. 22). For mid-Pliocene geography, precipitation patterns are in comparison strongly modified (Fig. 21). We find a complex pattern of latitudinal distribution of precipitation over the Atlantic Ocean, with increased rainfall in the Northern Hemisphere tropics, reduced precipitation at mid latitudes, and again increased precipitation at high latitudes (Fig. 21). For high concentrations of carbon dioxide in the mid-Pliocene simulations we again find

that these patterns are enhanced, and that there is again a pronounced export of freshwater by precipitation from the Atlantic Ocean to the Pacific and Indian Ocean basins (Fig. 21; Fig. 22). Simulations with modern and mid-Pliocene geography show that simulated precipitation patterns are similarly influenced by carbon dioxide and geography. The impact of carbon dioxide on precipitation is linked to the mean climate state, modern versus mid-Pliocene (cf. Fig. 20, Fig. 21a to 21b,c,d).

## 3.6   Comparison of reconstructed and simulated SST in dependency of mid-Pliocene carbon dioxide

Important for our understanding of the mid-Pliocene world is quantification of the agreement between reconstructions and simulations of climate conditions (e.g. Dowsett et al., 2013). In Fig. 23 and Table 6 we provide a comparison between reconstructed SST, based on PRISM3 (Dowsett et al., 2009, 2013), and simulations of SST with mid-Pliocene geography and different levels of carbon dioxide. One important feature of mid-Pliocene climate in the core simulation Eoi400 is reduced warming (in comparison to the reconstruction) in southeast Pacific and Atlantic upwelling regions off the coasts of South

America and Africa. Dowsett et al. (2013) pointed out that PlioMIP1 simulations cannot reproduce the extent of warming in upwelling regions that is suggested by proxy data. For the PlioMIP2 COSMOS mid-Pliocene core simulation this is still the case, for example there is a mismatch between reconstruction and simulation off the coast of Peru in Fig. 23c. The regional mismatch can be partly mitigated by increasing carbon dioxide to 450 ppmv and 560 ppmv (cf. Fig. 23c and Fig. 23e,f) but this leads to increasing mismatch between model simulations and reconstructions in other regions of the global oceans (Table 6).

Among simulations based on the PRISM4 mid-Pliocene paleogeography, the mid-Pliocene core simulation Eoi400 is globally in best agreement with reconstructed SST. It is followed, in this order, by simulations Eoi350, Eoi450, Eoi280, and Eoi560. If we take mid-Pliocene SST derived from the PRISM3 reconstruction and the COSMOS PlioMIP2 simulation ensemble as a baseline, we infer that the most likely carbon dioxide concentration in the mid-Pliocene is indeed the value of 400 ppmv as defined by Haywood et al. (2016) for PlioMIP2. This is bracketed by concentrations of 350 ppmv and 450 ppmv, with

slightly better global agreement for the simulation with the lower concentration of carbon dioxide. Globally, our model-data-comparison does obviously not support extremely high (560 ppmv) and extremely low (280 ppmv) concentrations of carbon dioxide in the mid-Pliocene. On the other hand, in the context of the North Atlantic to Arctic Ocean the result is very different. Here, the COSMOS PlioMIP2 simulations based on mid-Pliocene paleogeography are predominantly too cold. This mismatch is not completely removed even for the simulation with high carbon dioxide (Eoi560), but increasing carbon dioxide helps to

regionally reduce the mismatch (Fig. 23). In the region North Atlantic to Arctic Ocean there is a clear monotonic increase of mismatch starting with simulation Eoi560 and ending with simulation Eoi280 (Table 6). For the North Atlantic Ocean only, this inverse monotonic relationship between carbon dioxide and model-data mismatch is broken. There, the order of simulations with increasing model-data mismatch is Eoi400, Eoi450, Eoi350, Eoi560, Eoi280. Simulated differences in the impact of carbon dioxide for different parts of the North Atlantic to Arctic Ocean point to the inference that increased carbon dioxide

does likely mitigate other deficiencies of the model at the transition from North Atlantic to Arctic Ocean, e.g. sensitivity of sea ice in the model to mid-Pliocene boundary conditions, as suggested by Dowsett et al. (2013). Our results, however, do not hint that COSMOS might support high levels of carbon dioxide during the mid-Pliocene.

## 3.7 Comparison of reconstructed and simulated SST for emerging mid-Pliocene time slice reconstructions

While the PRISM3 reconstruction enables the relation of our PlioMIP2 simulations to previous model-data-intercomparisons (e.g. Dowsett et al., 2013) and provides reconstructions of SST for a rather large number of locations, it refers to a comparably broad time period of about 200,000 years (Dowsett et al., 2013). A need for progression from the PRISM3 time-slab to a more tightly defined mid-Pliocene time slice towards better constraining mid-Pliocene model forcing has been suggested (Haywood et al., 2013a, b). The first mid-Pliocene time slice reconstructions of SST have recently been published (Foley and Dowsett,
2019; McClymont et al., 2020), and we take the opportunity to compare the SST anomaly of the mid-Pliocene core simulation Eoi400, with respect to PI, to these. We employ the alkenone ($U^{K}_{37'}$)-based reconstruction by Foley and Dowsett (2019), with two different time windows of 10 ka and 30 ka around 3.205 Ma BP, as well as two $U^{K}_{37'}$-based, and two magnesium-to-calcium ratio (Mg/Ca)-based reconstructions, provided by McClymont et al. (2020). The $U^{K}_{37'}$ and Mg/Ca data sets by McClymont et al. (2020) each encompass a PlioVAR synthesis as well as a re-calibration based on Bayesian statistical analysis – BAYSPLINE
for $U^{K}_{37'}$, and BAYMAG for Mg/Ca. For the comparison of our mid-Pliocene model anomaly to the reconstruction of SST by McClymont et al. (2020) we employ the published temperature anomalies that refer to PI. Foley and Dowsett (2019) publish absolute SST values but do not provide anomalies with respect to PI. The latter are computed by us following the methodology by McClymont et al. (2020), who select the PI (1870–1899) average of the NOAA-ERSST5 dataset (Huang et al., 2017) at the model grid box that contains a specific core location. A comparison of modelled and reconstructed mid-Pliocene SST
anomalies is presented in Table 7 and Fig. 24. Note that the number of available data points strongly depends on the data set – it is very small for the Mg/Ca-based reconstructions (Table 7).

    When considering all available proxy records independently of geographic region, we find that neither the choice of the width of the time window, 10 ka versus 30 ka (Foley and Dowsett, 2019), nor calibration, original PlioVAR synthesis versus BAYSPLINE/BAYMAG (McClymont et al., 2020), has a major impact on the overall agreement between model and recon-
struction for a given reconstruction type, $U^{K}_{37'}$ and Mg/Ca, respectively. If, on the other hand, we focus on the North Atlantic Ocean, then there is appreciable impact of proxy-type and calibration on the agreement between simulation and geologic record (Table 7). Simulated SST fits similarly well with both $U^{K}_{37'}$-based data sets presented by McClymont et al. (2020). For $U^{K}_{37'}$-based data sets by Foley and Dowsett (2019), as well as for the Mg/Ca-based reconstructions by McClymont et al. (2020), choice of calibration and time window have an impact on agreement between simulation and reconstruction. Based
on the analysis in Table 7 our simulated North Atlantic SST agree better with the $U^{K}_{37'}$-based reconstructions than with those based on Mg/Ca. For the BAYMAG data set disagreement between our model and the reconstruction is particularly large in the Atlantic Ocean. Reasons for inhomogeneity of discord between model and reconstructions are explored based on Fig. 24.

    General findings with regard to the agreement between modelled (simulation Eoi400) and reconstructed mid-Pliocene climate (Fig. 24c), derived in Section 3.6 based on the PRISM3 reconstruction by Dowsett et al. (2009) and Dowsett et al. (2013),

are also true if we compare our model SST anomaly to the newer $U^K_{37'}$-based reconstructions by Foley and Dowsett (2019) and McClymont et al. (2020). The model cannot capture the high amplitude SST anomaly of the reconstructions in the Atlantic to Arctic Ocean, although discord between model and the time slice reconstructions is certainly reduced in comparison to the PRISM3 time-slab data. Note that we cannot exclude the possibility that better agreement between model and new data is a side effect of the lack of time slice reconstructions in some regions (cf. Fig. 23 and 24). The largest discord between model and reconstruction is present for the Foley and Dowsett (2019) data sets, independent of the reconstruction's time window. This is largely a result of the comparably mild mid-Pliocene SST anomaly modelled by COSMOS west of Svalbard, where both data sets by Foley and Dowsett (2019) include a site that suggests a very large SST anomaly, whereas that site is not included in the data sets by McClymont et al. (2020). Consequently, agreement between model and reconstruction is larger if considering the data sets by McClymont et al. (2020). For the data sets by Foley and Dowsett (2019) there is larger disagreement with our model in the west Atlantic Ocean to the Caribbean and in the Mediterranean Sea. Our model is generally too warm in the Caribbean. Agreement of model and reconstruction in the southeast Atlantic Benguela upwelling system is, based on the new $U^K_{37'}$-based reconstructions, weaker than for PRISM3. Note that the latter only provided one respective data point of low confidence. There are four reconstructions in the region based on the new data sets: for three of these the model shows a pronounced cold bias for both the reconstructions by Foley and Dowsett (2019) and McClymont et al. (2020), while the discrepancy is slightly smaller if compared to the reconstruction by McClymont et al. (2020). It has been stated that model-data mismatch in that region can be attributed at least partly to circulation rearrangements in the Mid-Pliocene, that cannot be represented by low-resolution atmosphere and ocean models (McClymont et al., 2020, and one reference therein), to which COSMOS certainly belongs.

If we focus on the two Mg/Ca-based reconstructions by McClymont et al. (2020) then we find that the disagreement between our model and the reconstruction is particularly large. The model is not able to reproduce the predominantly low SSTs in low- to mid-latitudes. Beyond the generally poor reproduction of the reconstructed Mg/Ca-based SST, our model agrees better with the PlioVAR synthesis than with the new BAYMAG calibration. We must note, though, that the larger difference between model and reconstruction in the case of BAYMAG is predominantly caused by the model's inability to produce sufficiently warm surface water in the central North Atlantic Ocean at about 40°N. Whether the poor skill of our model to reproduce Mg/Ca-inferred cooling at low latitudes is related to a model weakness, due to interpretation of the geologic record, or caused by a combination of both, must be the focus of future work. The possibility of a cold bias of Mg/Ca records, which requires further attention, has been stated (McClymont et al., 2020).

### 3.8 Large-scale shifts in Northern Hemisphere vegetation cover

Inferences from both reconstructions and simulations of mid-Pliocene vegetation include a shift of modern vegetation patterns towards the poles (Salzmann et al., 2008a, b). With COSMOS, and by means of its dynamic vegetation module, we have the unique opportunity to compare simulated and reconstructed vegetation patterns in PlioMIP2. This approach is different to the model-data comparison by Salzmann et al. (2013). There, temperature estimates based on land proxies were compared to temperature output from model simulations. Note that these simulations were forced with a reconstruction of mid-Pliocene vegetation (Haywood et al., 2010, 2011) which is not necessarily consistent with the modelled climate. Here, we investigate

large-scale shifts in vegetation cover directly, which is possible in PlioMIP2 because of the explicit support for dynamic vegetation in the modelling framework (Haywood et al., 2016). The advantage is that the combined climate-vegetation solution of COSMOS is self-consistent, and so are the respective feedbacks.

We show spatial patterns of simulated forest and grass cover for the PI reference state E280 as well as for three mid-Pliocene

model setups with carbon dioxide concentrations of 280 ppmv, 400 ppmv, and 560 ppmv (Fig. 25). In addition, we compute estimates of the spatial shift of the tree-line for all PlioMIP2 simulations with pure modern and mid-Pliocene geography (Table 8). Our results broadly confirm the findings of vegetation reconstructions that suggest a northward shift of the tree-line in the Northern Hemisphere. Vegetation establishes in the model in regions where high latitude land surface conditions are appropriate for vegetation growth. This is the case for regions outside modern and mid-Pliocene ice sheets. The obvious result

is that some regions cannot produce vegetation. For a modern geography this refers to Greenland. For mid-Pliocene geography this refers to parts of Greenland and, in the Southern Hemisphere, to East Antarctica, due to ice sheets prescribed in the model. Beyond that we find pronounced vegetation shifts that are very sensitive to the prescribed carbon dioxide forcing (Table 8). For modern geography, a carbon dioxide forcing of 560 ppmv is already sufficient to extend the tree-line towards the coast in Western Canada. For Eastern Siberia a similar finding is made for 400 ppmv. Generally, the impact of carbon dioxide on

vegetation is stronger for mid-Pliocene. In the regions of Western Canada and Eastern Siberia, forests already reach the coast of the Arctic Ocean for 400 ppmv and 350 ppmv, respectively. While the tree line does not yet reach the coast of Eastern Canada for 600 ppmv with modern geography, it does for 450 ppmv in the mid-Pliocene. Vegetation extends across Greenland in the mid-Pliocene simulation. The tree-line spreads from west to east, and the dependency on carbon dioxide is nearly linear. As trees are darker than grasses and shrubs, the northward shift of forest is linked to a pronounced albedo–feedback in high

latitudes of the Northern Hemisphere.

When considering the global distribution of simulated grass and forest as shown in Fig. 25 we find that larger northward shifts of the tree-line in the mid-Pliocene setup are enabled by the continents extending further to the north in this setup. Yet, Fig. 25 confirms the dependency of the vegetation shift on carbon dioxide concentrations. While vegetation shifts are pronounced at high latitudes of the Northern Hemisphere, they also occur in other regions. Across the mid-Pliocene West

Antarctic we find a monotonic southward expansion of grassland. Yet, our model cannot confirm the presence of trees on Antarctic mainland. Trees expand southwards across Patagonia, but tree cover does not reach the Antarctic Peninsula. Under the impact of increasing carbon dioxide, forests are in some parts of South America replaced by grass vegetation types. Extension of North American and Eurasian forests towards the north is at the expense of increased grass cover at lower latitudes. For southern and southeastern Asia there are further prominent modifications of tree cover. With increasing carbon dioxide in the

mid-Pliocene we find that trees expand southwards, reaching, and partly covering, the Himalayas. While trees are present in the Sahel and on the Arabian Peninsula in the mid-Pliocene, where they extend northwards with increasing concentrations of carbon dioxide, we see an opposite effect in southern to central Europe. For Africa south of the equator the result is ambiguous. While increasing carbon dioxide initially leads to more widespread forest cover, forests in some regions are vulnerable to further increased carbon dioxide. For Australia we find that mid-Pliocene paleogeography and moderate carbon dioxide (280 ppmv

and 400 ppmv) lead to an initial die-back of forests. This trend is reversed for high carbon dioxide, whereby the extent of tree cover returns from the northeast towards the centre of the continent.

## 3.9   Factorisation of drivers of mid-Pliocene warmth

We have already addressed the impact of mid-Pliocene geography and carbon dioxide on the steepness of the simulated meridional temperature gradient (Section 3.2). Yet, by considering simulations Eo280, Eo400, Ei280, and Ei400, together with the already presented simulations E280, E400, Eoi280, and Eoi400, we are able to clearly separate the impact of different contributors to mid-Pliocene warmth. Towards quantification of individual components of the mid-Pliocene temperature anomaly we follow the forcing factorisation outlined by Haywood et al. (2016) that is based on SATs of individual simulations and shown in equations 1–3. Relative warmth of the mid-Pliocene is split into contributions from carbon dioxide ($dT_{CO2}$; equation 1), topography ($dT_{orog}$; equation 2), and ice sheets ($dT_{ice}$; equation 3). Information necessary for factorisation is available in Table 2, column SAT. For brevity, we follow the nomenclature by Haywood et al. (2016) and denote SAT of a specific simulation simply by the simulation name.

$$dT_{CO2} = \frac{1}{4}\Big( (E400 - E280) + (Eo400 - Eo280) + (Ei400 - Ei280) + (Eoi400 - Eoi280) \Big) = 2.23^{\circ}\text{C} \tag{1}$$

$$dT_{orog} = \frac{1}{4}\Big( (Eo280 - E280) + (Eo400 - E400) + (Eoi280 - Ei280) + (Eoi400 - Ei400) \Big) = 0.91^{\circ}\text{C} \tag{2}$$

$$dT_{ice} = \frac{1}{4}\Big( (Ei280 - E280) + (Ei400 - E400) + (Eoi280 - Eo280) + (Eoi400 - Eo400) \Big) = 0.38^{\circ}\text{C} \tag{3}$$

Note that the total change in global average SAT from E280 to Eoi400 ($\Delta T$ =3.37 °C) is slightly smaller (by 4.1%) than the respective change that arises from summing over individual warming components $dT_{CO2}$, $dT_{orog}$, $dT_{ice}$. Hence, there is an inconsistency in derived results, albeit a small one. Factor analysis of contributors to mid-Pliocene warmth in COSMOS roughly suggests a partition of the total effect $\Delta T$, mid-Pliocene core simulation versus PI, into contributions by $dT_{CO2}$, $dT_{orog}$, and $dT_{ice}$ as given by equation 4.

$$\Big(dT_{CO2}, dT_{orog}, dT_{ice}\Big) \approx \Big(\frac{2}{3}, \frac{1}{4}, \frac{1}{8}\Big) \tag{4}$$

## 3.10   The impact of absent Northern Hemisphere gateways on mid-Pliocene climate

The mid-Pliocene has been characterised by differences in specific gateways with regard to modern geography (Dowsett et al., 2016; Haywood et al., 2016). Of these, a combination of three Northern Hemisphere gateways and coastal seas (Bering Strait, Canadian Arctic Archipelago, Hudson Bay) has been found to impact on the state of the North Atlantic Ocean in that they amplify positive mid-Pliocene temperature anomalies if absent (Otto-Bliesner et al., 2017b). As the mid-Pliocene configuration

of these gateways tends to decrease the model-data mismatch evident in the North Atlantic Ocean (Dowsett et al., 2013), we explicitly test here the impact of the respective details in mid-Pliocene paleogeography on climate in COSMOS. To this end we conduct an additional simulation Eoi400_GW, that is beyond the official PlioMIP2 simulation ensemble. It is identical to the core mid-Pliocene simulation Eoi400, with the exception that Hudson Bay, Bering Strait, and Canadian Arctic Archipelago

are present in the model setup. The geographic change in Eoi400_GW has only a minor impact on large scale mid-Pliocene climate characteristics as evident from a comparison with simulation Eoi400 (Table 2). There is a moderate increase in global average SAT if modern states of the gateways are considered in the mid-Pliocene (+0.06 °C). Evaporation and precipitation are nearly unchanged, and cloud cover as well as albedo are similar on a global scale. If we consider the ocean realm, then we find that the gateway change leads to a slight cooling of global average SST (-0.12 °C) and to increased sea ice extent in the

Northern Hemisphere, both for FMA and ASO. In contrast, in the Southern Hemisphere the gateway change causes reduced sea ice cover.

Analysing the impact of gateway reconfiguration on spatial patterns of SAT, SST, total precipitation, and strength of the AMOC we find a diverse picture. In comparison to the impact, that the presence of mid-Pliocene versus modern ice sheets and carbon dioxide forcing have on SAT (Fig. 26) and total precipitation (Fig. 27), we find a comparably minor effect of the

state of Northern Hemisphere gateways on SAT that is in many regions of the world statistically insignificant (cf. Fig. 26f). Yet, it is evident that the presence of a modern configuration is contrary to the overall tendency of a comparably warm mid-Pliocene Atlantic Ocean, while leading to additional warming in particular on the North American continent towards the Canadian Arctic Archipelago and Greenland (cf. Fig. 26f to Fig. 26e; Fig. 28a-c). Seasonal analysis illustrates that the North Atlantic Ocean realm is, for both summer and winter, cooler if the gateways are implemented in their modern state in contrast

to the configuration with PRISM4 boundary conditions (cf. Fig. 28a-c). For boreal summer, the tendency towards a cooling by modern gateways extends across all longitudes, with pronounced extremes over North America and North Pacific Ocean, regionally exceeding 1 °C. The impact of the gateway change towards modern is weaker for precipitation than for temperature. Yet, we note that over the Sahel, the North Atlantic, and the tropical Atlantic Ocean there is drying that opposes the general trend towards more rainfall in the mid-Pliocene (cf. Fig. 27f to Fig. 4). This effect is again strongest during boreal summer

(Fig. 28).

Modern gateways in an otherwise mid-Pliocene model setup tend to reduce the strength of the AMOC in comparison to simulation Eoi400 (cf. Fig. 17b to Fig. 6; Table 4) and lead to a more pronouncedly shallow upper cell (cf. Fig. 18b to Fig. 7c). Their effect on the extent of the EqWP vanishes (Table 3). A PlioMIP2 mid-Pliocene core simulation with modern states of the Bering Strait, Canadian Arctic Archipelago, and Hudson Bay (Eoi400_GW) provides global SST that is very similar to that in

the respective simulation with full PRISM4 configuration (Eoi400), as evident from Fig. 23c,d. Hence, the impact of gateway-related SST change in a comparison between simulated and reconstructed SST (PRISM3; Dowsett et al., 2009, 2013) is minor, but does indeed increase the model-data mismatch in the simulation with modern gateways – globally, as well as in North Atlantic Ocean to Arctic Ocean (Table 6). This points to the conclusion that the reconstructed mid-Pliocene state of the Bering Strait, Canadian Arctic Archipelago, and Hudson Bay indeed is indeed one contributor to reduced discord between COSMOS

simulation and PRISM3 reconstruction in the high-latitude Atlantic-Arctic Ocean realm in PlioMIP2. In other words, and

reverting the argument: COSMOS supports evidence from PRISM4 that Northern Hemisphere gateways differed in their mid-Pliocene state from modern conditions. Yet, as illustrated, the large scale effects of the gateways are in comparison to other drivers of mid-Pliocene climate rather small.

## 4   Discussion

### 4.1   Comparison of COSMOS mid-Pliocene global SAT anomaly to PlioMIP2 and PlioMIP1 models

One aim of the PlioMIP is to identify differences and similarities in the different models' representation of mid-Pliocene climate patterns. Analysing the change of simulated climate characteristics over time, i.e. from PlioMIP1 to PlioMIP2, may reveal whether model sensitivity to boundary conditions has changed due to ongoing model development or increased spatial resolution of the simulations (Haywood et al., 2020). Comparing simulated climate characteristics across models provides evidence of model-dependency of specific mid-Pliocene climate patterns (e.g. Haywood et al., 2013a). According to Haywood et al. (2013a) and Haywood et al. (2020), COSMOS is one of seven models that contribute to both PlioMIP1 and PlioMIP2. For this model there has been no further development or change in spatial resolution since the work by Stepanek and Lohmann (2012). Hence, COSMOS allows us to examine the extent to which changes in modelling methodology from PlioMIP1 to PlioMIP2, and in boundary conditions from PRISM3 (Dowsett et al., 2009, 2013) to PRISM4 (Dowsett et al., 2016), have impacted on the modelled mid-Pliocene climate. A detailed analysis of mid-Pliocene climate patterns simulated by COSMOS in comparison to other models is beyond the scope of this manuscript and shall be the focus of future research. Deeper insight into the impact of different characteristics of PRISM3 versus PRISM4 boundary conditions on the mid-Pliocene climate simulated by COSMOS is provided by Samakinwa et al. (2020). Here, we provide only a brief comparison of global average annual mean SAT anomaly simulated by COSMOS to that of other models. We consider the relation of simulated temperature anomaly to model ECS and hence explore a first-order characteristic of mid-Pliocene climate both for the PlioMIP2 mid-Pliocene core simulation and its analogue in PlioMIP1 (experiment 2; Haywood et al., 2011). Our approach is based on the work by Haywood et al. (2013a), Hargreaves and Annan (2016), and Haywood et al. (2020).

An important characteristic of COSMOS is relatively large sensitivity to carbon dioxide. In PlioMIP1, COSMOS was the model with the highest published CS. It also produced the largest global average SAT anomaly in the coupled atmosphere-ocean model ensemble and provided the warm end member of mid-Pliocene climate simulations. The PlioMIP1 model with second highest CS (MIROC4m) produced the second highest mid-Pliocene temperature anomaly (Haywood et al., 2013a). In PlioMIP2 the situation has changed. While COSMOS is still among those models that are relatively sensitive to carbon dioxide, importance of a high ECS for the simulated mid-Pliocene SAT anomaly appears less clear than in PlioMIP1. Results by Haywood et al. (2020) show that the PlioMIP2 models with highest ECS (in order from highest to lowest ECS: CESM2, IPSLCM6A, COSMOS, CESM1.2, IPSLCM5A, MIROC4m, and IPSLCM5A2) are not necessarily those models that produce largest global annual mean mid-Pliocene SAT anomalies (in order from highest to lowest temperature anomaly: CESM2, EC-EARTH3.3, CCSM4-2deg, CESM1.2, CCSM4-UoT, IPSLCM6A, COSMOS). With CCSM4-2deg, CCSM4-UoT, CESM1.2, and EC-EARTH3.3 there are four PlioMIP2 models that produce larger mid-Pliocene SAT anomalies at lower ECS than

COSMOS. While CESM1.2 and CESM2 clearly represent examples of models that produce large relative warming at large ECS, IPSLCM5A generates a warming that is below the PlioMIP2 ensemble mean at an ECS that is clearly above that of the average PlioMIP2 model. Results for COSMOS show that the model provides an interesting case in that its ECS in PlioMIP2 is larger than the CS reported for PlioMIP1. Yet, it simulates a mid-Pliocene SAT anomaly that is 0.3 °C below that produced in

PlioMIP1. While the prescribed concentration of carbon dioxide is larger in PlioMIP1, the climatic effect caused by a reduction of carbon dioxide by 5 ppmv is too small (-0.05 °C) to account for the relatively cooler Mid-Pliocene climate in PlioMIP2. This is shown by a simulation by Samakinwa et al. (2020) that is similar to Eoi400 but employs the carbon dioxide concentration of PlioMIP1. As COSMOS has not changed since PlioMIP1, reasons for this model behaviour must be related to changes in modelling methodology from PlioMIP1 to PlioMIP2 – among which is a switch from prescribed mid-Pliocene vegetation to

vegetation dynamics.

Knorr et al. (2011) suggest that, for generation of a warm Late Miocene (11 Ma–7 Ma BP) climate with COSMOS, albedo changes, that are due to reconstructed vegetation cover changes (prescribed), are a key contribution, especially at relatively low PI atmospheric carbon dioxide levels (cf. Knorr and Lohmann, 2014). In PlioMIP1, Stepanek and Lohmann (2012) employed a static implementation of a reconstruction of mid-Pliocene vegetation. In this model setup the simulated mid-Pliocene tem-

15 perature anomaly was higher than it is in the PlioMIP2 setup with dynamic vegetation. Sensitivity studies that test the impact of dynamic vegetation on the mid-Pliocene climate of COSMOS are not considered for PlioMIP2 and are beyond the scope of the already extensive simulation ensemble provided with this model. Nevertheless, it is interesting to note that adjustments of soil albedo via a soil-dynamics scheme, leading towards consistency with simulated vegetation cover, provide a positive feedback in COSMOS (Stärz et al., 2016), which is expected to result in a warmer global mean SAT anomaly for COSMOS in

PlioMIP2. This could possibly reduce the difference in the simulated PlioMIP1 and PlioMIP2 SAT anomaly.

The relationship between simulated mid-Pliocene SAT anomaly and ECS in COSMOS may change from PlioMIP1 to PlioMIP2 due to changes in modelling methodology, including the addition of dynamic vegetation. Below, using reasonable assumptions, we investigate how much the change in this relationship impacts on the PlioMIP2 ensemble relationship between model sensitivity to carbon dioxide and modelled temperature anomaly. We illustrate this by summarising inferences by Hay-

25 wood et al. (2013a) and Haywood et al. (2020) on the PlioMIP1 and PlioMIP2 model ensembles, respectively. The revised study by Haywood et al. (2020) contains an updated PlioMIP2 model ensemble. Here, we consider both the 15 member ensemble by Haywood et al. (2020, Fig. S2 of our supplement) and the updated 16 member ensemble of the revised study by Haywood et al. (2020, Fig. S3 of our supplement) in our analysis. We study changes in the relationship between model sensitivity to carbon dioxide and the simulated mid-Pliocene anomaly of global annual average SAT for COSMOS, in relation to

30 other PlioMIP1 and PlioMIP2 models, based on four different assumptions. First, we exclude two and three models with high ECS from the analysis (Fig. S2c and Fig. S3c). Second, we change COSMOS' ECS to its PlioMIP1 value (Fig. S2d and Fig. S3d). Third, we carried out a test to investigate the impact that COSMOS would have on the model ensemble if the simulated mid-Pliocene PlioMIP2 SAT were as high as in PlioMIP1 (Fig. S2e and Fig. S3e). Fourth, we compute a regression where we assume COSMOS provides a larger temperature anomaly at a reduced CS as in PlioMIP1 (Fig. S2f and Fig. S3f). Fifth (Fig.

S2g and Fig. S3g), we compare the impact, that all these assumptions with regard to COSMOS would have in comparison to

an exclusion of models from our ensemble analysis that are known to reduce significance of the correlation between modelled SAT anomaly and model sensitivity to carbon dioxide (Haywood et al., 2020). For reference, we also show plots similar to those presented by Hargreaves and Annan (2016) for all PlioMIP1 models (Fig. S2a, S3a), and by Haywood et al. (2020) for all PlioMIP2 models (Fig. S2b, S3b).

We find that the linear relationship between CS/ECS and simulated global annual mean mid-Pliocene SAT anomaly is indeed more pronounced in PlioMIP1 (Fig. S2a,b; Fig. S3a,b). While the p-value of the regression for PlioMIP1 and for the 15 member PlioMIP2 ensemble is so large that we fail to reject the null-hypothesis of presence of a zero slope, which we interpret as a lack of any dependency of SAT on model ECS, we find that the slope is larger (0.93 °C global annual SAT warming per degree CS) in the PlioMIP1 model ensemble than in PlioMIP2 (0.54 °C). This is confirmed in the revised 16 member

PlioMIP2 ensemble where the p-value is small enough (<0.05) to reject the null hypothesis. Note that CS and ECS relate modelled warming to a doubling of carbon dioxide with regard to PI, an amplitude that is not reached in the mid-Pliocene simulations of either PlioMIP2 core or experiment 2 of PlioMIP1. Hence, the slope is below unity. If we remove models with high ECS (IPSLMC6A and COSMOS, Fig. S2c; CESM2, IPSLMC6A, and COSMOS, Fig. S3c) from the regression, we find increased impact of model ECS on temperature change in the ensemble via increased slope, but also an increased

p-value. Adopting the lower CS of PlioMIP1 for COSMOS (Fig. S2d; Fig. S3d) also leads to increased slope but unchanged p-value. For the 15 member ensemble this implies that ensemble sensitivity to ECS is increased, but the relation between simulated temperature anomaly and model sensitivity to carbon dioxide is still not significant. If we hypothetically assume a COSMOS simulation with increased temperature anomaly as in PlioMIP1 (Fig. S2e; Fig. S3e), this increases the slope and reduces the p-value, but for the 15 member ensemble still does not cause a slope that is significantly different from zero. If,

on the other hand, we assume a large temperature anomaly at lower CS of COSMOS, as in PlioMIP1 (Fig. S2f; Fig. S3f), the finding is similar. To conclude previous assumptions: only when we remove models (excluding COSMOS) which do not show a significant relationship between ECS and SAT, do we find a relation between model ECS and simulated SAT anomaly with increased slope in the 15-member ensemble that is significant (Fig. S2g), and an increased slope with vanishing p-value in the 16 member ensemble (Fig. S3g). In summary, while changes in modelling methodology and boundary conditions from

PlioMIP1 to PlioMIP2 causes COSMOS to exhibit opposing differences in ECS and SAT anomaly, these characteristics do not have an appreciable influence on the ECS/SAT anomaly relationship of the PlioMIP2 ensemble. Other models have a larger impact.

Results of this model-intercomparison of CS/ECS and SAT anomalies in PlioMIP2 and PlioMIP1 simulations have implications for both COSMOS and other PlioMIP2 models. In PlioMIP2, model characteristics other than ECS, and the potential

impact of PRISM4 boundary conditions, may be more important for the simulated amplitude of global annual average SAT anomaly. As COSMOS has not been further developed since PlioMIP1 and a change of spatial resolution has not been implemented, for this model we can clearly identify updated modelling methodology and PRISM4 boundary conditions as a cause for reduced dependency of mid-Pliocene warming on model ECS. The difference of PRISM4 boundary conditions from modern geography is certainly larger than it is in the case for PRISM3 (Dowsett et al., 2009, 2013, 2016). Improved contributions from

boundary conditions to warming include a reduced Greenland Ice Sheet and adjusted gateway states in PlioMIP2. While the

latter have been shown to improve warming in the North Atlantic Ocean (Otto-Bliesner et al., 2017a), we note that the impact on global mid-Pliocene temperature is rather small in COSMOS. Whether these inferences are also robust for other models must be studied in the future. To this end refined sensitivity simulations that test, for example, the dependency of mid-Pliocene warming on the gateway state, or the impact of dynamic versus static vegetation, are necessary. Such work could be considered

for PlioMIP3. We suggest that the inference of potentially reduced ties between simulated mid-Pliocene temperature and model ECS in some PlioMIP2 models is of particular relevance to the question: "To what extent may the mid-Pliocene serve as an analogue to future climate?" On the short- to medium-term, climate projections are mostly controlled by increased levels of greenhouse gases in the atmosphere. While carbon dioxide sensitivity is a major driver of simulated mid-Pliocene temperature, certainly for COSMOS, other contributors appear to have become more important in PlioMIP2. We will further explore the

applicability of mid-Pliocene climate, as simulated with COSMOS, to future climate states in Section 4.3.

## 4.2    The added value of COSMOS mid-Pliocene simulations in PlioMIP2

Based on the PlioMIP2 simulation ensemble, which is greatly extended in comparison to PlioMIP1 (Haywood et al., 2010, 2011, 2016), we derive valuable results that were not retrievable by means of our efforts in PlioMIP1. Considering dynamic vegetation in the model as an additional subsystem of the Earth has extended the ways in which COSMOS simulations of

the mid-Pliocene can be compared to reconstructed climate-geographical patterns of the mid-Pliocene. The added value of analysing the relative contribution of different drivers of mid-Pliocene climate patterns and dynamical climate features (e.g. AMOC) have enabled a much more thorough investigation of the climate state of the mid-Pliocene. Thanks to this effort, interpretation of mid-Pliocene climate in the light of future climate change is simplified. By employing the same model as for PlioMIP1, albeit using dynamic vegetation as an additional model component that was not permitted in PlioMIP1, we provide

an extensive data set that constrains the PMIP triangle of uncertainty (Haywood et al., 2013a, 2016) at one corner: modelling uncertainty has been excluded as much as practically feasible. As a result, a more detailed analysis of the remaining two triangle corners, that represent boundary condition uncertainty and data uncertainty, is in reach in the framework of PlioMIP2 – in particular, if this task includes results from other models that have been employed both in PlioMIP1 and PlioMIP2.

     In extending our mid-Pliocene climate research from PlioMIP1 to PlioMIP2 we have been able to substantially increase

inferences in the context Pliocene4Pliocene with respect to PlioMIP1. While many large scale climate features derived with COSMOS for the most likely mid-Pliocene model setups (experiment 2 in PlioMIP1, simulation Eoi400 in PlioMIP2) are very similar, we find differences in details as shown by Samakinwa et al. (2020). In particular, by means of our additional sensitivity study with modern states of Northern Hemispheric gateways in an otherwise mid-Pliocene core simulation setup, we have shown that, as proposed by others (Otto-Bliesner et al., 2017b), the mid-Pliocene geography of these regions acted towards

reducing the mismatch between reconstructed and simulated SST – a finding that was elusive in PlioMIP1. Yet, we also find that the global effect of the mid-Pliocene gateway state is rather small, and, according to our results, certainly did not contribute to a pronounced reduction in the meridional temperature gradient between equator and poles.

     Providing simulations with a similar model to both PlioMIP1 and PlioMIP2 has also enabled us to analyse the impact of changes in boundary conditions, modelling methodology, and last but not least, the availability of emerging time slice

reconstructions for the mid-Pliocene on agreement between model and inference from the geologic record. The model-data disagreement is reduced in PlioMIP2 in many regions of the North Atlantic Ocean based on the new time slice reconstructions by Foley and Dowsett (2019) and McClymont et al. (2020). On the other hand, a larger number of SST reconstructions of higher confidence are now available in regions where the model is known to not reproduce the reconstructed amplitude of mid-Pliocene

climate anomalies as evident from the geologic record. This is the case in particular for the Benguela upwelling system and in the high latitudes of the Atlantic Ocean to the Arctic Ocean. Consequently, and also with respect to the updated reconstructions, discord between model and reconstructions has not completely disappeared. Newer models with higher spatial resolution may be able to mitigate the discord that results from low model resolution (e.g. McClymont et al., 2020). Nonetheless, similarly to the PlioMIP2 model ensemble (Haywood et al., 2020), COSMOS is able to reproduce reconstructions of mid-Pliocene SST

in many regions of the world, both for the PRISM3 time-slab reconstruction and newly emerging time slice reconstructions. This gives confidence that large scale mid-Pliocene patterns, as modelled by COSMOS, are a reasonable approximation to mid-Pliocene climate as interpreted from the geologic record.

In PlioMIP2 we have been able to confirm with our model that carbon dioxide may have been the predominant driver of mid-Pliocene warmth (Pagani et al., 2010) while topography and ice sheet changes provide a comparably modest contribution.

We also confirm a previous statement that the mid-Pliocene is unlikely to represent a world of doubled PI concentration of carbon dioxide (Raymo et al., 1996). The impact of carbon dioxide in an equilibrium climate is larger than fast feedbacks, in agreement with findings by Lunt et al. (2010). Our results also suggest that carbon dioxide at, or close to, PI levels was unlikely during the mid-Pliocene. We derive an intensified meridional streamfunction in the Atlantic Ocean, that has been found by some of the PlioMIP1 models (Zhang et al., 2013a) and proposed before (Raymo et al., 1996). As COSMOS is a

model with comparably high CS (Haywood et al., 2013a), and as carbon dioxide is the main driver of mid-Pliocene warmth in our model (this study), our estimates of mid-Pliocene global average temperature anomalies are, as in PlioMIP1, at the upper end of the likely value range (Jansen et al., 2007; Haywood et al., 2013a; Raymo et al., 1996) and above the PlioMIP2 multi model mean (Haywood et al., 2020). Pagani et al. (2010) state that, based on proxy-data, mid-Pliocene ESS was at a minimum $7.1\pm1\,°C$, which overlaps with our result of $5.9\pm0.5\,°C$. Hence, the model is able to reproduce mid-Pliocene ESS as interpreted

from the geologic record. Yet, model sensitivity to carbon dioxide, and to related geographic change, is at the lower end of the proxy-based estimate. The ratio of ESS to ECS in the mid-Pliocene for our model is 1.2, which is less than the values in the PlioMIP1 simulation – 1.5 for the model ensemble and 1.7 for COSMOS (Haywood et al., 2013a). A finding based on our additional simulation Eoi560, that is beyond the official PlioMIP2 simulation ensemble, is that in COSMOS, carbon dioxide is less effective in reducing the mid-Pliocene meridional temperature gradient, and in increasing global average SAT,

when raising the concentration from 400 ppmv to 560 ppmv in the mid-Pliocene, in comparison to the initial change in carbon dioxide, from 280 ppmv to 400 ppmv. We have discussed in Section 4.1 that at least part of this effect is due to a change in modelling methodology in PlioMIP2, including representation of vegetation dynamics. Whether this result faithfully presents a characteristic of the real world, or whether it is also imprinted into our simulations due to the lack of further missing dynamic components in our model (in particular ice sheets and aerosols) must remain for now an open question. This statement is also

related to the finding that COSMOS does not show pronounced state-dependency of ECS – that is found by Hansen et al.

(2013), where CS increases for warmer climates. We find that COSMOS shows a slight increase of the ECS from modern to mid-Pliocene geography, but that result is not significant.

From the perspective of Pliocene4Future, COSMOS recommended itself – due to its high CS – in the PlioMIP1 for the study of potential mid-Pliocene analogies to future climate under the assumption of comparably strong projected warming. The latter may also be interpreted as being pessimistic with regard to the ability of humankind to either mitigate anthropogenic emissions of greenhouse gases or to limit their climatic impact, and is consequently of societal relevance. Results of PlioMIP2 show that mid-Pliocene climate computed with COSMOS is still towards the warmer end of the model ensemble. Yet, the results also point to the possibility that sensitivity of the Earth system to mid-Pliocene boundary conditions, including carbon dioxide, is likely even larger than represented by COSMOS. This once more underlines the fact that a climate, that is in equilibrium with 400 ppmv of carbon dioxide, may look quite different from conditions observed and felt today, when levels of greenhouse gas concentrations in the atmosphere are likely to be already higher than those during the mid-Pliocene.

## 4.3 Implications of COSMOS PlioMIP2 simulations for future climate

The mid-Pliocene has been characterised by carbon dioxide concentrations that are elevated with regard to PI levels, and likely similar to modern conditions (about $407$ ppmv in 2018, Friedlingstein et al., 2019), as indicated by various authors (Kürschner et al., 1996; Raymo et al., 1996; Seki et al., 2010; Pagani et al., 2010; Haywood et al., 2010, 2016). The mid-Pliocene combines estimates of carbon dioxide levels in the atmosphere, that are similar to today, with a much warmer-than-present climate state. It is hence not surprising that this time period has been suggested as a potentially suitable time slice towards the study of large scale climate patterns and mechanisms which may be representative of, or at least similar to, future warmer-than-present climate states in the context of ongoing global warming caused by human activity (e.g. Jansen et al., 2007). On the one hand, the mid-Pliocene represents the most recent geologic time interval when global average surface temperatures were persistently elevated above PI conditions, while geographic conditions at that time were characterised by a land surface that had already reached its modern shape in many aspects (Jansen et al., 2007). This simplifies the interpretation of mid-Pliocene climate signals in the context of Pliocene4Future. On the other hand, there are marked differences between the paleogeography of the mid-Pliocene and of modern day, or rather the near-future. In particular, the latter will be different with regard to details in coastlines, state of ocean gateways, and height and extent of ice sheets (e.g. Dowsett et al., 2016; Haywood et al., 2016). Difference in ice sheets will last at least until the modern cryosphere has reached a new equilibrium under the influence of elevated carbon dioxide. Different states of ocean gateways have been shown to cause additional warmth in the mid-Pliocene North Atlantic realm (Otto-Bliesner et al., 2017b, this study), which is not an effect to be expected in the foreseeable future. Furthermore, current trends in carbon dioxide levels are far beyond the natural variability that occurred during the mid-Pliocene. The Pliocene concentration of carbon dioxide is, in comparison to anthropogenically forced changes since the PI, relatively stable as inferred from records of carbon dioxide by Pagani et al. (2010), Seki et al. (2010), and Bartoli et al. (2011), with a potential change of about 50 ppmv only in the earlier part of the mid-Pliocene based on the reconstruction by Seki et al. (2010). From the viewpoint of carbon dioxide related radiative forcing it can be assumed that the state of mid-Pliocene ice sheets was, in comparison to that during the Anthropocene, relatively stable.

Hence, making inferences from modelled or reconstructed climate conditions of the mid-Pliocene with respect to future elevated temperatures, that are on the short-term mostly driven by anthropogenic forcing, must be done with caution. This has been stated by Haywood et al. (2009b), who pose the question whether time can be reversed for the past to become the key to the future. It is relevant to test individual contributions of various drivers, that act at different time scales, to the overall mid-Pliocene climate state. Such analysis may help to identify those mid-Pliocene warmth aspects that may be of relevance in the near or more distant future.

The focus of PlioMIP1 was on simulations that provide a climate state based on a best-knowledge representation of mid-Pliocene geography and climate forcing. These are well-suited for the study of large scale features of the climate of the mid-Pliocene (Haywood et al., 2010, 2011). Analyses in the framework of the PlioMIP1 have already provided insights into a warmer-than-present climate state by means of model-data (Haywood et al., 2013a; Dowsett et al., 2013; Salzmann et al., 2013) and model-model (Haywood et al., 2013a; Zhang et al., 2013a, b; Hill et al., 2014; Howell et al., 2016) intercomparison studies. Across the model ensemble, PlioMIP1 produced a simulated mid-Pliocene climate that is overall comparably warm and wet (Haywood et al., 2013a) and bears similarity in many aspects with reconstructions of SST, although there are also pronounced model-data discords (Dowsett et al., 2013, again confirmed in our study). It has been shown that carbon dioxide contributes to mid-Pliocene warmth, but is not the only responsible agent, with albedo effects related to the cryosphere being the most important factor for high latitude climate (e.g. Hill et al., 2014). Simulated PlioMIP1 climate patterns include strongly reduced sea ice, both in the Northern (Howell et al., 2016) and Southern (Hill et al., 2014) Hemisphere. These results are, in general, also reproduced by COSMOS within PlioMIP2. In the context of Pliocene4Future it is remarkable that current climate trends show sea ice reduction in the Arctic over the last few decades (Meier et al., 2014), and more recently also in the Southern Ocean (Parkinson, 2019), hinting at increasing similarity between modern-to-future and mid-Pliocene climate with regard to these aspects. It is noteworthy that Burke et al. (2018) have stated that for a high emission scenario (RCP8.5) mid-Pliocene climate may be realised again as early as 2030 CE in transition towards even warmer climates of earlier times of the Cenozoic, and that 10 years later, and for an intermediate scenario (RCP4.5), mid-Pliocene climate may become the new stable background state.

In PlioMIP2 we extend this work and distinguish between the warming components, carbon dioxide, ice sheets and topography, courtesy of the simulation ensemble proposed by Haywood et al. (2016). Furthermore, we study the impact of the state of Northern Hemisphere gateways on mid-Pliocene warmth by means of an additional sensitivity study that is beyond the official PlioMIP2 model ensemble. Results from our factor analysis show that indeed carbon dioxide is the dominant driver of mid-Pliocene warmth in the model, with the total contribution from mid-Pliocene topography and ice sheets being about one-third. Quantification of the singular contribution of Northern Hemisphere gateway changes to the mid-Pliocene climate state, as modelled by COSMOS, shows that this leads to an amplification of mid-Pliocene warmth, but only regionally in the North Atlantic to Arctic Ocean. In principle this information is useful for correcting the modelled mid-Pliocene climate state for gateway effects in order to obtain a simpler interpretation of the model output in the framework of Pliocene4Future. Yet, as it currently stands, similar information is not available from other PlioMIP2 models, impeding a model-model intercomparison. For future phases of the PlioMIP, an additional mid-Pliocene sensitivity simulation with a modern state of relevant

ocean gateways may further improve our understanding of the applicability of mid-Pliocene climate simulations as a tool for the study of future climate in those regions, where ocean gateways are found to be a relevant driver of mid-Pliocene conditions.

When discussing the applicability of results from our mid-Pliocene equilibrium simulation ensemble for inferences on future climate, we stress that the time-frame considered in the term *future climate* certainly plays a role. For example, we find a stable and strong AMOC for high carbon dioxide concentrations of up to 600 ppmv (modern geography) and 560 ppmv (mid-Pliocene geography). This finding is at first sight contrary to the expectation that thermohaline circulation of the 21st century, at carbon dioxide concentrations in the range expected for lower scenarios of anthropogenic emissions (RCP2.6, RCP4.5; van Vuuren et al., 2011), that are employed in our simulation ensemble, will be very likely reduced with respect to modern conditions (Collins et al., 2013). Yet, the results presented here are from equilibrium integrations of the model. In such types of simulations model physics had the opportunity to adapt to the new climate forcing, over a time period of 1,850 years in our case (Table 1). This modelling methodology may be appropriate for inferences on the likely shape of climate in a more distant future, but not at decadal to centennial time scales. If we consider the results of the CMIP6 simulation 1pctCO2, forced with a monotonic increase of the carbon dioxide concentration, then we indeed find appreciable reduction of the simulated AMOC that is ongoing for at least as long as carbon dioxide concentrations increase. A similar finding is present during adaptation of the model to the new carbon dioxide forcing in simulation abrupt4xCO2 (Fig. 11). Hence, our simulations suggest that a future climate, that is in equilibrium with a higher carbon dioxide forcing, will, like the mid-Pliocene in our model, also have a stronger AMOC than at present, but only after a new quasi-equilibrium climate state has been reached. Hence, characteristics of AMOC dynamics of mid-Pliocene climate as simulated by us are not directly transferable to the near future, but they may be transferable to the more distant future.

By means of COSMOS we find in PlioMIP2 further effects for which simulated or reconstructed mid-Pliocene climate must be corrected for an interpretation in terms of the near to distant future climate. We find that Arctic sea ice is more vulnerable to changes in carbon dioxide for mid-Pliocene geography. This is related to polar amplification being stronger in this setup. In this context it seems that modern geography may provide a dampening effect to carbon dioxide related changes. We also show that simulated vegetation patterns change more in the mid-Pliocene. This is also confirmed if we consider that, for modern geography, vegetation cannot grow on Greenland and Antarctica, as a result of ice sheet cover. One of our results is that carbon dioxide strongly influences precipitation both for modern and mid-Pliocene topography. Mid-Pliocene precipitation is influenced by both the geography and carbon dioxide. Similar findings in COSMOS hold for AMOC, SST, and for the state of the EqWP. In summary, in the framework of Pliocene4Future we find that some aspects of mid-Pliocene climate may be more directly transferred to future conditions, while others need to be more carefully examined. We note that our findings are based on a model where further dynamic components of the Earth system are not available in PlioMIP2, in particular dynamic ice sheets.

## 4.4 Shifts in vegetation patterns in mid-Pliocene and Future

Based on proxy data, it has been inferred that high latitude vegetation was distinctly different from modern in the mid-Pliocene (Salzmann et al., 2008a, b). Ballantyne et al. (2006) show that trees were present over Northern Canada and Greenland. With

COSMOS we have been able to simulate mid-Pliocene vegetation changes for the first time in PlioMIP2. Our results show major shifts in vegetation patterns, both under the impact of mid-Pliocene topography and elevated global average temperature as a result of increased atmospheric concentrations of carbon dioxide. Vegetation shifts are found both for modern and mid-Pliocene topography. While the vegetation model employed by COSMOS is relatively simple, considering only eight different PFTs, our results are broadly in agreement with expected vegetation patterns during the mid-Pliocene. Simulation results are also in agreement with observed ongoing vegetation shifts and expected future vegetation patterns. Lamsal et al. (2017) find that specific tree types will be able to migrate towards higher elevations of the Himalayas. Furthermore, based on recent satellite observations, Anderson et al. (2020) find that vegetation is already expanding across the Himalayas. These results are confirmed by our simulations, and the agreement is larger the higher the prescribed level of carbon dioxide. We note that a decaying Amazon rain forest is ambiguous in future projections (Good et al., 2013), so robustness of that finding in our increased carbon dioxide simulations is unclear. On the other hand, our simulations suggest reduction of rainfall across Australia, both for modern and mid-Pliocene geography. There is a related reduction of forest cover on the Australian continent for lower concentrations of carbon dioxide. This finding, if confirmed by other models, may be of relevance in the context of large scale bush fires that have recently plagued Australian population and wildlife.

## 4.5   Model improvements in PlioMIP2 and beyond

The step from PlioMIP1 to PlioMIP2 has certainly marked a milestone in the modelling of mid-Pliocene climate. The PlioMIP2 brought additional sensitivity simulations, an updated mid-Pliocene geography, as well as an improved representation of the Earth system in our simulations via consideration of vegetation dynamics. Yet, ongoing model development and absence of model representations of further dynamic earth system components in PlioMIP2 are a fact. Some missing model components are of relevance for past and future warmer-than-present climate states. Consequently, not all dynamical aspects of the mid-Pliocene climate, geared towards understanding the future, are covered by COSMOS in particular, and by PlioMIP2 in general. This is a potential target of further development of the PlioMIP model and simulation ensemble.

Although adaptation of soil distribution in response to simulated climate and related vegetation changes would further improve consistency of climate and vegetation in the model, our PlioMIP2 model setup lacks this capability. Based on COSMOS, a modelling scheme has been developed that facilitates adjustment of soil characteristics to simulated climate conditions after model spin-up (Stärz et al., 2016). Yet, employing this capability consistently across the extensive PlioMIP2 simulation ensemble is beyond our contribution to PlioMIP2 due to a large additional spin-up time of the soil system, and subsequently of the climate system. Instead, we implemented mid-Pliocene soils into the model based on the reconstruction by Pound et al. (2014). In the future, there could be dedicated sensitivity studies in PlioMIP that consider the impact of soil evolution in the model on the derived climate state. This may be fruitful towards further exploration of persisting discord between reconstructed and modelled mid-Pliocene (land) surface temperatures (Salzmann et al., 2013), as soil evolution has been shown to have a positive feedback on climate (Stärz et al., 2016).

Dynamic treatment of the land cryosphere is, in principal, possible in COSMOS (Barbi et al., 2014) via a coupled ice sheet model but would prevent the implementation of the PRISM4 reconstruction of ice sheet extent and height (Dowsett et al., 2016)

as per modelling protocol (Haywood et al., 2016). Hence, ice sheets are a fixed component of prescribed paleogeography in COSMOS PlioMIP2 simulations. Their climatic effect is parameterised in the model via elevation and albedo of the Earth's surface. Dolan et al. (2018) highlight the uncertainty in the simulation of ice sheet characteristics of the mid-Pliocene. Sources of uncertainty include the choice of climate or ice sheet model. Keeping in mind problems with the interpretation of mid-Pliocene ice sheet modelling results as summarised by Dolan et al. (2018), future phases of PlioMIP could propose simulations that test the climatic effect of different plausible ice sheet geometries, just as proposed for the Last Interglacial in PMIP4 (Otto-Bliesner et al., 2017a). Such simulations could then be used to propagate uncertainties in simulated ice sheet characteristics into large scale mid-Pliocene climate patterns. This would enable an estimate of the climatic impact of uncertainty in ice sheet reconstructions of the mid-Pliocene on model-data comparison for relevant climate variables, e.g. SST or SAT. Consequently, the approach may shed further light on persisting discord between simulations and reconstructions of mid-Pliocene climate in a similar manner as presented by Pfeiffer and Lohmann (2016) for the Last Interglacial.

At tectonic time scales, and in particular under the impact of elevated atmospheric carbon dioxide and the related ocean acidification in the context of Pliocene4Future, simulation of ocean chemistry becomes an interesting component of modelling the Earth system. Since the Mid-Miocene Climatic Optimum (14–17 Ma BP) there has been a general trend towards reduced levels of carbon dioxide and, as a result, increased values of ocean pH and alkalinity (Sosdian et al., 2018). During this time the saturation state of aragonite has always been above a value of 2 (Sosdian et al., 2018). A saturation state below a value of 1 leads to dissolution of aragonite and poses severe problems for calcifying marine organisms, with potential threats to the food chain and higher organisms. Sosdian et al. (2018) have shown that under high emission scenarios, beyond the year 2100 CE at carbon dioxide concentrations of 1,000 ppmv in the atmosphere, the saturation state of aragonite will generally fall below 1. Even the highest level of carbon dioxide employed by us with the mid-Pliocene setup (560 ppmv) is well below that critical threshold of atmospheric carbon dioxide. Yet, it has been observed that the aragonite saturation horizon, below which seawater is undersaturated, is already rising in the major ocean basins (Feely et al., 2004), and upwelling of acidified water on the western North-American continental shelf has been shown (Feely et al., 2008). These processes lead to shrinking habitats for calcifiers. With marine carbon cycle and ecosystem models included in the mid-Pliocene modelling framework, PlioMIP may, in the future, also become a test bed to analyse stress levels of calcifying marine organisms on the currently projected path to severe ocean acidification. Their reaction to rising levels of pH, and the effects of resulting shifts in calcifier populations on the food chain, may be studied. We propose to consider such research during later phases of PlioMIP, with the knowledge in mind that the necessary models (COSMOS, for example) are already available. Employing marine carbon cycle models for a large number of simulations may not be possible in the near future due to the substantial computational load caused by spinning up the ocean carbon cycle to simulate paleoclimatic conditions. Yet, such studies may be done at a smaller scale in a dedicated model setup.

## 5 Conclusions

With this publication we have documented the modelling methodology employed by us towards generating the COSMOS PlioMIP2 simulation ensemble. Our contribution to PlioMIP2 with COSMOS contains 11 genuine PlioMIP2 simulations, three simulations that are related to CMIP6, and three self-contributed simulations. Of the latter, two quantify the effect of further increased carbon dioxide for modern and mid-Pliocene geography. We have characterised the COSMOS mid-Pliocene core simulation with a focus on global average quantities and large scale climate patterns. Furthermore, we have studied the impact of various contributors to mid-Pliocene warmth and to large scale characteristics of the mid-Pliocene climate in our model. These results have been discussed, also with respect to the question of how far results from PlioMIP2 COSMOS mid-Pliocene climate simulations may be transferable to future warmer-than-present climates.

We have found that our mid-Pliocene core simulation confirms many results from the PlioMIP1 model ensemble. The simulated mid-Pliocene is both warmer (+3.37 °C) and wetter (+0.17 mm d$^{-1}$) than the PI control climate. There is an increase in the AMOC of the equilibrium state, a characteristic observed in only some models of PlioMIP1. Our simulation confirms strong polar amplification, a reduced meridional temperature gradient, and a mid-Pliocene summer Arctic that is nearly free of sea ice – the simulated sea ice extent being sensitive to the assumed level of carbon dioxide. Since COSMOS has a comparably high climate sensitivity, it is not surprising that the prescribed carbon dioxide of 400 ppmv is the major driver of simulated mid-Pliocene warmth. About two-thirds of the temperature anomaly, mid-Pliocene core versus PI, are caused by carbon dioxide, followed by much smaller contributions from changes in topography and ice sheets.

Our contribution differs from that of other current PlioMIP2 model groups in that we have employed vegetation dynamics rather than a prescribed mid-Pliocene reconstruction (Haywood et al., 2020). Climate-vegetation dynamics are hence explicitly represented in model dynamics that lead to COSMOS PlioMIP2 simulations. Furthermore, beyond the official PlioMIP2 protocol, we have further explored details of the PRISM4 reconstruction of mid-Pliocene geography. Specifically, we have quantified the impact of Northern Hemisphere gateways on mid-Pliocene climate. By means of the dynamic vegetation module we were able to confirm inference from the geologic record. Our simulations show that during the mid-Pliocene, high latitudes of the Northern Hemisphere were covered by forests that are not present in the simulated PI control state. By means of our gateway sensitivity simulation we have found that in the PRISM4 set of model boundary conditions a combination of closed Bering Strait, Canadian Arctic Archipelago, and Hudson Bay indeed leads to increased SST in parts of the Atlantic-Arctic Ocean basin. The reconstructed gateway states hence contribute to a warmer mid-Pliocene Atlantic Ocean sector, although their global impact is small.

We have also highlighted that some deviations between modelled and reconstructed mid-Pliocene climate, that persisted in PlioMIP1, are still present in our PlioMIP2 simulations. One example is the mismatch between modelled and reconstructed mid-Pliocene SST. The discord is still present although we have considered, in our simulations, additional drivers of mid-Pliocene warmth in the North Atlantic and Arctic Ocean basins. The adjustment of Northern Hemisphere gateways and of carbon dioxide mitigates the mismatch to some extent, but not completely. One possible future solution to this problem could be the study of the impact of uncertainty in the state of mid-Pliocene ice sheets. These are currently prescribed based on

reconstructions that are known to be uncertain. Studying the SST response to various equally plausible ice sheet reconstructions may shed light on the potential contribution of ice sheet uncertainty to high latitude SST mismatch in the model.

*Code and data availability.* Output of COSMOS PlioMIP2 simulations is available from a data repository at the University of Leeds. Specific data requests, and data requests beyond simulations of the official PlioMIP2 protocol, should be addressed to Christian Stepanek (Christian.Stepanek@awi.de). PlioMIP2 boundary conditions based on PRISM4 can be retrieved from the official PlioMIP2 web page of the United States Geological Survey: http://geology.er.usgs.gov/egpsc/prism/7.2_pliomip2_data.html (last accessed: 31$^{st}$ of January, 2020). Requests for COSMOS source code should be directed to the Max Planck Institute for Meteorology, Bundesstrasse 53, 20146 Hamburg, Germany.

*Author contributions.* CS and GL designed the study. CS developed the software framework for preparation of mid-Pliocene (as well as derived) geographic boundary conditions and for analysis of COSMOS model output. CS prepared model setups and conducted model spin-up and analysis, supported by ES. CS compiled the manuscript and created graphical illustrations and tables, considering comments by ES, GK, and GL. For correspondence and requests contact CS.

*Competing interests.* The authors declare that they have no conflict of interest.

*Acknowledgements.* COSMOS PlioMIP2 simulations have been conducted at the Computing and Data Centre of the Alfred-Wegener-Institut – Helmholtz-Zentrum für Polar und Meeresforschung on a NEC SX-ACE high performance vector computer. Analysis of model outputs has been performed based on free and open source software (Python, Climate Data Operators). The authors would like to thank the PlioMIP2 steering committee for devising, planning and organising PlioMIP activities over more than a decade. Harry Dowsett and colleagues from the United States Geological Survey are acknowledged for improving our proxy-based knowledge on climate and geography of the mid-Pliocene. We thank all scientists that contributed reconstructions of boundary conditions to PlioMIP2's gridded data sets of PRISM4 paleogeography. Stefanie Meier is acknowledged for analysing simulated vegetation distributions in the framework of a bachelor thesis project. The Max Planck Institute for Meteorology (MPI-Met) is acknowledged for providing the COSMOS model code and associated modern and Pre-Industrial boundary and initial conditions. We thank Tobias Stacke of the MPI-Met for adjustment of ECHAM5's river routing scheme to PlioMIP2 mid-Pliocene and derived geography. The authors would like to express their gratitude to Wing-Le Chan for kindly editing this publication, and in particular for his efforts in providing valuable suggestions towards streamlining the language of our manuscript. Two anonymous referees are acknowledged for their input that helped the authors to substantially improve the manuscript. CS thanks Erin McClymont for clarifying discussions regarding the interpretation of reconstructions of mid-Pliocene sea surface temperature. GL acknowledges funding via the Alfred Wegener Institute's research programme PACES2. CS and GK acknowledge funding by the Helmholtz Climate Initiative REKLIM and the Alfred Wegener Institute's research programme PACES2.

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

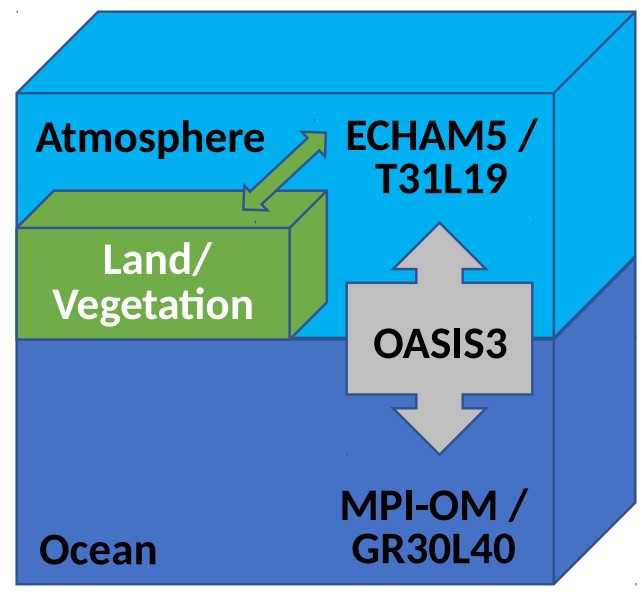

**Figure 1.** Schematic of the Community Earth System Models (COSMOS) toolbox that is composed of two model components and a coupler: atmosphere/land/vegetation (ECHAM5/JSBACH) and ocean/sea ice (MPIOM) are coupled by OASIS3. See text for details.

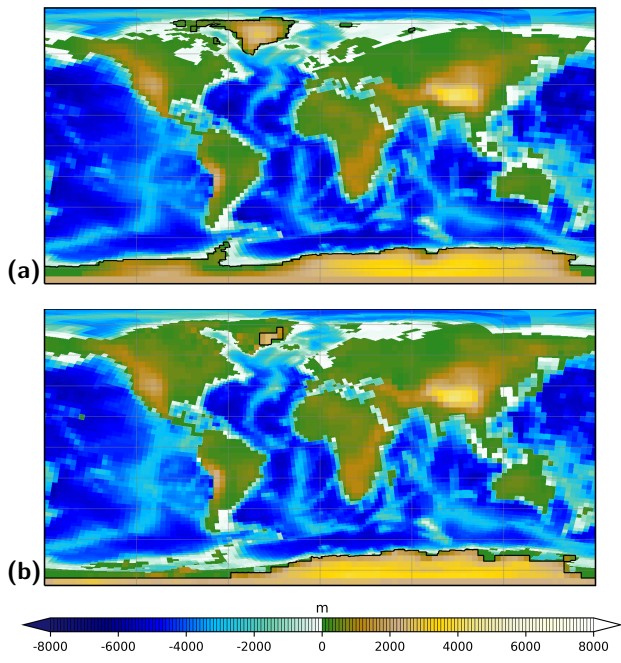

**Figure 2.** Model setup of COSMOS PlioMIP2 core simulations for (a) Pre-Industrial and (b) mid-Pliocene geography. Shown are bathymetry (dark blue to white shading), land topography (green to white shading), and the extent of ice sheets (black contours). From this setup sensitivity simulations of the PlioMIP2 simulation ensemble are derived.

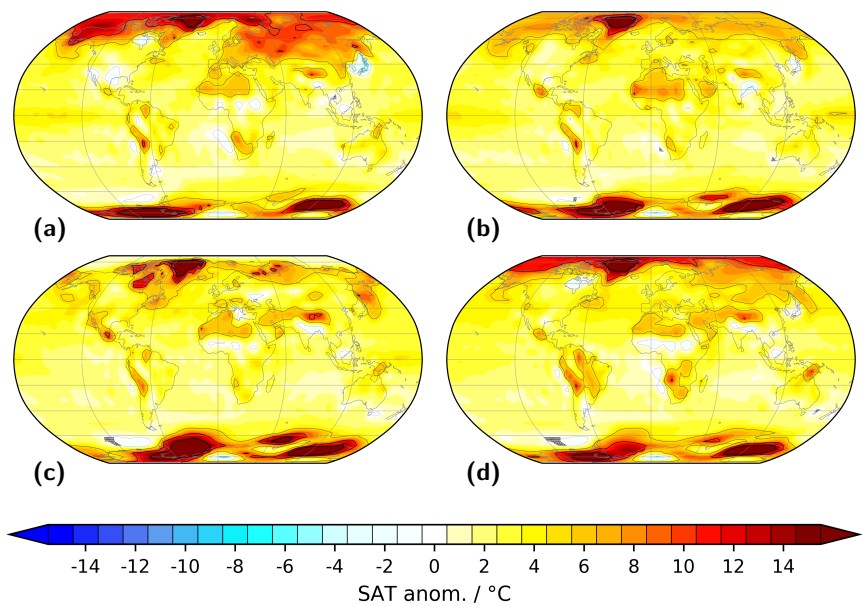

**Figure 3.** Seasonal mean surface air temperature (SAT) for the PlioMIP2 mid-Pliocene core simulation Eoi400. Shown are anomalies with respect to Pre-Industrial (simulation E280) for: a) boreal winter (DJF), b) boreal spring (MAM), c) boreal summer (JJA), and d) boreal autumn (SON). Contours illustrate isotherms of -5 °C (dashed, thin), 0 °C (dotted, thin), 5 °C (solid, thin), 10 °C (solid), and 15 °C (solid, thick). In hatched regions the anomaly is insignificant at 95% confidence interval based on a t-test. We define SAT as the temperature at a height of two metres above the surface.

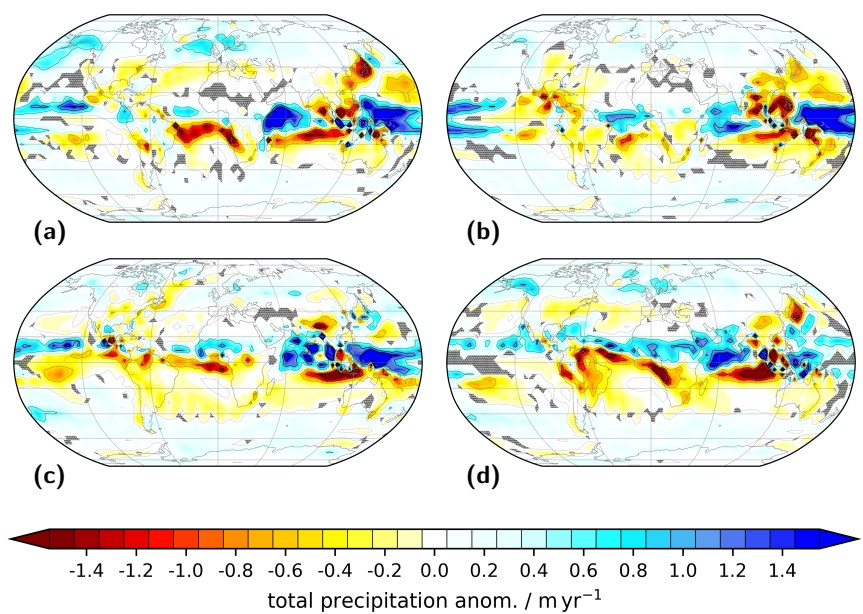

**Figure 4.** Seasonal mean total precipitation for the PlioMIP2 mid-Pliocene core simulation Eoi400. Shown are anomalies with respect to Pre-Industrial (simulation E280) for: a) boreal winter (DJF), b) boreal spring (MAM), c) boreal summer (JJA), and d) boreal autumn (SON). Contours illustrate isolines of -1.5 m yr$^{-1}$ (dashed), -1.0 m yr$^{-1}$ (dashed, thin), 0.0 m yr$^{-1}$ (dotted, thin), 0.5 m yr$^{-1}$ (solid, thin), 1.0 m yr$^{-1}$ (solid), and 1.5 m yr$^{-1}$ (solid, thick). In hatched regions the anomaly is insignificant at 95% confidence interval based on a t-test. Total precipitation integrates contributions from large scale and convective precipitation in liquid and solid phase.

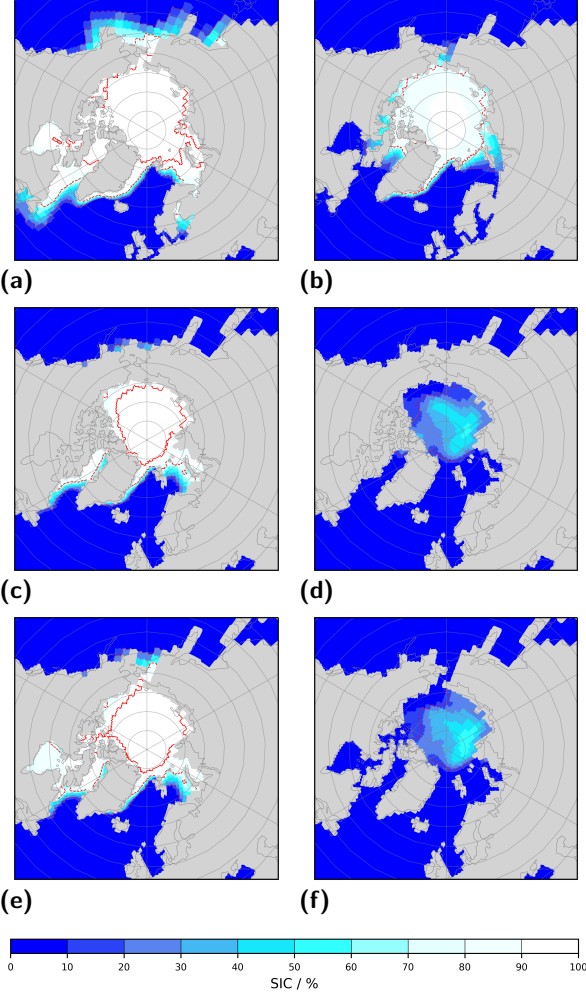

**Figure 5.** Sea ice coverage (SIC) in the Northern Hemisphere for boreal spring (MAM, left) and boreal autumn (SON, right). Shown are: a), b) E280 (Pre-Industrial); c), d) Eoi400 (PlioMIP2 mid-Pliocene core simulation); e), f) Eoi400_GW (as Eoi400, but with modern Bering Strait, Hudson Bay, and Canadian Arctic Archipelago). Grey shading illustrates the land sea mask. Red contours illustrate SIC-isolines of 15% (dotted), 75% (dashed), and 95% (solid).

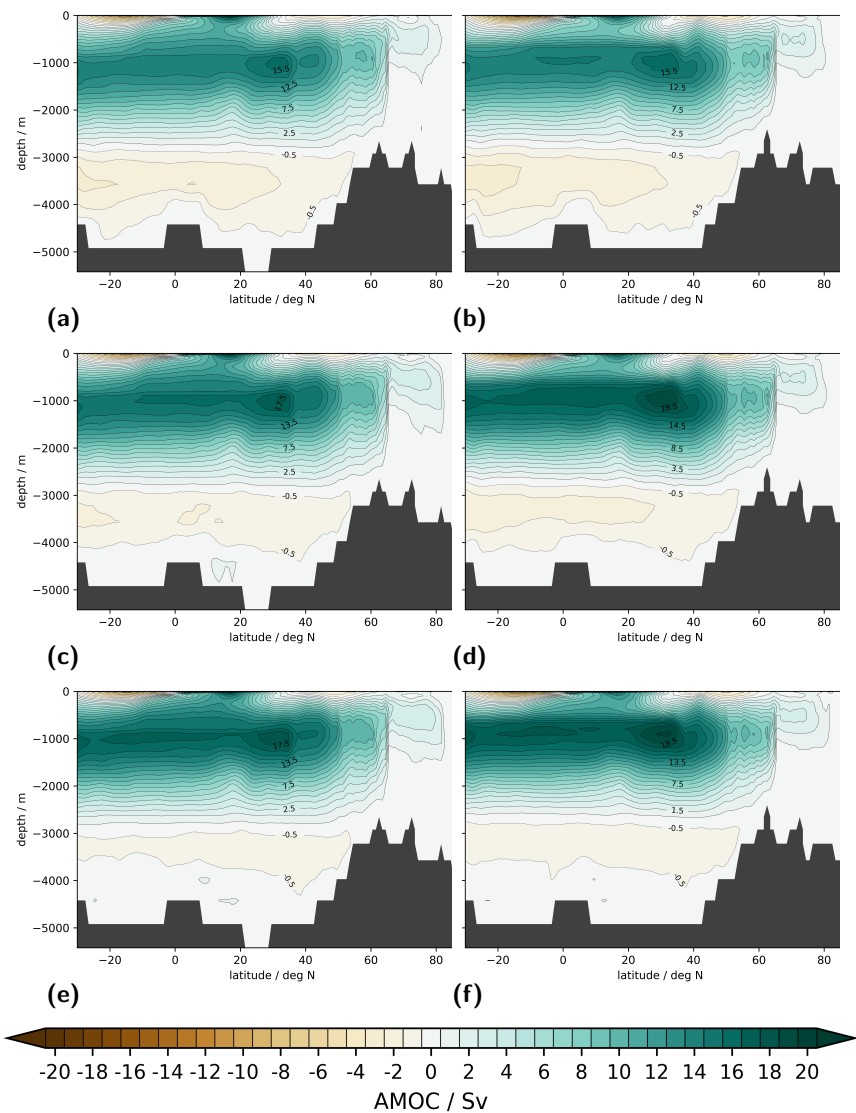

**Figure 6.** Atlantic Ocean Meridional Overturning Circulation (AMOC) in units of Sv (1 Sv $\equiv 10^6$ m$^3$ s$^{-1}$), visualised via the basin-wide zonal- and time-integrated stream function. Shown are: a) E280, b) Eoi280, c) E400, d) Eoi400, e) E560, and f) Eoi560. Positive AMOC values illustrate clockwise circulation as from the viewpoint of Europe and Africa. Dark grey shading illustrates bottom topography, zonally averaged across the basin.

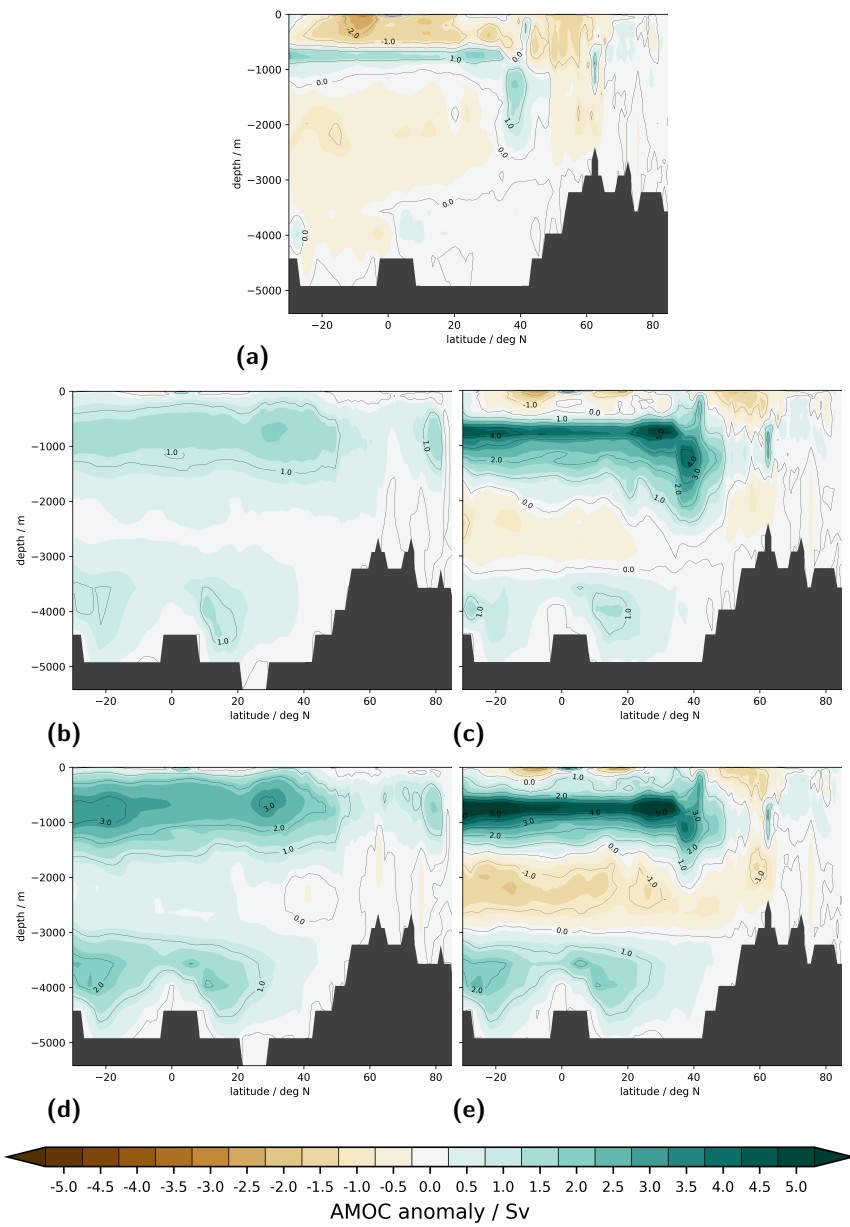

**Figure 7.** Atlantic Ocean Meridional Overturning Circulation (AMOC) anomaly with respect to Pre-Industrial (E280) in units of Sv ($1\,\mathrm{Sv} \equiv 10^6\,\mathrm{m}^3\,\mathrm{s}^{-1}$), visualised via the basin-wide zonal- and time-integrated stream function. Shown are: a) Eoi280, b) E400, c) Eoi400, d) E560, and e) Eoi560. Positive AMOC anomalies illustrate a change towards clockwise circulation as from the viewpoint of Europe and Africa. Dark grey shading illustrates bottom topography, zonally averaged across the basin.

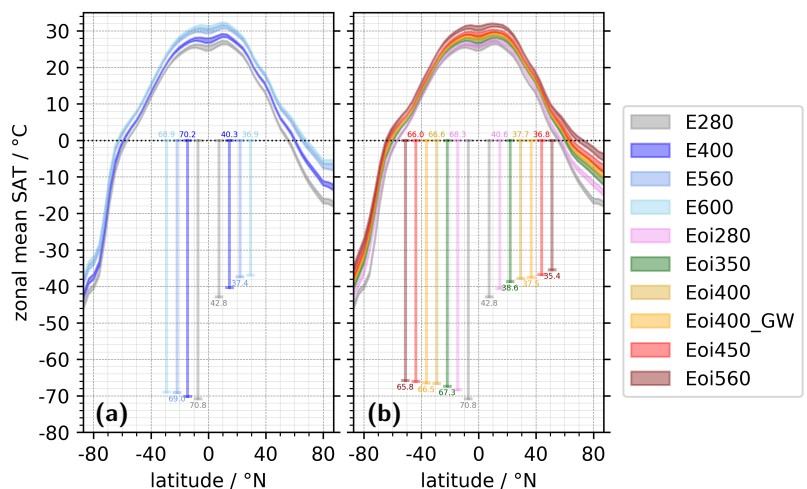

**Figure 8.** Meridional range of zonal mean surface air temperature (SAT), averaged over both ocean and land, for various levels of atmospheric carbon dioxide. Shading illustrates time-variability over the analysis period. Shown are simulations for: a) modern geography, b) mid-Pliocene geography. Vertical bars provide a visualisation of the meridional gradient of SAT both for Southern Hemisphere (left of 0°N) and Northern Hemisphere (right of 0°N). The computation is based on the lowest and highest latitude of each hemisphere, which does not necessarily provide the largest temperature range available within a hemisphere. We define SAT as the temperature at a height of two metres above the surface.

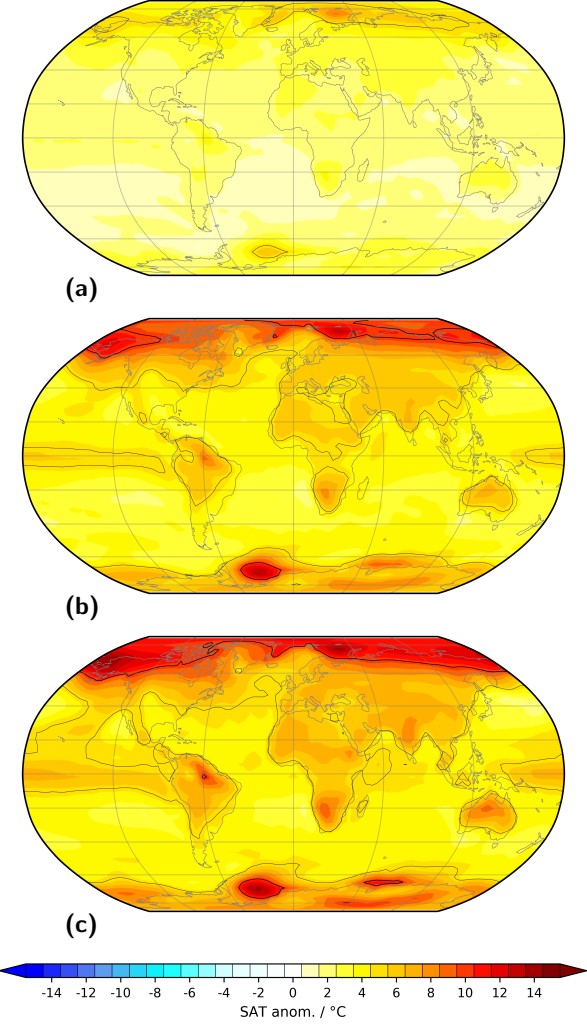

**Figure 9.** Sensitivity of annual mean surface air temperature (SAT) to variations in volume mixing ratio of carbon dioxide ($CO_2$) for modern geography. Shown are anomalies with respect to Pre-Industrial (E280) for simulations: a) E400, b) E560, and c) E600. In hatched regions the anomaly is insignificant at 95% confidence interval based on a t-test. Contours illustrate isotherms of $5\,°C$ (solid, thin) and $10\,°C$ (solid). We define SAT as the temperature at a height of two metres above the surface.

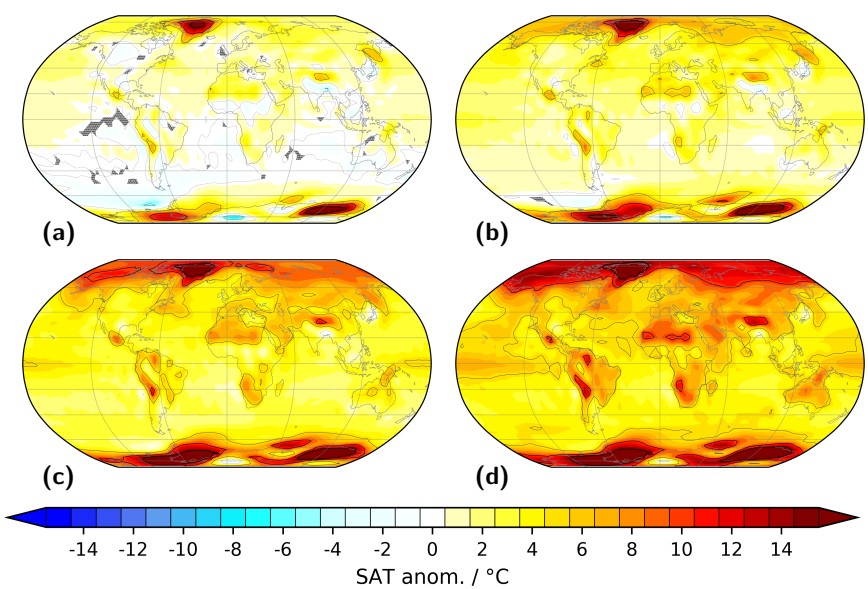

**Figure 10.** Sensitivity of annual mean surface air temperature (SAT) to variations in volume mixing ratio of carbon dioxide ($CO_2$) for mid-Pliocene paleogeography. Shown are anomalies with respect to Pre-Industrial (E280) for simulations: a) Eoi280, b) Eoi350, c) Eoi450, and d) Eoi560. In hatched regions the anomaly is insignificant at 95% confidence interval based on a t-test. Contours illustrate isotherms of -5 °C (dashed, thin), 0 °C (dotted, thin), 5 °C (solid, thin), 10 °C (solid), and 15 °C (solid, thick). We define SAT as the temperature at a height of two metres above the surface.

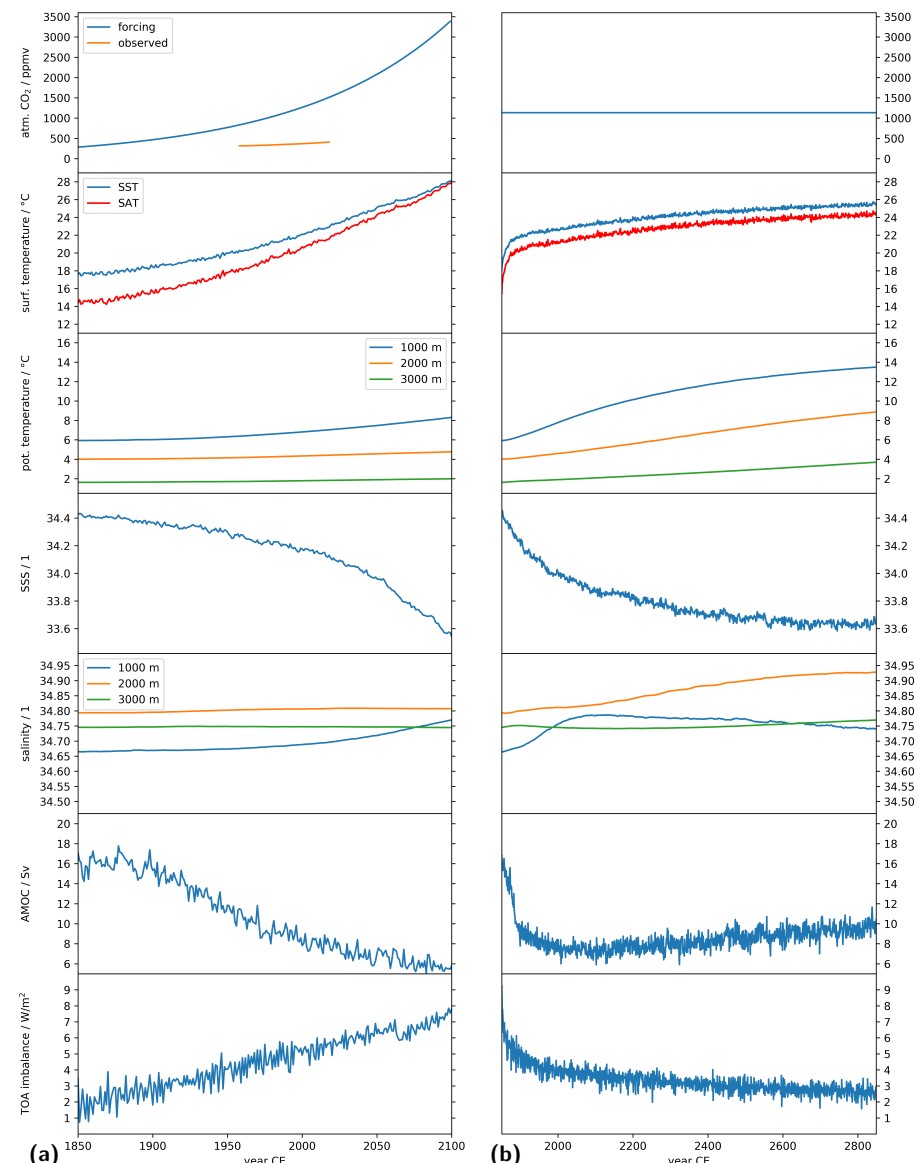

**Figure 11.** Time evolution of selected climate indices of CMIP6 simulations as a result of changes in carbon dioxide after initialisation from a CMIP6 PI control state (E280C): a) 1pctCO2 and b) abrupt4xCO2 (Eyring et al., 2016). Carbon dioxide forcing shown in uppermost graphs; observations by Keeling et al. (2001) for reference in a. Surface air temperature (SAT) here refers to the surface skin temperature; sea surface temperature (SST) and sea surface salinity (SSS) represent the uppermost ocean layer (0–12 m); Atlantic Meridional Overturning Circulation (AMOC) is the maximum streamfunction in the Atlantic Ocean north of 20°N and between 500 m and 1,500 m; top of the atmosphere (TOA) radiative imbalance computed from incoming and outgoing short- and long-wave radiation at TOA.

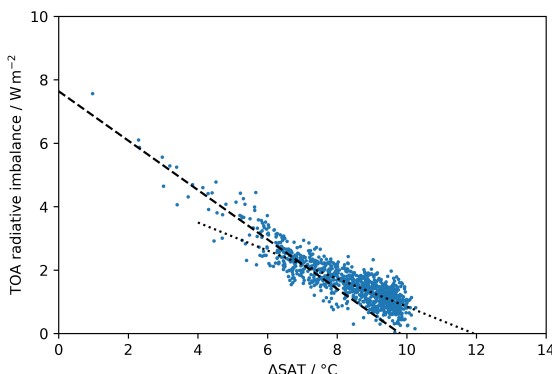

**Figure 12.** Regression of top of the atmosphere (TOA) radiative imbalance versus change of global average surface skin temperature (SAT) in simulation abrupt4xCO2 based on annual means. Reference is the average over the last 100 model years of the CMIP6 PI state E280C before branching off simulation abrupt4xCO2. Shown are annual mean values (dots) and two regressions: regression of the first 10% of the simulation illustrated via the dashed line, regression of the remainder of the data set illustrated via the dotted line. Intercept of the dotted line with the SAT anomaly axis estimates the equilibrium response of the climate system to a quadrupling of carbon dioxide. Equilibrium Climate Sensitivity is derived by halving this value.

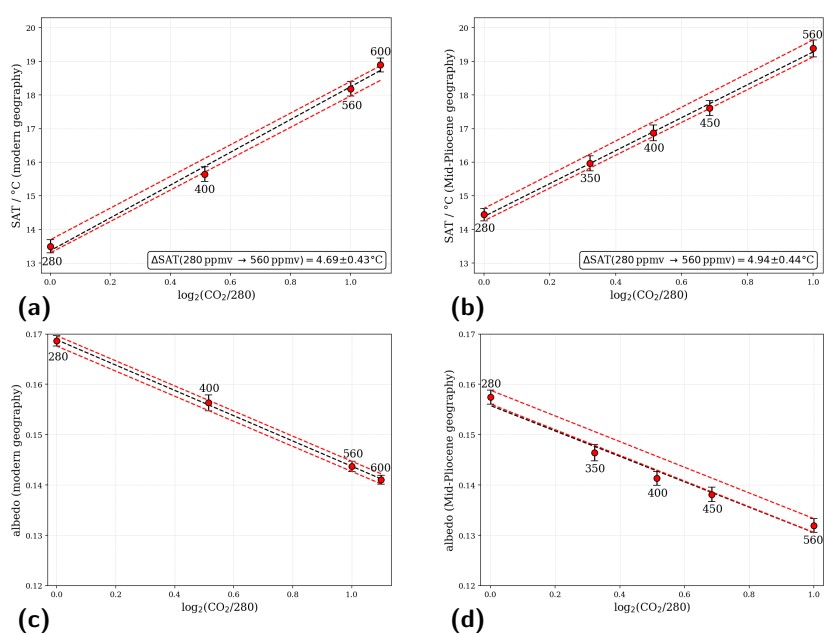

**Figure 13.** Sensitivity of modelled global mean surface air temperature (SAT) (a,b) and surface albedo (c,d) to variations in volume mixing ratio of carbon dioxide ($CO_2$) in units of parts per million by volume (ppmv). Shown are model configurations of a, c) modern geography, and b, d) mid-Pliocene paleogeography. Data points are annotated with corresponding $CO_2$ concentrations. Variability of SAT and surface albedo over the analysis period is illustrated by vertical error bars. Black dashed regression lines represent a best linear fit for all simulations considered, red dashed regression lines highlight a corridor of perfect linearity considering simulations with 280 ppmv and 560 ppmv as a guidance. For mid-Pliocene surface albedo, the strong nonlinearity with regard to $CO_2$ leads to the best linear fit being outside this corridor.

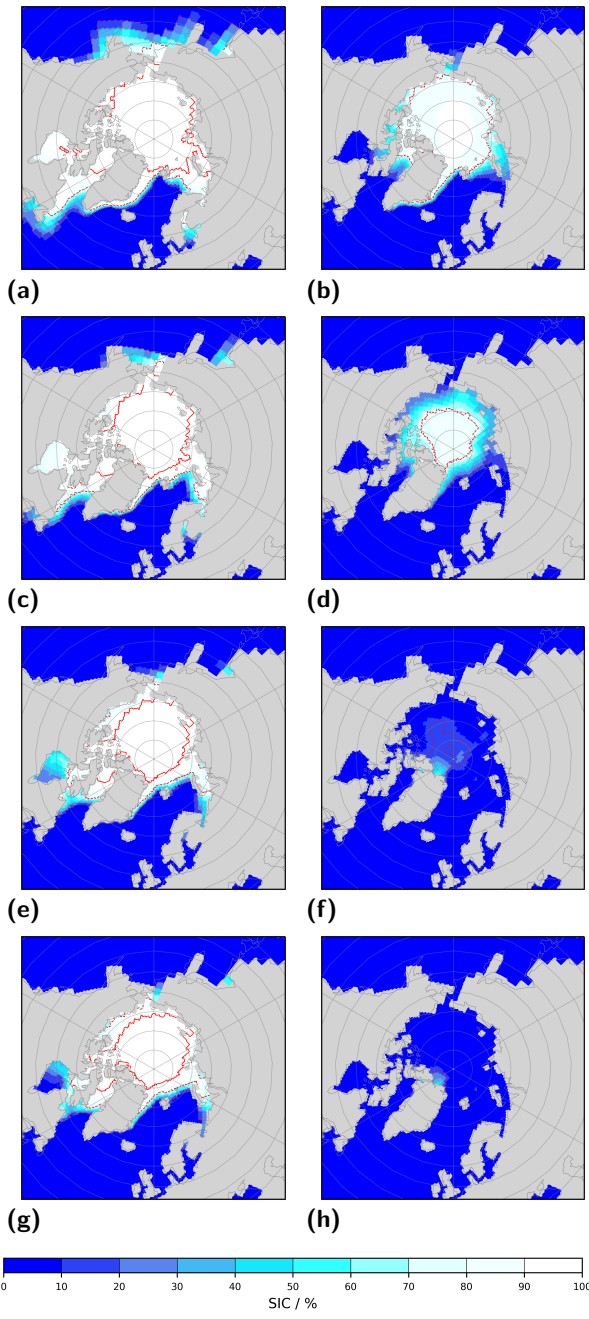

**Figure 14.** Sea ice coverage (SIC) in the Northern Hemisphere for boreal spring (MAM, left) and boreal autumn (SON, right) depending on the volume mixing ratio of carbon dioxide for modern geography. Shown are: a), b) E280; c), d) E400; e), f) E560; g), h) E600. Grey shading illustrates the land sea mask. Red contours illustrate SIC-isolines of 15% (dotted), 75% (dashed), and 95% (solid).

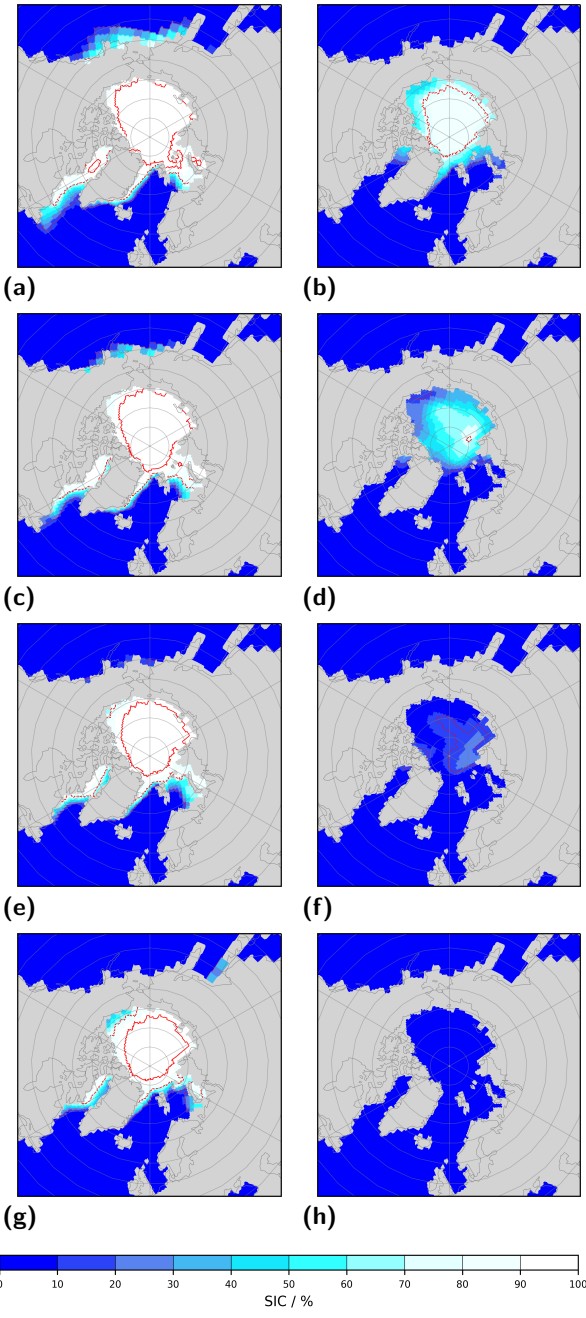

**Figure 15.** Sea ice coverage (SIC) in the Northern Hemisphere for boreal spring (MAM, left) and boreal autumn (SON, right) depending on the volume mixing ratio of carbon dioxide for mid-Pliocene geography. Shown are: a), b) Eoi280; c), d) Eoi350; e), f) Eoi450; g), h) Eoi560. Grey shading illustrates the land sea mask. Red contours illustrate SIC-isolines of 15% (dotted), 75% (dashed), and 95% (solid).

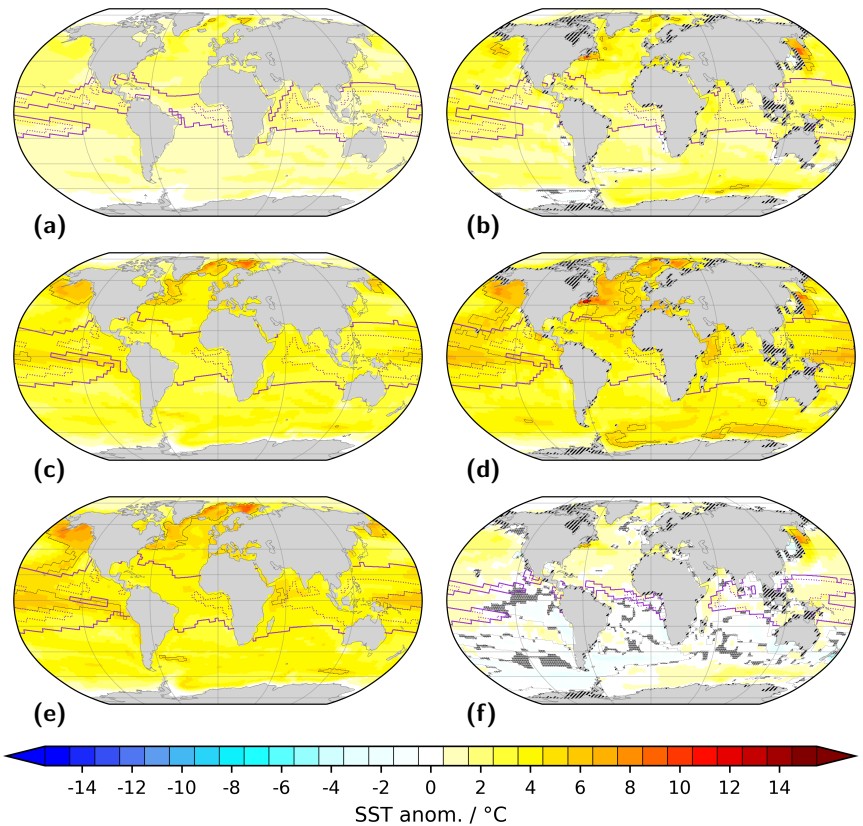

**Figure 16.** Annual mean sea surface temperature (SST) anomaly with respect to Pre-Industrial (PI, E280). We show the impact of comparable radiative forcing in setups with both modern (left) and mid-Pliocene (right) geography: a) E400, b) Eoi400, c) E560, d) Eoi560. We also show the impact of further increased concentration of $CO_2$, e) E600, and of Pre-Industrial $CO_2$ in a model setup with mid-Pliocene geography, f) Eoi280. Light grey shading illustrates land sea masks. In hatched ocean regions the anomaly is insignificant at 95% confidence interval based on a t-test. Hatches over land indicate differences in land-sea configuration between a specific simulation and the PI reference. Black contours illustrate isotherms of 0 °C (dotted, thin), 5 °C (solid, thin), and 10 °C (solid, thick). Extent of the equatorial warm pool, defined as regions above 28 °C, is indicated by violet contours for PI (dotted) and for a specific simulation (solid). We define SST as the temperature of the uppermost ocean layer, centred around a depth of six metres.

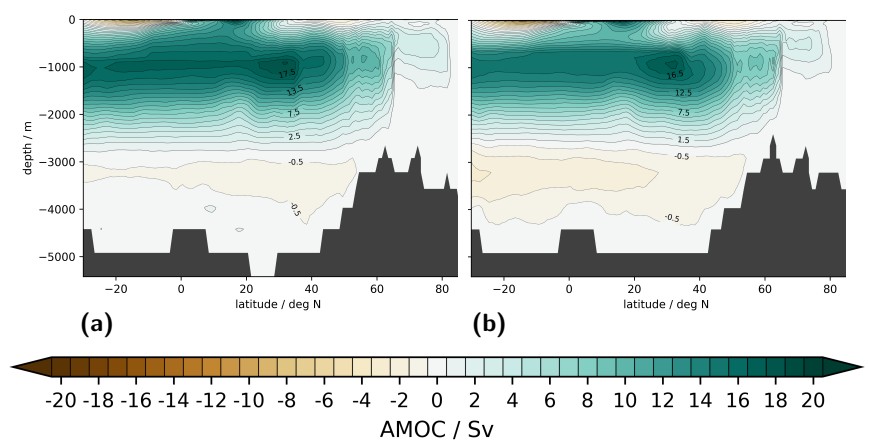

**Figure 17.** Atlantic Ocean Meridional Overturning Circulation (AMOC) in units of Sv ($1\,\mathrm{Sv} \equiv 10^6\,\mathrm{m}^3\,\mathrm{s}^{-1}$), visualised via the basin-wide zonal- and time-integrated stream function. Shown are: a) E600 and b) Eoi400_GW. Positive AMOC values illustrate clockwise circulation from the viewpoint of Europe and Africa. Dark grey shading illustrates bottom topography, zonally averaged across the basin.

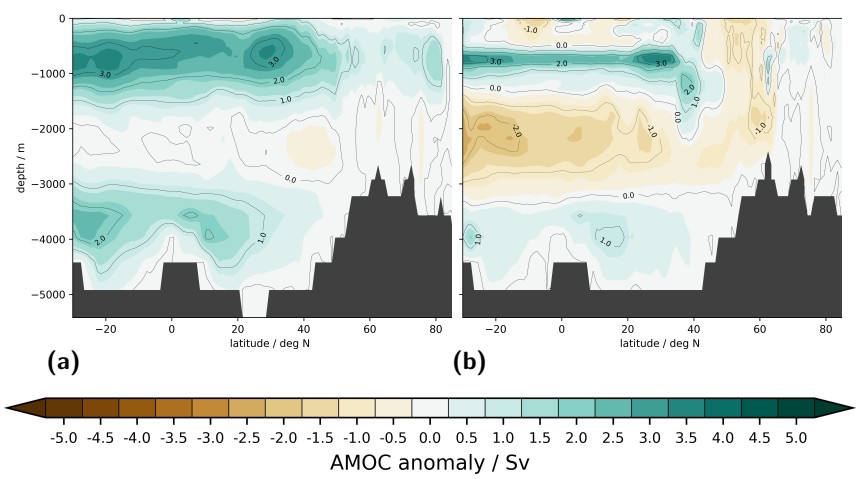

**Figure 18.** Atlantic Ocean Meridional Overturning Circulation (AMOC) anomaly with respect to Pre-Industrial (E280) in units of Sv ($1\,\mathrm{Sv} \equiv 10^6\,\mathrm{m^3\,s^{-1}}$), visualised via the basin-wide zonal- and time-integrated stream function. Shown are: a) E600 and b) Eoi400_GW. Positive AMOC anomalies illustrate a change towards clockwise circulation from the viewpoint of Europe and Africa. Dark grey shading illustrates bottom topography, zonally averaged across the basin.

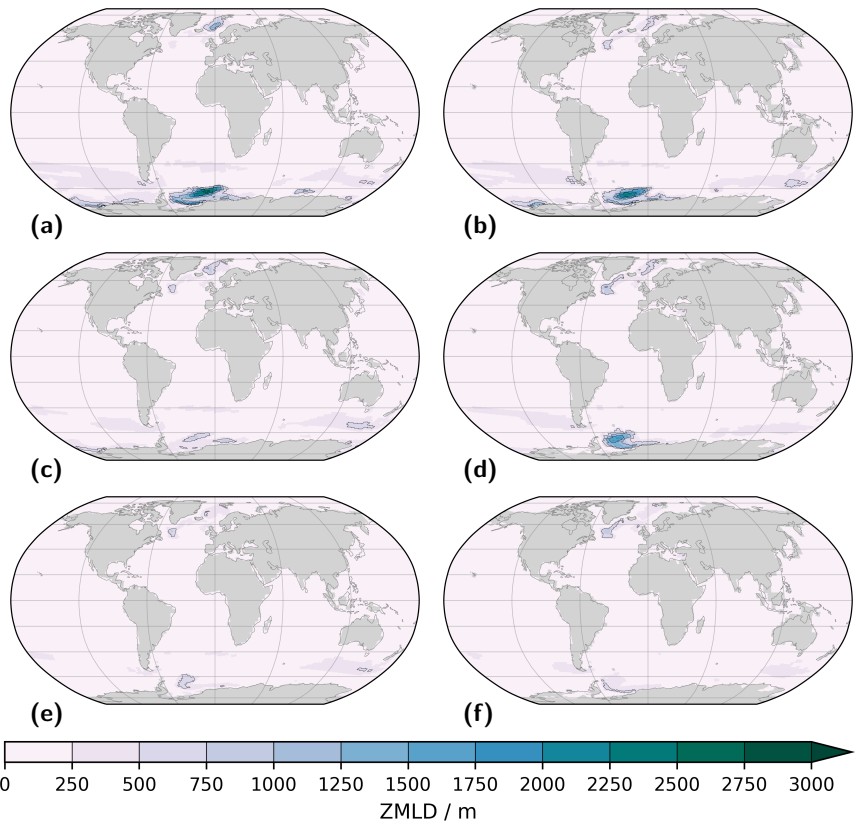

**Figure 19.** Annual mean depth of the mixed layer in the ocean (ZMLD, units of m). The ZMLD is defined by an increase of $0.125\,\mathrm{kg\,m^{-3}}$ in seawater density in comparison to the uppermost ocean layer. Shown are: a) E280, b) Eoi280, c) E400, d) Eoi400, e) E560, and f) Eoi560. Contours given in intervals of $500\,\mathrm{m}$, starting at $500\,\mathrm{m}$. Grey shading illustrates the land sea mask.

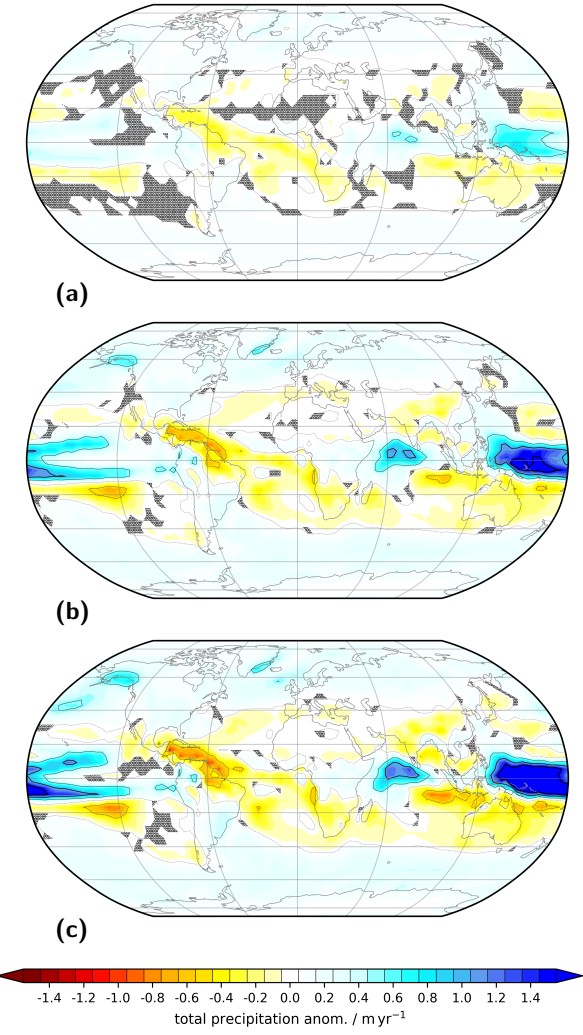

**Figure 20.** Sensitivity of annual mean total precipitation to variations in volume mixing ratio of carbon dioxide for modern geography. Shown are anomalies with respect to Pre-Industrial (E280) for simulations: a) E400, b) E560, c) E600. Contours illustrate isolines of -1.0 m yr[-1] (dashed, thin), 0.0 m yr[-1] (dotted, thin), 0.5 m yr[-1] (solid, thin), 1.0 m yr[-1] (solid), and 1.5 m yr[-1] (solid, thick). In hatched regions the anomaly is insignificant at 95% confidence interval based on a t-test. Total precipitation integrates contributions from large scale and convective precipitation in liquid and solid phase.

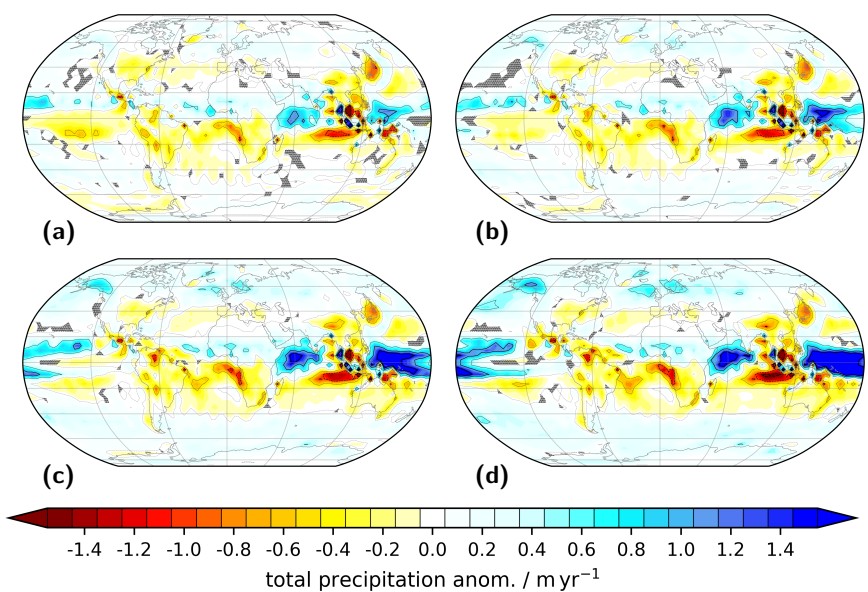

**Figure 21.** Sensitivity of annual mean total precipitation to variations in volume mixing ratio of carbon dioxide for mid-Pliocene paleogeography. Shown are anomalies with respect to Pre-Industrial (E280) for simulations: a) Eoi280, b) Eoi350, c) Eoi450, and d) Eoi560. Contours illustrate isolines of -1.5 m yr$^{-1}$ (dashed), -1.0 m yr$^{-1}$ (dashed, thin), 0.0 m yr$^{-1}$ (dotted, thin), 0.5 m yr$^{-1}$ (solid, thin), 1.0 m yr$^{-1}$ (solid), and 1.5 m yr$^{-1}$ (solid, thick). In hatched regions the anomaly is insignificant at 95% confidence interval based on a t-test. Total precipitation integrates contributions from large scale and convective precipitation in liquid and solid phase.

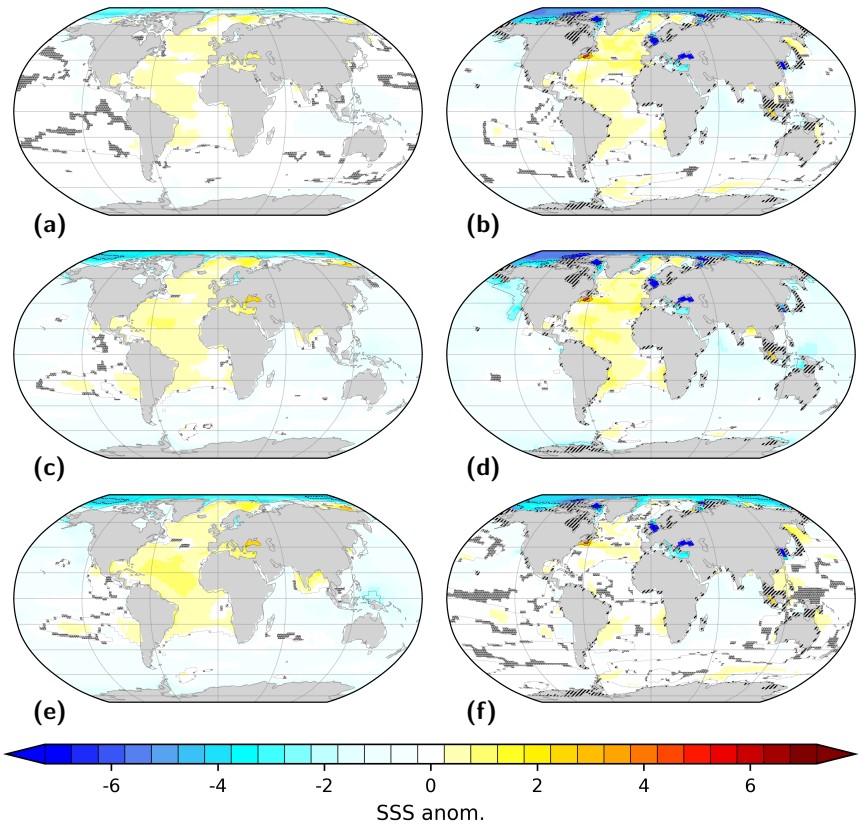

**Figure 22.** Annual mean anomaly of sea surface salinity (SSS) in PSU with respect to Pre-Industrial (PI, E280). We show the impact of comparable radiative forcing in setups with both modern (left) and mid-Pliocene (right) geography: a) E400, b) Eoi400, c) E560, d) Eoi560. We also show the impact of further increased concentration of $CO_2$, e) E600, and of PI $CO_2$ in a model setup with mid-Pliocene geography, f) Eoi280. Light grey shading illustrates land sea masks. In hatched ocean regions the anomaly is insignificant at 95% confidence interval based on a t-test. Hatches over land indicate differences in land-sea configuration between a specific simulation and the PI reference. Contours illustrate isolines of -6 (dashed, thick), -4 (dashed), -2 (dashed, thin), 0 (dotted, thin), 2 (solid, thin), and 4 (solid) of salinity. We define SSS as the salinity of the uppermost ocean layer, centred around a depth of six metres.

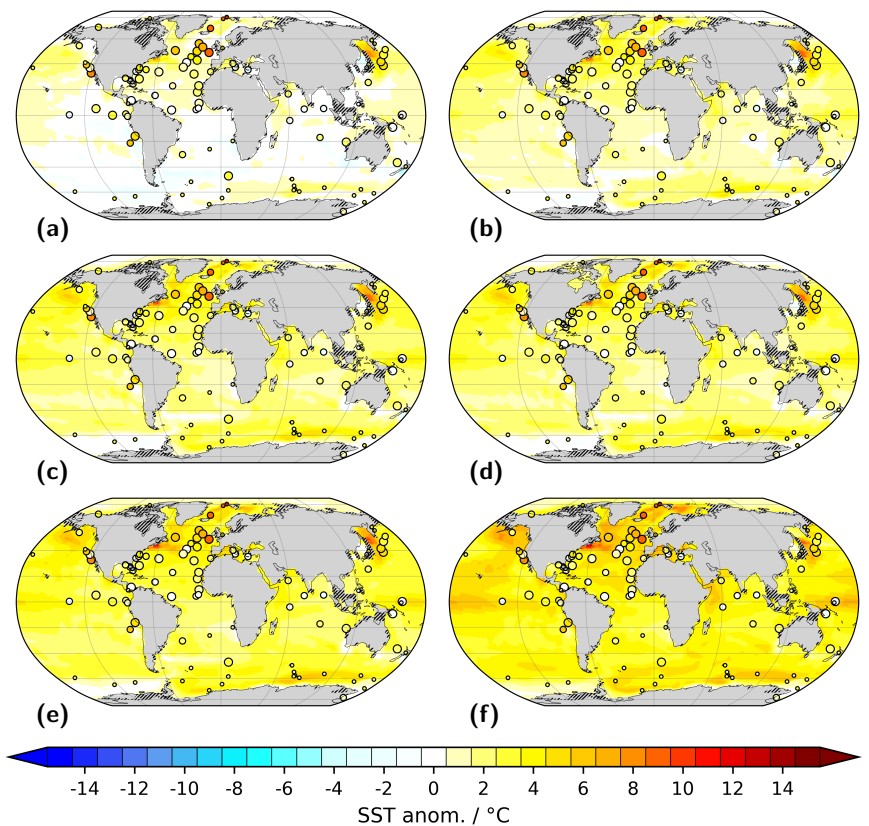

**Figure 23.** Comparison of modelled and reconstructed (PRISM3; Dowsett et al., 2009, 2013) sea surface temperature (SST) anomaly, with respect to Pre-Industrial (PI, E280), for simulations: a) Eoi280, b) Eoi350, c) Eoi400, d) Eoi400_GW, e) Eoi450, and f) Eoi560. Shading represents modelled annual mean SST anomaly. Circles illustrate the respective PRISM3 SST reconstruction, with the size of the circle indicating the associated degree of confidence (Dowsett et al., 2013) – higher confidence being represented by a larger circle. Grey shading illustrates the land sea mask. Hatches over land indicate differences in land-sea configuration between a specific simulation and PI reference. We define model SST as the temperature of the uppermost ocean layer, centred around a depth of six metres.

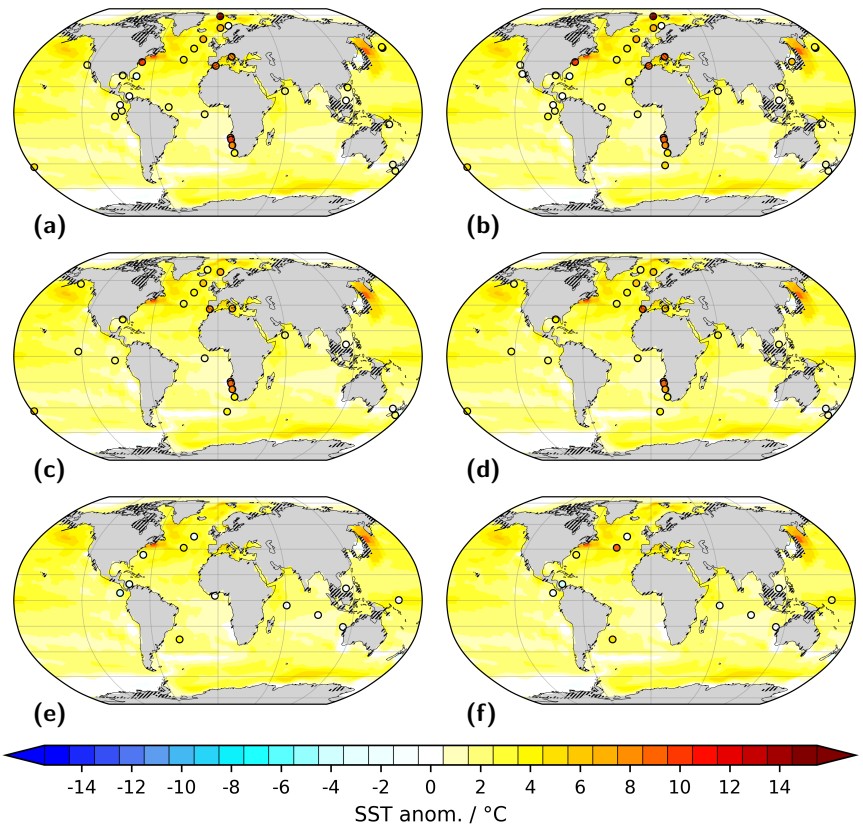

**Figure 24.** Comparison of modelled and reconstructed sea surface temperature (SST) anomaly of simulation Eoi400 with respect to Pre-Industrial (PI, E280). Shown are: a) alkenone ($U^K_{37'}$) with 10 ka window (Foley and Dowsett, 2019), b) $U^K_{37'}$ with 30 ka window (Foley and Dowsett, 2019), c) $U^K_{37'}$ PlioVAR synthesis (McClymont et al., 2020), d) $U^K_{37'}$ BAYSPLINE (McClymont et al., 2020), e) magnesium-to-calcium ratio (Mg/Ca) PlioVAR synthesis (McClymont et al., 2020), and f) Mg/Ca BAYMAG (McClymont et al., 2020) – see text for details with regard to reconstructions. Shading represents modelled annual mean SST anomaly. Circles illustrate the temperature anomaly of the respective mid-Pliocene SST reconstruction. Grey shading illustrates the land sea mask. Hatches over land indicate differences in land-sea configuration between Eoi400 and E280 reference. We define model SST as the temperature of the uppermost ocean layer, centred around a dept of six metres.

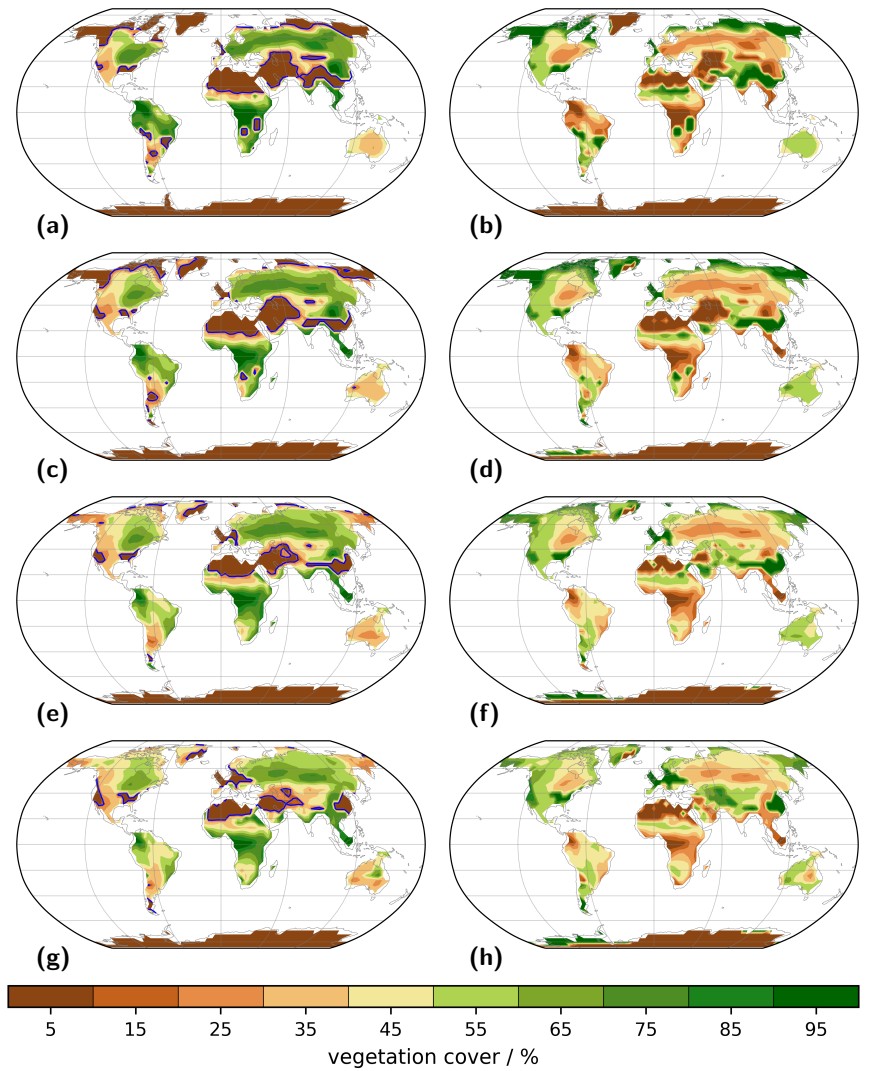

**Figure 25.** Simulated distribution of vegetation for forest- (left) and grass- (right) related plant functional types (PFTs) for modern and mid-Pliocene geography and various concentrations of carbon dioxide. Shown are E280 (a, b; for reference), Eoi280 (c, d), Eoi400 (e, f), and Eoi560 (g, h). Presence of forest and grass vegetation is quantified by the respective contribution (in %) to the total plant cover. Forest includes JSBACH tree PFTs 1–4 (tropical evergreen, tropical deciduous, extratropical evergreen, extratropical deciduous); grass includes JSBACH PFTs 5–8 (raingreen shrubs, deciduous shrubs, C3 grasses, and C4 grasses). In maps of forest vegetation the location of the tree line is illustrated by the blue contour. Here, we arbitrarily define the tree line to be equivalent to the 15% isoline of forest cover.

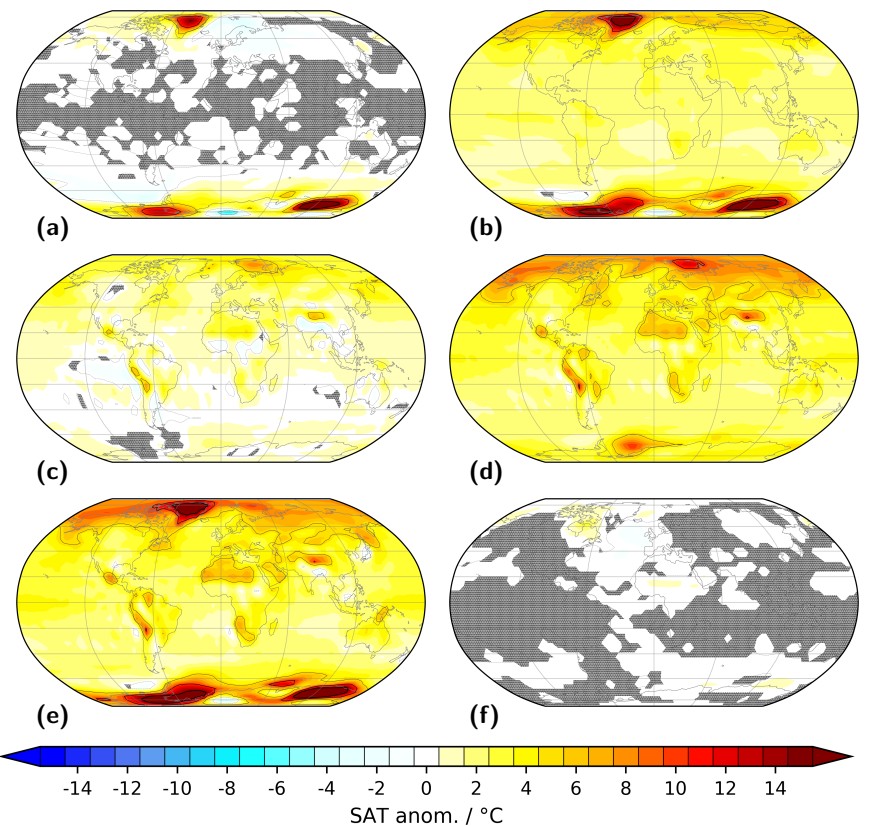

**Figure 26.** Sensitivity of annual mean surface air temperature (SAT) to variations in geography. Shown are anomalies with respect to Pre-Industrial (E280) for simulations: a) Ei280, b) Ei400, c) Eo280, d) Eo400, and e) Eoi400_GW. Subfigure f) illustrates the impact of gateway changes in simulation Eoi400_GW in comparison to simulation Eoi400. Contours illustrate isotherms of -5 °C (dashed, thin), 0 °C (dotted, thin), 5 °C (solid, thin), 10 °C (solid), and 15 °C (solid, thick). In hatched regions the anomaly is insignificant at 95% confidence interval based on a t-test. We define SAT as the temperature at a height of two metres above the surface.

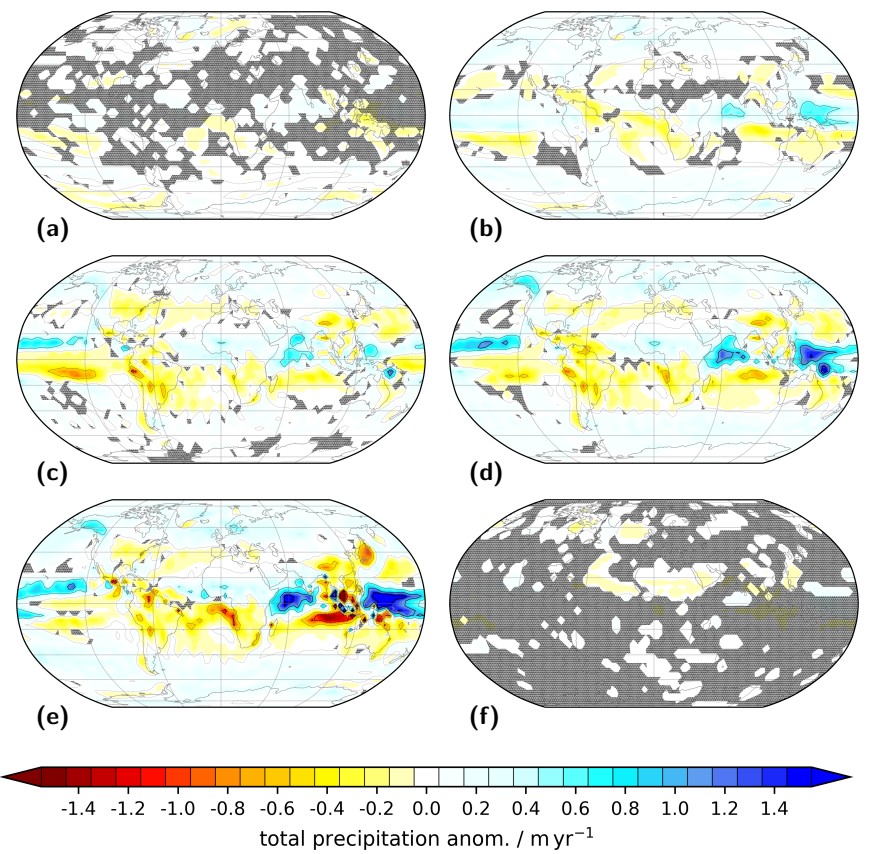

**Figure 27.** Sensitivity of annual mean total precipitation to variations in geography. Shown are anomalies with respect to Pre-Industrial (E280) for simulations: a) Ei280, b) Ei400, c) Eo280, d) Eo400, and e) Eoi400_GW. Subfigure f) illustrates the impact of gateway changes in simulation Eoi400_GW in comparison to simulation Eoi400. Contours illustrate isolines of -1.5 m yr⁻¹ (dashed), -1.0 m yr⁻¹ (dashed, thin), 0.0 m yr⁻¹ (dotted, thin), 0.5 m yr⁻¹ (solid, thin), 1.0 m yr⁻¹ (solid), and 1.5 m yr⁻¹ (solid, thick). In hatched regions the anomaly is insignificant at 95% confidence interval based on a t-test. Total precipitation integrates contributions from large scale and convective precipitation in liquid and solid phase.

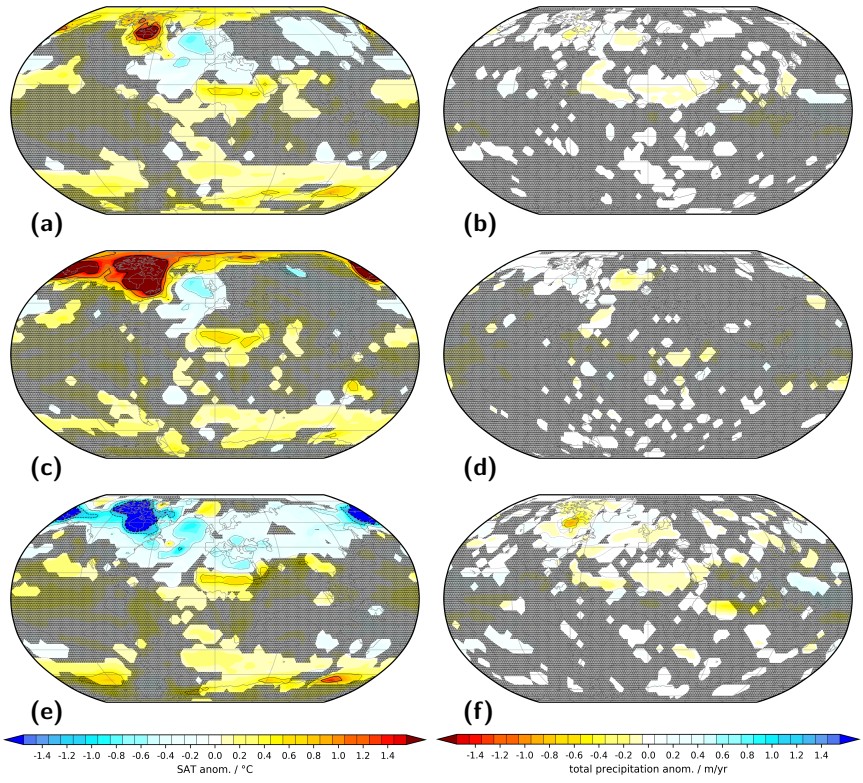

**Figure 28.** Impact of changes in Northern Hemisphere gateways on simulated mid-Pliocene surface air temperature (SAT, left) and total precipitation (right). Shown are anomalies Eoi400_GW with respect to Eoi400 for: a), b) annual mean; c), d) boreal winter (DJF) mean; e), f) boreal summer (JJA) mean. Contours illustrate: isolines of -1.5 (dashed, thick), -1.0 (dashed), -0.5 (dashed, thin), 0.0 (dotted, thin), 0.5 (solid, thin), 1.0 (solid), and 1.5 (solid, thick) °C (left) and m yr$^{-1}$ (right). We define SAT as the temperature at a height of two metres above the surface and total precipitation as the sum of large scale and convective precipitation in both solid and fluid phases. In hatched regions the anomaly is insignificant at 95% confidence interval based on a t-test.

**Table 1.** Simulations prepared with COSMOS in the framework of PlioMIP2. COSMOS setups based on Pre-Industrial (respectively modern) boundary conditions (PI), on implementation of mid-Pliocene paleogeography in COSMOS (MP), or on a mixed setup of modern geography with mid-Pliocene ice sheets (I), mid-Pliocene geography with modern ice sheets (O), or mid-Pliocene geography with modern ocean gateways (MP_G). Mid-Pliocene land sea mask (LSM) and geography (GEO) as provided by PRISM4 (Dowsett et al., 2016). Vegetation (VEG) always computed dynamically (dyn). Context of a specific simulation (Haywood et al., 2016): PlioMIP2 core simulation (CORE); PlioMIP2 Tier1 simulation (T1); PlioMIP2 Tier2 simulation (T2); Pliocene4Future (P4F), Pliocene4Pliocene (P4P). Total run time of a simulation given from initial year to end year, PlioMIP2 analysis period given in brackets. We present simulations beyond the official curriculum of PlioMIP2: additional sensitivity (sens), CMIP6 DECK and PMIP4. Carbon dioxide ($CO_2$, parts per million by volume (ppmv)) specified explicitly; other forcings (FORC) per specification of methane ($CH_4$) and nitrous oxide ($N_2O$) in parts per billion by volume (ppbv), eccentricity of the earth orbit (ecc), obliquity of the earth axis (obl), and longitude of perihelion (lonp):

– PlioMIP2 as per PI COSMOS: $CH_4$ – 808.000 (ppbv); $N_2O$ – 273.000 ppbv; ecc – 0.0167643; obl – 23.459277°; lonp – 280.32687°
– PMIP4 (Otto-Bliesner et al., 2017a): $CH_4$ – 808.249 ppbv; $N_2O$ – 273.021 ppbv; ecc – 0.016764; obl – 23.459°; lonp – 280.33°
Specifications of $CO_2$ for simulations not part of PlioMIP2:
– abrupt4xCO2: four times the PMIP4 PI concentration
– 1pctCO2: time-dependency as defined by Meinshausen et al. (2017)

| simulation | LSM | GEO | VEG | FORC | $CO_2$ | context | run (analysis) time / year |
|---|---|---|---|---|---|---|---|
| E280C (piControl) | PI | PI | dyn | PMIP4 | 284.317 | PMIP4/CMIP6 DECK | 800–2749 (2650–2749) |
| abrupt4xCO2 | PI | PI | dyn | PMIP4 | 1137.268 | CMIP6 DECK | 1850–2850 (1850–2850) |
| 1pctCO2 | PI | PI | dyn | PMIP4 | varying | CMIP6 DECK | 1850–2100 (1850–2100) |
| E280 | PI | PI | dyn | PlioMIP2 | 280.0 | CORE | 800–2749 (2650–2749) |
| E400 | PI | PI | dyn | PlioMIP2 | 400.0 | T2:P4F – T2:P4P | 800–2749 (2650–2749) |
| E560 | PI | PI | dyn | PlioMIP2 | 560.0 | T1:P4F | 800–2749 (2650–2749) |
| E600 | PI | PI | dyn | PlioMIP2 | 600.0 | sens | 800–2749 (2650–2749) |
| Eo280 | PI | O | dyn | PlioMIP2 | 280.0 | T2:P4P | 800–2749 (2650–2749) |
| Eo400 | PI | O | dyn | PlioMIP2 | 400.0 | T2:P4P | 800–2749 (2650–2749) |
| Ei280 | PI | I | dyn | PlioMIP2 | 280.0 | T2:P4P | 800–2749 (2650–2749) |
| Ei400 | PI | I | dyn | PlioMIP2 | 400.0 | T2:P4P | 800–2749 (2650–2749) |
| Eoi280 | MP | MP | dyn | PlioMIP2 | 280.0 | T2:P4P – T2P4F | 800–2749 (2650–2749) |
| Eoi350 | MP | MP | dyn | PlioMIP2 | 350.0 | T1:P4F – T1:P4P | 800–2749 (2650–2749) |
| Eoi400 | MP | MP | dyn | PlioMIP2 | 400.0 | CORE | 800–2749 (2650–2749) |
| Eoi400_GW | MP_G | MP_G | dyn | PlioMIP2 | 400.0 | sens | 800–2749 (2650–2749) |
| Eoi450 | MP | MP | dyn | PlioMIP2 | 450.0 | T1:P4F – T1:P4P | 800–2749 (2650–2749) |
| Eoi560 | MP | MP | dyn | PlioMIP2 | 560.0 | sens | 800–2749 (2650–2749) |

**Table 2.** Selected large-scale climate characteristics of equilibrium model simulations. Shown are: global averages of surface air temperature (SAT at 2 m above the ground / °C), the sum of large scale and convective precipitation (precip / mm d$^{-1}$), evaporation (evap / mm d$^{-1}$), cloud cover (cloud, fractional), surface albedo (alb., fractional), land surface albedo (alb. land, fractional), radiative imbalance at the top of the atmosphere (TOA imb. / W m$^2$), and sea surface temperature (SST / °C, uppermost ocean layer). Furthermore, we show hemispheric averages of sea ice area (SIA) for boreal winter to spring (FMA) and summer to autumn (ASO) of Northern (N) and Southern (S) Hemisphere in units of $10^6$ km$^2$. Ocean characteristics have been computed after conservative remapping of ocean model output to a regular 1°×1° grid. Seasonal sea ice given for N and S based on the definition of Northern Hemisphere sea ice summer and winter (Howell et al., 2016). See text for details of setup and configuration of the simulations.

| simulation | SAT | precip | evap | cloud | alb. | alb. land | TOA imb. | SST | SIA | | | |
| --- | --- | --- | --- | --- | --- | --- | --- | --- | --- | --- | --- | --- |
| | | | | | | | | | N FMA | N ASO | S FMA | S ASO |
| E280 | 13.50 | 2.72 | 2.73 | 0.619 | 0.169 | 0.301 | 1.73 | 17.53 | 17.86 | 9.20 | 7.55 | 18.24 |
| E280C | 13.56 | 2.73 | 2.74 | 0.619 | 0.169 | 0.301 | 1.69 | 17.58 | 17.68 | 9.17 | 7.65 | 18.33 |
| E400 | 15.64 | 2.84 | 2.85 | 0.606 | 0.156 | 0.288 | 1.79 | 19.00 | 14.58 | 5.29 | 5.51 | 14.36 |
| E560 | 18.19 | 2.98 | 3.00 | 0.591 | 0.144 | 0.278 | 1.96 | 20.86 | 11.88 | 0.78 | 2.80 | 8.51 |
| E600 | 18.89 | 3.01 | 3.03 | 0.588 | 0.141 | 0.276 | 1.99 | 21.38 | 11.02 | 0.29 | 2.20 | 7.65 |
| Ei280 | 13.79 | 2.73 | 2.75 | 0.622 | 0.163 | 0.291 | 1.70 | 17.58 | 17.57 | 8.90 | 4.54 | 18.24 |
| Ei400 | 16.04 | 2.86 | 2.87 | 0.608 | 0.148 | 0.274 | 1.73 | 19.14 | 14.50 | 4.21 | 1.91 | 12.41 |
| Eo280 | 14.59 | 2.78 | 2.80 | 0.611 | 0.158 | 0.281 | 1.84 | 18.14 | 14.65 | 6.90 | 7.08 | 16.81 |
| Eo400 | 16.69 | 2.90 | 2.92 | 0.598 | 0.147 | 0.270 | 1.97 | 19.64 | 12.75 | 2.58 | 4.59 | 12.26 |
| Eoi280 | 14.44 | 2.75 | 2.77 | 0.616 | 0.157 | 0.270 | 1.86 | 17.94 | 11.52 | 5.52 | 6.89 | 20.18 |
| Eoi350 | 15.96 | 2.84 | 2.86 | 0.607 | 0.146 | 0.260 | 1.86 | 19.00 | 9.90 | 3.12 | 3.60 | 15.07 |
| Eoi400 | 16.87 | 2.89 | 2.91 | 0.601 | 0.141 | 0.254 | 1.89 | 19.66 | 9.14 | 1.81 | 2.33 | 12.71 |
| Eoi400_GW | 16.93 | 2.88 | 2.90 | 0.601 | 0.141 | 0.254 | 1.92 | 19.54 | 11.80 | 1.89 | 2.09 | 12.29 |
| Eoi450 | 17.81 | 2.94 | 2.96 | 0.595 | 0.137 | 0.251 | 1.91 | 20.37 | 8.63 | 0.76 | 1.30 | 10.34 |
| Eoi560 | 19.39 | 3.02 | 3.05 | 0.587 | 0.132 | 0.245 | 2.08 | 21.54 | 7.70 | 0.05 | 0.77 | 8.70 |

**Table 3.** Average temperature and extent of the Equatorial Warm Pool (EqWP) for simulations with modern and mid-Pliocene geography. The EqWP is defined as the region where sea surface temperature (SST) exceeds 28 °C (Watanabe, 2008). Standard deviations of SST and area of the EqWP computed from annual mean data, spanning 100 model years in the case of model simulations, and the 30 most recent years (1989–2018 CE) in the case of ErSST.v5 observations (Huang et al., 2017) which are given for reference.

| simulation / data set | avg. SST of EqWP / °C | area of EqWP / $10^6$ km$^2$ |
|---|---|---|
| E280 | 29.0±0.1 | 57±11 |
| E400 | 29.8±0.2 | 100±8 |
| E560 | 31.0±0.2 | 139±4 |
| E600 | 31.4±0.2 | 148±3 |
| Eoi280 | 29.3±0.1 | 66±8 |
| Eoi350 | 29.7±0.2 | 92±7 |
| Eoi400 | 30.1±0.2 | 107±6 |
| Eoi400_GW | 30.1±0.2 | 108±6 |
| Eoi450 | 30.5±0.2 | 122±5 |
| Eoi560 | 31.4±0.2 | 143±3 |
| ErSST.v5 observations | 29.0±0.1 | 67±6 |

**Table 4.** Time average and variability of the maximum of the Atlantic Meridional Overturning Circulation (AMOC) in units of Sv ($1\,\text{Sv}\equiv10^6\,\text{m}^3\,\text{s}^{-1}$). The maximum AMOC is defined as the maximum of the basin-wide zonal- and time-integrated stream function in the Atlantic Ocean between 500 m and 1,500 m depth north of 20°N. Time variability indicated as ± one standard deviation.

| simulation | max. AMOC |
|---|---|
| E280 | 16.06±1.02 |
| E400 | 17.63±0.92 |
| E560 | 18.45±0.91 |
| E600 | 18.61±1.14 |
| Eoi280 | 16.54±1.07 |
| Eoi350 | 18.91±1.03 |
| Eoi400 | 19.48±1.32 |
| Eoi400_GW | 17.80±1.16 |
| Eoi450 | 19.36±1.20 |
| Eoi560 | 19.84±1.06 |

**Table 5.** Transient climate sensitivity (CS), equilibrium climate sensitivity (ECS), and earth system sensitivity (ESS), based on COSMOS PlioMIP2 simulations of 2 m temperature following the definition by Lunt et al. (2010). We define here the terms CS, ECS, and ESS as the surface air temperature change ($\Delta$SAT) that arises if the volume mixing ratio of carbon dioxide ($CO_2$) is doubled from 280 parts per million by volume (ppmv) to 560 ppmv. We show: ECS for modern geography ($ECS_{modern}$, based on E560 and E280); ECS for mid-Pliocene geography ($ECS_{mid-Pliocene}$, based on Eoi560 and Eoi280); ESS of the mid-Pliocene ($ESS_{mid-Pliocene}$, based on Eoi560 and E280); transient CS, derived from the CMIP6 one-percent 'ramp-up' simulation ($CS_{ramp}$, based on 1pctCO2 and E280C). All values based on annual averages and standard deviations over the analysis period of 100 model years; except for $CS_{ramp}$, that is based on an average over 29 model years, centred on the year that is closest to a $CO_2$ forcing of 560 ppmv.

| climate system metric | $\Delta$SAT / °C |
|:---:|:---:|
| $ECS_{modern}$ | 4.7$\pm$0.4 |
| $ECS_{mid-Pliocene}$ | 4.9$\pm$0.4 |
| $ESS_{mid-Pliocene}$ | 5.9$\pm$0.5 |
| $CS_{ramp}$ | 2.1$\pm$0.6 |

**Table 6.** Root mean square deviation (RMSD) between simulated and reconstructed (PRISM3; Dowsett et al., 2009, 2013) sea surface temperature. Model data is derived at core location by bilinear interpolation. We present: global RMSD ($RMSD_{glob}$), considering all available reconstruction sites; North Atlantic to Arctic Ocean RMSD ($RMSD_{AtlArc}$, considering reconstructions north of 30.00°N and between 70.00°E and 8.25°W); North Atlantic Ocean only RMSD ($RMSD_{Atl}$, considering reconstructions between 70.00°E and 8.25°W and between 30.00°N and 60.00°N).

| simulation | $RMSD_{glob}$ | $RMSD_{AtlArc}$ | $RMSD_{Atl}$ |
|:---:|:---:|:---:|:---:|
| Eoi280 | 2.94 | 5.04 | 4.13 |
| Eoi350 | 2.35 | 3.72 | 2.85 |
| Eoi400 | 2.29 | 3.18 | 2.57 |
| Eoi400_GW | 2.32 | 3.26 | 2.84 |
| Eoi450 | 2.49 | 2.95 | 2.64 |
| Eoi560 | 3.09 | 2.89 | 2.95 |

**Table 7.** Root mean square deviation (RMSD) between simulated and various reconstructed sea surface temperature data sets. Considered are alkenone ($U^K_{37'}$) based reconstructions by Foley and Dowsett (2019) referring to time windows from $3.205 \pm 0.005$ (10 ka) and $3.205 \pm 0.015$ (30 ka) million years before present (Ma BP), $U^K_{37'}$ based on PlioVAR synthesis and the new BAYSPLINE calibration (McClymont et al., 2020), as well as magnesium-to-calcium ratio (Mg/Ca) based on PlioVAR synthesis and the new BAYMAG calibration (McClymont et al., 2020). See text for details. Similar to Table 6 we present: global RMSD ($RMSD_{glob}$), considering all available reconstruction sites; North Atlantic to Arctic Ocean RMSD ($RMSD_{AtlArc}$, considering reconstructions north of 30.00°N and between 70.00°E and 8.25°W); North Atlantic Ocean only RMSD ($RMSD_{Atl}$, considering reconstructions between 70.00°E and 8.25°W and between 30.00°N and 60.00°N).

| reconstruction | number of records | $RMSD_{glob}$ | $RMSD_{Atl}$ | $RMSD_{AtlArc}$ |
|---|---|---|---|---|
| $U^K_{37'}$, 10 ka (Foley and Dowsett, 2019) | 33 | 3.94 | 3.00 | 3.87 |
| $U^K_{37'}$, 30 ka (Foley and Dowsett, 2019) | 37 | 3.92 | 2.77 | 3.72 |
| $U^K_{37'}$, PlioVAR synthesis (McClymont et al., 2020) | 23 | 3.81 | 2.66 | 2.54 |
| $U^K_{37'}$, BAYSPLINE (McClymont et al., 2020) | 23 | 3.61 | 2.55 | 2.41 |
| Mg/Ca, PlioVAR synthesis (McClymont et al., 2020) | 12 | 3.45 | 3.84 | 3.84 |
| Mg/Ca, BAYMAG (McClymont et al., 2020) | 11 | 3.55 | 5.82 | 5.82 |

**Table 8.** Vegetation shifts at Northern Hemisphere high latitudes simulated by the dynamic vegetation module of JSBACH depending on carbon dioxide concentration. Shown are simulated shifts for three climate states with contemporary (E400, E560, and E600) and five climate states with mid-Pliocene (Eoi280, Eoi350, Eoi400, Eoi450, Eoi560) geography. Illustrated are spatial shifts in the tree line, that is arbitrarily defined here as the isoline of 15% tree cover, for four different regions: Western Canada (WC), Eastern Canada (EC), Greenland (G), and Eastern Siberia (ES). We specify the northward shift for WC, EC, and ES, and the eastward shift for G. Absolute geographical coordinates of the tree line provided in units of degrees longitude/latitude, shift of the tree line (in brackets) with respect to simulation E280 provided in units of kilometres. For reference, the geographic location of the tree line in simulation E280 is given as well. Note that the tree line location is spatially very variable within the regions (Fig. 25). In simulations with high concentrations of carbon dioxide, the tree line defined here is absent as it reaches beyond the northern margin of the continent. This is reflected by absence of a further shift in simulations with higher carbon dioxide.

| simulation | WC | | EC | | G | | ES | |
|---|---|---|---|---|---|---|---|---|
| E280 | 52°N | | 57°N | | 296°E | | 59°N | |
| E400 | 56°N | (450 km) | 61°N | (450 km) | 296°E | (n.a.) | 72°N | (1,450 km) |
| E560 | 69°N | (1,900 km) | 71°N | (1,550 km) | 296°E | (n.a.) | 72°N | (1,450 km) |
| E600 | 69°N | (1,900 km) | 72°N | (1,650 km) | 296°E | (n.a.) | 72°N | (1,450 km) |
| Eoi280 | 54°N | (200 km) | 67°N | (1,100 km) | 309°E | (500 km) | 71°N | (1,350 km) |
| Eoi350 | 57°N | (550 km) | 71°N | (1,550 km) | 313°E | (650 km) | 72°N | (1,450 km) |
| Eoi400 | 72°N | (2,200 km) | 77°N | (2,200 km) | 315°E | (700 km) | 72°N | (1,450 km) |
| Eoi450 | 72°N | (2,200 km) | 80°N | (2,550 km) | 316°E | (750 km) | 72°N | (1,450 km) |
| Eoi560 | 72°N | (2,200 km) | 80°N | (2,550 km) | 318°E | (850 km) | 72°N | (1,450 km) |