# Peer review of "Contribution of the coupled atmosphere–ocean–sea ice–vegetation model COSMOS to the PlioMIP2"

_Climate of the Past, 2020_

## Referee Comment (RC1) · Anonymous Referee #1 · 2 Mar 2020

This draft and a companion paper (Samakinwa et al. 2020) nicely summarize the results of PlioMIP2 experiments conducted by COSMOS. The current draft reports the results of the ESM simulations with dynamic vegetation that is a challenging and a novel point of this study compared to other PlioMIP2 studies. The authors also show results of additional experiments, ocean gateway experiments, that is not included in the official PlioMIP2 protocol. The results are very usefulness because such drastic changes in ocean gateways would be essential features of the Mid-Pliocene climate. Due to the meaningful contributions above, I recommend acceptance of this paper, after some revisions to clarify some points listed below.

[Figure]

Major comments

More than four models participated to the PlioMIP2 and the results of the simulations were already reported in the PlioMIP2 special issue (https://www.clim-past-discuss.net/special_issue91.html). Why did you limit the comparison of your results with only four models (Page 4 line 15)? In sections 3 and 4, the large-scale features of this model should be briefly compared with other models. Such simple comparisons may be effective to demonstrate that the dynamic vegetation or idealized closed gateways are unique points of this study compared to the other PlioMIP2 studies.

You mentioned that the simulated Mid-Pliocene climate in this model are less equilibrated than the control run (Page 13 line 13-15). It is also noted that the centennial-scale AMOC fluctuations are found after the target period of this study (Page 11, line 11-12). It is much better to add figure showing evolutions of SAT (and AMOC or deep ocean temperature, for example) during the model integrations, especially for e280, eoi400 and eoi400_GW runs, similar to what you showed in your previous paper (Figure 6 of 2012 GMD paper). It would be OK to include it into the supplement if there would be nothing particular to be noted in the main text.

Page 22, line 14: "a northward shift" Does this also mean a northward shift of vegetation in the Southern Hemisphere? Or a poleward shift? If a northward shift is also found over the Southern Hemisphere, it indicates a equatorward shift, contrasting to the NH. What are the factors?

Specific comments

Page 13, line 20: lower elevation and the absence of ice sheets

Page 14, line 9-16: The discussion here is closely related to variation in ITCZ related to interhemispheric asymmetry in energy balance. It is better to refer any previous studies here, for example doi: 10.1175/2007JCLI2146.1.

Page 14, Line 33: model. Raomo et al.

Page 24 line 6: to to

---

## Referee Comment (RC2) · Anonymous Referee #2 · 31 Mar 2020

Stepanek et al. have produced a very thorough coverage of their contribution to phase 2 of PlioMIP and associated climate model simulations. This is an important individual contribution to a significant undertaking of the international palaeoclimate modelling community and there is lots of great science contained within this manuscript. With the inclusion of more than the PlioMIP phase 2 simulations, I do think this work could easily have been split across more than one paper, particularly the more detailed analysis of some of the extra simulations. However, having them all in one place also has some advantages, even if this makes it a long paper. Although it is generally a very good manuscript, there are a few things that could be improved. Firstly, the language needs to be improved and simplified to make it more understandable. In the detailed

comments below are some suggestions of sentences that need changing, but many more could be improved.

Secondly, and probably most importantly, although there is a good data-model comparison, this is only done against PRISM3, rather than the more up-to-date and probably more appropriate PRISM4 dataset. There is some utility in doing this comparison, partly as there is a long legacy of doing so, enabling comparison with previous data-model comparisons. However, in PlioMIP2 the move is towards simulating a timeslice within the mid-Pliocene, trying to select an appropriate set of boundary conditions to one particular time and refining the proxy datasets to allow a more appropriate data-model comparison (Haywood et al., 2013b). There are now two datasets that allow for data-model comparison with the marine isotope stage KM5c timeslice, the PRISM4 reconstruction of the North Atlantic (Dowsett et al., 2016), as set out in the experimental design (Haywood et al., 2016) and the global reconstruction of McClymont et al. (2020). The discussion of the data-model comparison, covering the second half of page 21, will need to be updated considering the timeslice and the subsequent data-model comparison.

There are several figures that could be changed to make things clearer to the reader. A number of times in the figure captions you refer to SIC as sea ice compactness. In the standard CMOR variable names SIC is sea ice coverage (or sea ice area fraction) and the images look like they are indeed this variable. In Figure 6 and 16 it is very hard to see changes in the North Atlantic Deep Water (NADW). Consider showing these in both anomaly and absolutes, so that the magnitude of changes can be seen even if they are small. Figure 13 would be improved by plotting the pre-industrial (E280) sea ice, probably as a first row at the top of the figure. If a PRISM4 comparison is done, it would be useful to include this in Figure 21 (or a new figure).

Minor corrections:

Page 1, line 5: Should read "With this manuscript we present . . ."

Page 2, line 2: Should read "They deliver knowledge that is key to preparing humankind for future environmental conditions . . ."

Page 2, line 14: Should read ". . . furthermore enables us to test our model against climate states that are warmer . . ."

Page 2, line 16: Should read "Successful reproduction of past climates increases confidence in a climate model . . ."

Page 2, line 18: Should read ". . . a warmer-than-present climate state has been found.

Page 2, line 22 and throughout the manuscript: The mid-Pliocene is not a formal stratigraphic unit, so it should not be capitalised. All "Mid-Pliocene" occurrences should be changed to "mid-Pliocene".

Page 2, line 24: Remove "respectively disagreement".

Page 4, line 10: Should read "One difference is the utilization of the dynamic vegetation . . ."

Page 4, line 16: Should read "Yet, the COSMOS has characteristics . . ."

Page 4, line 21: Should read "Furthermore, in PlioMIP1 the COSMOS was shown to predict . . ."

Page 5, line 28: Should read "It is able to adapt global vegetation distribution and related albedo- and evapotranspiration-feedbacks in the presence of changes in ambient climate . . ."

Page 6, line 12: Should read "As an important process for breaking stratification, the MPIOM . . ."

Page 8, line 15: Should read "The starting points are the PRISM4 . . ."

Page 8, line 31: Remove "as well".

Page 11, line 15: Should read "we follow the extended modelling protocol".

Page 12, line 31: Should read "results presented below are based on an averaging period"

Page 13, line 5: Is the 2.13°C surface air temperature (SAT) at the ocean surface or sea surface temperature (SST)? I suspect the latter, but it is not entirely clear at the moment.

Page 13, line 13: I'm not sure what albedo changes are being compared here (-16.6% vs -15.6%) is this ocean vs land? Whatever this is, it needs to be made clear.

Page 13, line 20: Should read "There are only a few regions . . ."

Page 14, line 15: Should read "Predominant drying is apparent . . ."

Page 14, line 20: Should read "In contrast, changes in the boreal autumn . . ."

Page 14, line 23: Should read "Low latitudes of the oceans also have different characteristics . . ."

Page 14, line 23: Should read "We demonstrate this with the example . . ."

Page 14, line 32: Should read ". . . confirms in our model, as suggested by Raymo et al. (1996) and Dowsett et al. (2009), that mid-Pliocene . . ."

Page 16, line 13: Should read ". . . temperature gradient are also seen in the annual mean of global SAT anomalies under changes in carbon dioxide"

Page 17, line 2: Should read ". . . SST, causing global mean values of and SAT to have reached similar values at the end of the simulation."

Page 17, line 8: Should read ". . . we find a large impact on the hydrological cycle . . ."

Page 18, line 9: Should read ". . . the possibility to go beyond CS and ECS for both modern and mid-Pliocene geography and derive earth system sensitivity . . ."

Page 18, line 17: Should read "There is a significant difference between these ECS values and those derived . . ."

[Figure]

Page 19, line 30: Should read "Yet, high temporal variability . . ."

Page 21, line 13: Should read "still the case, for example there is a significant mismatch . . ."

Page 26, line 3: I don't think that you should use the verb "confirm" when only some of the models agree with this statement. Many of the models also disagree.

Page 26, line 24: Should read "The mid-Pliocene combines estimates of carbon dioxide levels . . ."

Page 27, line 10: Should read "Hence, making inferences from modelled or reconstructed climate conditions of the mid-Pliocene with respect to . . ."

Page 27, line 12: Should read "This has been stated by . . ."

Page 30, line 21: Should read ". . . in the context of Pliocene4Future, . . ."

Page 30, line 25: Should read ". . . with potential threats to the food chain . . ."

New references:

Haywood, A. M., Dolan, A. M., Pickering, S. J., Dowsett, H. J., McClymont, E. L., Prescott, C. L., Salzmann, U., Hill, D. J., Hunter, S. J., Lunt, D. J., Pope, J. O., and Valdes, P. J., 2013b. On the identification of a Pliocene time slice for data-model comparison, Phil. Trans. Roy. Soc. A, 371, 20120515.

McClymont, E. L., Ford, H. L., Ling Ho, S., Tindall, J. C., Haywood, A. M., Alonso-Garcia, M., Bailey, I., Berke, M. A., Littler, K., Patterson, M., Petrick, B., Peterse, F., Ravelo, A. C., Risebrobakken, B., De Schepper, S., Swann, G. E. A., Thirumalai, K., Tierney, J. E., van der Weijst, C., and White, S., 2020. Lessons from a high CO2 world: an ocean view from $\sim$ 3 million years ago, Clim. Past Discuss., https://doi.org/10.5194/cp-2019-161, in review, 2020.

---

## Author Comment (AC1) · 29 Apr 2020

**Author response to comments by anonymous referee#1**

**Major comments**

**"Why did you limit the comparison of your results with only four models? In sections 3 and 4, the large-scale features of this model should be briefly compared with other models. Such simple comparisons may be effective to demonstrate that the dynamic vegetation or idealized closed gateways are unique points of this study compared to the other PlioMIP2 studies."**

We apologize if our formulation was ambiguous here. We wrote:

*"The COSMOS is of comparably low spatial resolution and there is only one other PlioMIP2 climate model that employs a similarly low resolution in the atmosphere. The COSMOS is also an older model in the PlioMIP2 model ensemble, in particular in comparison to four PlioMIP2 models that were published since 2017. Hence, the model is not anymore state–of–the–art."*

Our aim in this paper is actually focused on presenting the simulations that we contribute to the PlioMIP2 ensemble; our aim is not so much to compare results from our model to those derived from other models. The model-intercomparison within PlioMIP2 is the focus of the ensuing model-model-intercomparison phase. There is already one such publication present that studies the model-model intercomparison of large-scale features of Mid-Pliocene climate (Haywood et al., 2020).

In the text passage, that anonymous referee #1 points to, we actually just wanted to clarify that some *characteristics (not results)* of our model are different to those from other models. We argued based on an already available overview of characteristics of the contributing models (Table 1 of Haywood et al., 2020). Our aim was not to refer to detailed results from individual models.

Yet, we fully acknowledge that anonymous reviewer #1 has a valid point in that we could present a – very brief – summary of differences in large scale characteristics of mid-Pliocene climate produced with our model in comparison to other studies. Since an extensive and very detailed model-model-intercomparison has already been presented by Haywood et al. (2020), we will base our comparison on the overview on PlioMIP2 model results presented by them.

We feel that a comparison of our results to those from other modelling groups has, in the context of this model description paper, rather the characteristic of a discussion (more precisely, a discussion of the comparison of our results to those by other modelling groups). We believe such text is not a genuine part of the results section. Hence, we add the model comparison as an additional subsection to section 4 "Discussion". We will place this new subsection right at the start of section 4, so that subsection 4.1, "The added value of COSMOS Mid-Pliocene simulations in PlioMIP2", will become subsection 4.2 in a revised manuscript.

The additional subsection will be named "4.1 Comparison of selected Mid-Pliocene large scale climate patterns to the PlioMIP2 model ensemble". The text will be incorporated in a revised version of the manuscript.

Unfortunately, we will not be able to produce a model-intercomparison that aims at quantification of the importance of dynamic vs. prescribed vegetation, and at modern vs. Pliocene gateways, which the reviewer aims at. There are no other model simulations that employ dynamic vegetation (see Table 1 by Haywood et al., 2020), and our model seems to be the only one for which a dedicated PlioMIP2 mid-Pliocene simulation with modern gateways is available. Hence, we lack the data from other models on which we could base a robust multi-model comparison "dynamic vs. fixed vegetation" and "modern vs. reconstructed gateways".

**"It is much better to add figure showing evolutions of SAT (and AMOC or deep ocean temperature, for example) during the model integrations, especially for e280, eoi400 and eoi400_GW runs, similar to what you showed in your previous paper (Figure 6 of 2012 GMD paper). It would be OK to include it into the supplement if there would be nothing particular to be noted in the main text."**

We thank the reviewer for the proposal to better highlight the state of equilibrium of our simulations, beyond what has already been done based on Table 2 (column "TOA imbalance") of our manuscript. We already have prepared an illustration that shows the evolution of global average potential temperature in the ocean over the spinup- and analysis-periods of the various PlioMIP2 simulations with COSMOS. As suggested by reviewer #1, we propose to present an analysis of the time evolution of ocean temperature in simulations E280, Eoi400, and Eoi400_GW at the surface, and at three different ocean depth layers (1000 m, 2000 m, and 3000 m) in a supplement to the manuscript. Yet, we propose to show SST, that is closely coupled to SAT outside sea ice regions, instead of SAT, towards a better comparability of surface temperature and subsurface ocean temperature. This supplementary figure is presented at the end of this comment. In the main text we refer to this supplementary analysis in the following manner by means of an extension of the text at page 13, line 15 (new text in red):

"The simulated Mid-Pliocene climate state Eoi400 is over the analysis period slightly less equilibrated (by 0.16 W m²) than the reference climate state E280, as evidenced by top of the atmosphere (TOA) radiative imbalance. The generally high TOA radiative imbalance across the simulation ensemble (1.7–2.0 W m²) is comparable to imbalances in the model that are present in the framework of PlioMIP1 (Stepanek and Lohmann, 2012). Radiative imbalance is related to the slow response of the ocean to changes in carbon dioxide forcing as shown before (Li et al., 2013). In our case, a combination of changes in carbon dioxide and geographic boundary conditions causes a slow equilibration process that is not fully finished at the end of the model spinup (Fig. S1 in the supplement to this article). In particular, simulation Eoi400 still exhibits a temperature trend of about $7 \cdot 10^{-4}$ °C yr⁻¹ at 3000 m depth over the analysis period. On the other hand, the ocean surface, on which PlioMIP2 analyses focus heavily, is in simulation Eoi400 in quasi-equilibrium. During the analysis period, the simulation is subject to an ocean surface temperature trend that is actually below the respective trend in the PI control state E280. Furthermore, the ocean surface slightly cools in the analysed portion of Mid-Pliocene core simulation Eoi400. This suggests that the diagnosed surface temperature trend is, from the view point of the ocean surface, largely overprinted by internal variability. Similarity of TOA imbalance and small residual ocean surface temperature trends across the PlioMIP2 COSMOS simulation ensemble demonstrate, despite incomplete model equilibration, expediency of simulation Eoi400 and other COSMOS PlioMIP2 simulations for the study of climate anomalies."

**"Page 22, line 14: "a northward shift" Does this also mean a northward shift of vegetation in the Southern Hemisphere? Or a poleward shift? If a northward shift is also found over the Southern Hemisphere, it indicates a equatorward shift, contrasting to the NH. What are the factors?"**

Although the COSMOS simulate vegetation dynamics across the entire land surface of both the Northern and Southern Hemisphere (with the exception of regions covered by ice sheets), our focus on analysis is largely on vegetation shifts of the Northern Hemisphere. Polar amplification of global climate anomalies is much larger in the Northern Hemisphere as evidenced in many other publications for both future and Cenozoic climate, and this is also evident in our simulation ensemble. At the text location, to which the reviewer points, we actually specifically talk about northward shifts in the Northern Hemisphere, and we clarify the text in a revised manuscript accordingly.

Regarding the question "What are the factors?" we assume that the reviewer asks us to link shifts in vegetation cover to changes in the simulated climate variables, foremost temperature and precipitation. We have analysed the correlation between simulated vegetation cover and monthly mean climate variables independently of this manuscript. This research has produced a complex picture on correlations between climate and vegetation. This is understandable as the analysis has to go beyond simulated climate and must also consider changes in the prescribed soil cover and the various PFT's tolerance to extremes in temperature and rain. To present such an analysis we would have to include additional data that is beyond the monthly averages of temperature and precipitation presented in our publication. Currently, we are analysing climate extremes in the Pliocene. Studying the response of vegetation will likely be a part of this. Yet, we fear that additional information, that needs to be conveyed towards a proper analysis of the factors, is well beyond the scope of this modelling paper and must be left for a future publication.

**Specific comments**

Page 13, line 20: lower elevation and the absence of ice sheets
The suggested change has been implemented into a revised manuscript.

Page 14, line 9-16: The discussion here is closely related to variation in ITCZ related to interhemispheric asymmetry in energy balance. It is better to refer any previous studies here, for example doi: 10.1175/2007JCLI2146.1.

We thank the reviewer for pointing out that reporting our results could be done in a less descriptive way, paying more attention to the mechanisms behind the changes in low latitude rain patterns. We refer now to previous literature, pointing out the connection between asymmetry of interhemispheric energy balance and related shifts of precipitation patterns in low latitudes. Consequently, we have added the following text at the end of line 16:

"Pronounced changes in low-latitude precipitation, Mid-Pliocene vs. PI, are related to the asymmetric warming between hemispheres. It has been shown by various authors that the ITCZ shifts towards the warmer hemisphere via links between tropical and extratropical climate (Haug et al., 2001; Broccoli et al., 2006; Kang et al., 2008; Deplazes et al., 2013; Schneider et al., 2014). The warmer hemisphere is in our Mid-Pliocene simulations the Northern Hemisphere, that warms more widespread than the Southern Hemisphere and across all seasons (Fig. 3)."

Page 14, line 33: model. Raomo et al.
We agree with the reviewer that the sentence was far too long to easily grasp it's meaning. We have split the text accordingly as follows (changes in red):

"In contrast, we find that the maximum stream function of the Atlantic Meridional Overturning Circulation (AMOC) is increased in simulation Eoi400 (Table 4) with respect to simulation E280. Hence, our model confirms Raymo et al. (1996) and Dowsett et al. (2009) in that Mid-Pliocene thermohaline circulation was higher than today."

Page 24, line 6: to to
Erroneous repeating of the word "to" has been fixed.

**References**

Broccoli, A. J., Dahl, K. A., and Stouffer, R. J.: Response of the ITCZ to Northern Hemisphere cooling, Geophys. Res. Lett., 33, L01702, doi:10.1029/2005GL024546, 2006.

Deplazes, G., Lückge, A., Peterson, L., Timmermann, A., Hamann, Y., Hughen, K. A., Röhl, U., Laj, C., Cane, M. A., Sigman, D. M., and Haug, G. H.: Links between tropical rainfall and North Atlantic climate during the last glacial period, Nature Geosci., 6, 213–217, https://doi.org/10.1038/ngeo1712, 2013.

Dowsett, H. J., Robinson, M. M., and Foley, K. M.: Pliocene three-dimensional global ocean temperature reconstruction, Clim. Past, 5, 769--783, https://doi.org/10.5194/cp-5-769-2009, 2009.

Haug, G. H., Hughen, K. A., Sigman, D. M., Peterson, L. C., and Röhl, U.: Southward Migration of the Intertropical Convergence Zone Through the Holocene, Science, 293, 1304–1308, https://doi.org/10.1126/science.1059725, 2001.

Haywood, A. M., Tindall, J. C., Dowsett, H. J., Dolan, A. M., Foley, K. M., Hunter, S. J., Hill, D. J., Chan, W.-L., Abe-Ouchi, A., Stepanek, C., Lohmann, G., Chandan, D., Peltier, W. R., Tan, N., Contoux, C., Ramstein, G., Li, X., Zhang, Z., Guo, C., Nisancioglu, K. H., Zhang, Q., Li, Q., Kamae, Y., Chandler, M. A., Sohl, L. E., Otto-Bliesner, B. L., Feng, R., Brady, E. C., von der Heydt, A. S., Baatsen, M. L. J., and Lunt, D. J.: A return to large-scale features of Pliocene climate: the Pliocene Model Intercomparison Project Phase 2, Clim. Past Discuss., https://doi.org/10.5194/cp-2019-145, in review, 2020.

Kang, S. M., Held, I. M., Frierson, D. M., and Zhao, M.: The Response of the ITCZ to Extratropical Thermal Forcing: Idealized Slab-Ocean Experiments with a GCM, J. Climate, 21, 3521–3532, https://doi.org/10.1175/2007JCLI2146.1, 2008.

Li, C., von Storch, J., and Marotzke, J.: Deep-ocean heat uptake and equilibrium climate response, Clim. Dyn., 40, 1071--1086, https://doi.org/10.1007/s00382-012-1350-z, 2013.

Raymo, M. E., Grant, B., Horowitz, M., and Rau, G. H.: Mid-Pliocene warmth: stronger greenhouse and stronger conveyor, Mar. Micropaleontol., 27, 313--326, https://doi.org/10.1016/0377-8398(95)00048-8, 1996.

Schneider, T., Bischoff, T., and Haug, G.: Migrations and dynamics of the intertropical convergence zone, Nature, 513, 45–53, https://doi.org/10.1038/nature13636, 2014.

Stepanek, C. and Lohmann, G.: Modelling mid-Pliocene climate with COSMOS, Geosci. Model Dev., 5, 1221--1243, https://doi.org/10.5194/gmd-5-1221-2012, 2012.

**Supplement to cp-2020-10**

[Figure]

**Figure S1:** Time evolution of potential seawater temperature at various ocean depths between surface and 3000 m as a diagnostic for model equilibration. Shown is the evolution of temperature over the runtime of PlioMIP2 COSMOS core simulations Eoi400 and E280 and of the sensitivity study with Mid-Pliocene geography but modern states of Bering Strait, Hudson Bay, and Canadian Arctic Archipelago, Eoi400_GW. Vertical red bars denote the start of the PlioMIP2 analysis period that ends at model year 1949 at the end of the time period shown in the illustration. Temperature trends during the analysis period are given in brackets in the legend after the respective simulation name. These indicate that simulations are in a quasi-equilibrium over the PlioMIP2 analysis period. Note the difference in scale between ocean surface and ocean subsurface temperatures.

---

## Author Comment (AC2) · 29 Apr 2020

**Author response to comments by anonymous referee#2**

**Major comments**

**"Firstly, the language needs to be improved and simplified to make it more understandable."**

We very much appreciate the reviewer's effort in pointing out various locations where the language needs to be overhauled. Those locations, where the reviewer gives suggestions, have been already fixed in a revised version of the manuscript. Furthermore, we will carefully check other locations prior to submission of the revised manuscript. Does the editor suggest submission of the manuscript to a language editing service?

**"Secondly, and probably most importantly, although there is a good data-model comparison, this is only done against PRISM3, rather than the more up-to-date and probably more appropriate PRISM4 dataset. There is some utility in doing this comparison, partly as there is a long legacy of doing so, enabling comparison with previous data-model comparisons. However, in PlioMIP2 the move is towards simulating a timeslice within the mid-Pliocene, trying to select an appropriate set of boundary conditions to one particular time and refining the proxy datasets to allow a more appropriate data-model comparison (Haywood et al., 2013b). There are now two datasets that allow for data-model comparison with the marine isotope stage KM5c timeslice, the PRISM4 reconstruction of the North Atlantic (Dowsett et al., 2016), as set out in the experimental design (Haywood et al., 2016) and the global reconstruction of McClymont et al. (2020). The discussion of the data-model comparison, covering the second half of page 21, will need to be updated considering the timeslice and the subsequent data-model comparison. […] If a PRISM4 comparison is done, it would be useful to include this in Figure 21 (or a new figure)."**

We thank the reviewer for highlighting that our model-data comparison could be more up to date. In the revised manuscript we will, in addition to the model-data comparison based on PRISM3 that is already part of the manuscript, also consider the reconstructions by McClymont et al. (2020) and by Foley and Dowsett (2019). To this end we will create an additional figure panel that presents a similar model-data comparison as done with PRISM3 data in Fig. 21. We propose to do this only for the CORE simulation Eoi400 to not create the need for more than one additional figure panel. The updated proxy data sets contain six different individual data sets - those by Foley and Dowsett (2019), with both a 10 ka and a 30 ka time window; and the BAYSPLINE and BAYMAG data sets by McClymont et al. (2020), as well as the respective previous data sets based on the calibration by Müller98 as outlined by McClymont et al. (2020) in their manuscript. We will treat these data sets in the model data comparison independently from each other. Table 6 will be extended to also convey information on model data comparison results based on the updated proxy data. Our discussion will be updated accordingly, where we will focus on results of our model-data-comparison that differ in dependence of the employed proxy reconstruction.

Please note, that we will need a bit more time to finish this part of the manuscript update, as the final PlioVAR synthesis product by McClymont et al. (2020) has not yet been released (Erin McClymont, pers. comm.).

**"A number of times in the figure captions you refer to SIC as sea ice compactness. In the standard CMOR variable names SIC is sea ice coverage (or sea ice area fraction) and the images look like they are indeed this variable."**

The variable "SICOMO" in the COSMOS is called "sea ice compactness" and it indeed represents sea ice concentration, or sea ice area fraction. To maintain consistency with other publications, we have replaced all occurrences of the term "sea ice compactness" in a revised manuscript by the term "sea ice coverage".

**"In Figure 6 and 16 it is very hard to see changes in the North Atlantic Deep Water (NADW). Consider showing these in both anomaly and absolutes, so that the magnitude of changes can be seen even if they are small."**

We agree with the reviewer that showing anomalies in addition to absolute values of the AMOC would improve the visibility of changes between simulations. We hence have created two additional Figures: a new Fig. 7, that is like Fig. 6, but shows AMOC anomalies, and a new Fig. 18, that is like Fig. 16 in the discussion manuscript (Fig. 17 in the revised manuscript), but shows AMOC anomalies. We have added additional text to section 3.1.2, 3.5, and 3.9 that describe results derived from these figures.

**"Figure 13 would be improved by plotting the pre-industrial (E280) sea ice, probably as a first row at the top of the figure."**

We have created an updated version of Fig. 13 where we show the results from E280 in the top row.

**Specific comments**

Page 1, line 5: Should read "With this manuscript we present . . ."

fixed

Page 2, line 2: Should read "They deliver knowledge that is key to preparing humankind for future environmental conditions . . ."

fixed

Page 2, line 14: Should read ". . . furthermore enables us to test our model against climate states that are warmer . . ."

fixed

Page 2, line 16: Should read "Successful reproduction of past climates increases confidence in a climate model . . ."

fixed

Page 2, line 18: Should read ". . . a warmer-than-present climate state has been found.

fixed

Page 2, line 22 and throughout the manuscript: The mid-Pliocene is not a formal stratigraphic unit, so it should not be capitalised. All "Mid-Pliocene" occurrences should be changed to "mid-Pliocene".

fixed

Page 2, line 24: Remove "respectively disagreement".

fixed

Page 4, line 10: Should read "One difference is the utilization of the dynamic vegetation . . ."

fixed

Page 4, line 16: Should read "Yet, the COSMOS has characteristics . . ."

fixed

Page 4, line 21: Should read "Furthermore, in PlioMIP1 the COSMOS was shown to predict . . ."

fixed

Page 5, line 28: Should read "It is able to adapt global vegetation distribution and related albedo- and evapotranspiration-feedbacks in the presence of changes in ambient climate . . ."

fixed

Page 6, line 12: Should read "As an important process for breaking stratification, the MPIOM . . ."

fixed

Page 8, line 15: Should read "The starting points are the PRISM4 . . ."

fixed

Page 8, line 31: Remove "as well".

fixed

Page 11, line 15: Should read "we follow the extended modelling protocol".

We mean here that we follow a modelling protocol that is extended in comparison to the one provided by Haywood et al. (2010, 2011). We have clarified the sentence accordingly.

Page 12, line 31: Should read "results presented below are based on an averaging period"

fixed

Page 13, line 5: Is the 2.13°C surface air temperature (SAT) at the ocean surface or sea surface temperature (SST)? I suspect the latter, but it is not entirely clear at the moment.

You are right – fixed.

Page 13, line 13: I'm not sure what albedo changes are being compared here (-16.6% vs -15.6%) is this ocean vs land? Whatever this is, it needs to be made clear.

You are right – fixed.

Page 13, line 20: Should read "There are only a few regions . . ."

fixed

Page 14, line 15: Should read "Predominant drying is apparent . . ."

fixed

Page 14, line 20: Should read "In contrast, changes in the boreal autumn . . ."

fixed

Page 14, line 23: Should read "Low latitudes of the oceans also have different characteristics . . ."

fixed

Page 14, line 23: Should read "We demonstrate this with the example . . ."

fixed

Page 14, line 32: Should read ". . . confirms in our model, as suggested by Raymo et al. (1996) and Dowsett et al. (2009), that mid-Pliocene . . ."

fixed

Page 16, line 13: Should read ". . . temperature gradient are also seen in the annual mean of global SAT anomalies under changes in carbon dioxide"

fixed

Page 17, line 2: Should read ". . . SST, causing global mean values of and SAT to have reached similar values at the end of the simulation."

fixed

Page 17, line 8: Should read ". . . we find a large impact on the hydrological cycle . . ."

fixed

Page 18, line 9: Should read ". . . the possibility to go beyond CS and ECS for both modern and mid-Pliocene geography and derive earth system sensitivity . . ."

fixed

Page 18, line 17: Should read "There is a significant difference between these ECS values and those derived . . ."

We are reluctant to state that the difference is significant, as significance has not been shown. Other than that, we have adjusted the text as suggested.

Page 19, line 30: Should read "Yet, high temporal variability . . ."

fixed

Page 21, line 13: Should read "still the case, for example there is a significant mismatch . . ."

We have implemented the suggestion minus the reference to significance.

Page 26, line 3: I don't think that you should use the verb "confirm" when only some of the models agree with this statement. Many of the models also disagree.

We agree with the reviewer and have adjusted the text.

Page 26, line 24: Should read "The mid-Pliocene combines estimates of carbon dioxide levels . . ."

fixed

Page 27, line 10: Should read "Hence, making inferences from modelled or reconstructed climate conditions of the mid-Pliocene with respect to . . ."

fixed

Page 27, line 12: Should read "This has been stated by . . ."

fixed

Page 30, line 21: Should read ". . . in the context of Pliocene4Future, . . ."

fixed

Page 30, line 25: Should read ". . . with potential threats to the food chain . . ."

fixed

**References**

Foley, K.M., and Dowsett, H.J.: Community sourced mid-Piacenzian sea surface temperature (SST) data, U.S. Geological Survey data release, https://doi.org/10.5066/P9YP3DTV, 2019.

McClymont, E. L., Ford, H. L., Ho, S. L., Tindall, J. C., Haywood, A. M., Alonso-Garcia, M., Bailey, I., Berke, M. A., Littler, K., Patterson, M., Petrick, B., Peterse, F., Ravelo, A. C., Risebrobakken, B., De Schepper, S., Swann, G. E. A., Thirumalai, K., Tierney, J. E., van der Weijst, C., and White, S.: Lessons from a high CO2 world: an ocean view from ~ 3 million years ago, Clim. Past Discuss., https://doi.org/10.5194/cp-2019-161, in review, 2020.

---

## Author Response (AR1)

**Contribution of the coupled atmosphere–ocean–sea ice–vegetation model COSMOS to the PlioMIP2**
**Author comment**

We once more thank both reviewers for their positive reviews and constructive comments. We have implemented all suggested changes into a revised manuscript. The two major points raised by the reviewers actually helped us to significantly improve the quality and the scientific conclusions of our manuscript, and we are very grateful for these remarks:

1. Anonymous reviewer #2 raised the point that we should expand our data-model comparison (DMC) beyond the PRISM3 data set by Dowsett et al. (2009, 2013). We have done so by considering six additional time-slice reconstructions by Foley and Dowsett (2019) and McClymont et al. (2020). We compare these to anomalies of sea surface temperature (SST) of our simulation Eoi400 with respect to pre-industrial (our simulation E280). Four of the additional reconstructions are based on reconstructions from alkenone, and two are based on magnesium-to-calcium ratio. We present our results in the following manner:

   1. we provide an additional chapter "3.7 Comparison of reconstructed and simulated SST for emerging mid-Pliocene time-slice reconstructions"

   2. we provide an expanded discussion in section "4.2 The added value of COSMOS mid-Pliocene simulations in PlioMIP2"

   3. we provide an additional Figure 24, where we present for each of the six reconstructions a model data comparison based on our modelled SST anomaly Eoi400-E280

   4. we provide an additional Table 7, where we show the root mean square deviation (RMSD) between modelled and reconstructed SST for all available SST records, for North Atlantic (from 30°N to 60°N) only, and for North Atlantic to Arctic Ocean (from 30°N onwards)

   Our main findings, presented in the manuscript, are:

   1. in many regions of the world we derive, when employing the new time-slice reconstructions that are based on alkenone, qualitatively similar results as when employing the PRISM3 time-slab data

   2. our model is able to faithfully reproduce many large scale features of reconstructed mid-Pliocene SST patterns while details of the model climate diverge in various regions from the interpretation of the geologic record

   3. we still find that the model shows weakness in reproducing high amplitude SST anomalies in the North Atlantic to Arctic Ocean, although model-data discord is here certainly reduced based on the new time-slice approach

   4. none of the choices made with regard to reconstruction (Foley and Dowsett, 2019 vs. McClymont et al., 2020), time window (10 ka vs. 30 ka by Foley and Dowsett, 2019), or calibration (PlioVar synthesis vs. Bayesian approach by McClymont et al., 2020) changes the big picture of a comparision between mid-Pliocene climate as simulated by COSMOS and as interpreted from the geologic record – an exception is the North Atlantic Ocean, where the choice of data set leads to appreciably different RMSD values

   5. on the other hand, based on the new time-slice reconstructions, the Benguela upwelling system is identified as an additional hotspot of model-data discord; there, our model is far too cold in comparision to the majority of reconstructions

6. furthermore, our model shows particular weakness in reproducing the rather cool temperatures reconstructed in the BAYMAG data set.

2. Anonymous reviewer #1 suggested to add a model-intercomparison based on already available output from other PlioMIP2 modelling groups. While our manuscript is not aimed at a model-model-intercomparison, and although an excellent publication, that explicitly focuses on model-model-intercomparison, is currently in review (Haywood et al., 2020), we have taken the suggestion into account.

    1. We provide in the additional section "4.1 Comparison of COSMOS mid-Pliocene global SAT anomaly to PlioMIP2 and PlioMIP1 models" an intercomparison of performance of the COSMOS. In this endeavour we consider contributions by COSMOS and other models to the two phases of PlioMIP.

    2. Note, that we have reworded the subsection header 4.1 in comparison to what we provided in our reply to reviewer #1. This is in order to streamline it to the new findings we have added to the new subsection of the manuscript during the last days.

    3. We are particularly grateful to the reviewer for raising the issue of a lack of studying the impact of dynamic vegetation on our model within the ensemble. This allowed us to considerably improve our discussion, that now more broadly covers the inference that COSMOS has a larger published ECS value in PlioMIP2, while it provides a lower global annual mean mid-Pliocene near surface air temperature (SAT) than it did in PlioMIP1.

    4. We discuss that this effect is at least partly related to dynamic vegetation.

    5. Furthermore, we show that the COSMOS' characteristic of increased ECS and lower simulated mid-Pliocene SAT anomaly does not appreciably influence the relationship between ECS and simulated SAT in the PlioMIP2 ensemble.

    6. To this end, based on work by Hargreaves and Annan (2016) and Haywood et al. (2020), we present regressions of SAT anomaly and model ECS of PlioMIP1 and PlioMIP2 models. We compare the statistical properties of SAT/ECS regressions for various assumptions of COSMOS characteristics in PlioMIP1 and PlioMIP2.

    7. Our conclusion is that the impact of our model's specific SAT/ECS behaviour (PlioMIP2 vs. PlioMIP1) on the SAT/ECS relationship of the PlioMIP2 ensemble is small, and in particular much smaller than the impact of other models.

    8. As our manuscript is already rather voluminous, we present the regression plots in a supplement to our manuscript and refer to these from within the manuscript. Based on the observation of a changed SAT/ECS relationship in the COSMOS in PlioMIP2 we provide an additional inference in the context of PlioMIP4Future at the end of the discussion of Section 4.2.

We are greatful to Dr. Gregor Knorr of the Alfred Wegener Institute's department for Paleoclimate Dynamics. He has provided extremely valuable input to the revised manuscript. In particular, a lot of the work, that has led to the discussion of COSMOS SAT/ECS in PlioMIP1 and PlioMIP2, has been provided by him. This discussion is substantial to the overall messages of our publication. In order to reflect his significant contribution to the manuscript we list him now as an additional author.

Beyond the two major and various minor suggestions, as well as adaptation of author list and acknowledgements section, we have made smaller adjustments to the manuscript to fix minor errors. Text changes are highlighted in a tracked-changes version of the manuscript that we will provide together with the resubmission of the manuscript. For clarity, we provide here a list of those smaller changes that do not directly stem from implementing reviewer's suggestions. Page and line numbers refer to the updated PDF with tracked changes:

- we fixed some typos at various locations, please refer to the tracked changes PDF
- we updated some numbers to match updated information in the revised manuscript by Haywood et al. (2020); please refer to the tracked changes PDF
- some sentences at various locations of the manuscript were reformulated towards better readability (please refer to the tracked changes PDF)
- on page 18, lines 24-25, we reformulated the sentence "Hence, ECS as derived here …" towards improved clarity regarding the components that are normally considered in the computation of ECS
- based on results from the ensuing model intercomparison (Haywood et al., 2020) we add on page 20, line 10ff, text, that highlights uncertainty in derivation of ESS; this finding has been noted by us after submitting the initial discussion version; yet, we believe that this fact is for PlioMIP2 and future iterations of the project important, and that hence it should be published with our results
- on page 20, line 29-30, we added a note pointing towards the discussion of model ECS that we added per suggestion of reviewer #1
- page 28, line 1-2: we corrected a grammatical error and clarified the sentence by adding "if absent"
- page 28, line 34: we corrected a grammatical error and clarified the sentence by adding "gateway-related"
- page 32, line 27: we have reformulated the sentence for better comprehension
- page 33, line 32: we adjusted the statement to better reflect the findings derived from the updated PlioMIP2 ensemble presented in the revised version of the manuscript by Haywood et al. (2020)
- page 34, line 5: we have added text to better reflect our additional findings of the impact of dynamic vegetation on the global average SAT
- page 36, line 28: we have clarified the meaning of the sentence via a minor reformulation
- page 38, line 38: we have replaced erroneous appearance of the word "data" by the word "climate"
- page 39, line 13: we have clarified the meaning of the sentence via a minor reformulation
- page 39, line 25: we have removed the text ", while one aims at isolating the impact of selected gateways that are known to have been in different states between modern and mid-Pliocene geography" as this statement would otherwise be repeated further down
- page 39, line 29: we have clarified the sentence
- we fixed the inconsistent use of the abbreviation "vs." versus the unabbreviated use "versus" by replacing "vs." at various locations by "versus"
- in reference to the updated discussion on dynamic vegetation we add in line 16 of the abstract the text ", and nonlinearities are negligible" after "and one-eightth, respectively"
- in the discussion version of the manuscript some publications were listed in the wrong order in the references list; this is fixed now and reflects in updated order of references to Haywood et al. (2009a,b), to Otto-Bliesner et al. (2017a,b), and to Zhang et al. (2013a,b); reordering of Haywood (2013) to Haywood (2013a) became necessary as in the revised manuscript we also refer to Haywood et al. (2013b) due to the extended data model comparison based on the suggestion by reviewer #2
- in reference to the remark by reviewer #1 with regard to evolution of modelled climate during model integration and, essentially, model equilibration: we refer on page 11 to Li et al. (2013) in order to state that a model integration in the order of 5,000 model years would be necessitated to bring the model into a full equilibrium with updated boundary conditions
- on page 14, line 25 we replace the erroneous use of the word "or" by the correct word "and"
- on page 27, equation 4, we fix the mistaken use of the mathematical operator "=" by replacing it with the more appropriate "approximately equal" operator
- on page 33, line 28, we improved the preciseness of our statements by changing "rather" to "also" and by adding the word "further":
  - Whether this result faithfully presents a characteristic of the real world, or whether it is rather (→also) imprinted into our simulations due to the lack of further missing dynamic components in our model (in particular ice sheets and aerosols) must remain for now an open question.
- on page 34 we fixed the errorenous citation (Otto-Bliesner et al., 2016, this study) by the correct citation (Otto-Bliesner et al., 2017b, this study)
- on page 35, line 33, we replace the word "in" by "for"
- the reference list has been amended as necessary due to text updates towards implementation of referee comments (see page 42ff in the tracked changes document)

- on page 56 we added an additional figure as per suggestion by reviewer #2
- on page 63 we added subfigures to Fig. 14 as per suggestion by reviewer #2
- on page 67 we added an additional figure as per suggestion by reviewer #2
- on page 73 we added an additional figure as per suggestion by reviewer #2
- on page 84 we added an additional table as per suggestion by reviewer #2

References:

[revised manuscript text omitted]
 [SUGG. REF. 2] near-surface air temperature by 3.37 °C [SUGG. REF. 2] and of sea surface temperature by 2.13 °C.[SUGG. REF. 2] Related to the warming from E280

to Eoi400 we find strongly decreased extent of sea ice across seasons and hemispheres. The highest sea ice decline is evident in the Northern Hemisphere during boreal summer to autumn (August, September, October – ASO; -80.3%), while Arctic winter to spring (February, March, April – FMA) sea ice drops by a comparably mildly -48.8%. Respective changes in the Southern Ocean are in comparison weaker, with a simulated loss of austral summer to autumn (FMA) sea ice by -69.1% and of austral winter to spring (ASO) sea ice by -30.3%. Noteworthy is that Arctic sea ice is confined to an average area below $2 \cdot 10^6$ km$^2$ during ASO. Furthermore, there are globally increased levels of precipitation (+0.17 mm d$^{-1}$) and evaporation (+0.18 mm d$^{-1}$).

Overall, the [SUGG. REF. 2] mid-Pliocene state is characterized by less cloud cover (-2.9%) and reduced [SUGG. REF. 2] planetary surface albedo, the albedo change being slightly biased towards [SUGG. REF. 2] the total earth surface, land and ocean (-16.6%),[SUGG. REF. 2] in comparison to [SUGG. REF. 2] albedo changes over the land surface [SUGG. REF. 2] alone (-15.6%). The simulated [SUGG. REF. 2] 
[revised manuscript text omitted]

---

## Editor Decision (ED1)

Dear Christian,

I have gone through the latest version of your manuscript, and while there seems to be no problems with the scientific aspects and the reviewers' comments and questions have been addressed, there is a long list of non-technical modifications which have to be made before proceeding to the final stage.

I have tried to list as many as possible, but first I will give some general remarks.

Despite COSMOS being an acronym for a plural word (Models), I think it may be better to refer to COSMOS as a singular noun, as is usually the case with other models. As it is a name, it would also sound better without the definite article (the) preceding it. In fact, I think both of these were done in Stepanek and Lohmann (2012). In the present manuscript, there are inconsistencies – sometimes there is indeed no definite article, and sometimes the following verb refers to a singular noun. Likewise, it would be better to not use the definite article with ECHAM5 and MPIOM, but I will leave that to you.

Throughout the manuscript, there are words written according to both British and American spelling. Sometimes, the same words appear with both spellings. I have listed British spellings, but you are free to choose American instead. Please do an automatic search for typical words, eg analyse, characterise, initialise, modelled, centre, etc.

The word order of many sentences is slightly awkward, in particular, adverbial phrases which should be moved to another position, or, in some cases, inserted between commas to make reading much easier.

Please check the list of recommended revisions below and make changes as you see fit. The page and line numbers below refer to the file *cp-2020-10-author\_response-version1.pdf*.

I am happy to see that you have included the latest proxy data and I look forward to seeing your revised manuscript.

Wing-Le Chan

page 1, line 17: comprises of a pronounced

page 1, line 19: at Northern Hemisphere high latitudes

page 2, line 15: provides us with a

page 3, line 16: Note that our (remove comma if using the word 'that')

page 3, line 17: In order to bridge the gap

page 3, line 26: high carbon dioxide levels

page 4, line 12: applied to

page 4, line 23: is not state-of-the-art anymore.

page 5, line 5: at all other latitude bands.

page 6, line 21: dependency on

page 6, line 34: once a day

page 7, line 23: difference in our

page 8, line 22: the Hudson Bay, Canadian Arctic Archipelago, as well as the Bering Strait

page 9, lines 3, 18: I don't understand the phrase 'modern, respectively PI' (as on line 15, page 11). 'modern, namely PI'? 'modern, in other words PI'?

page 9, line 24: described above

page 9, line 32: gateways, the Bering Strait

page 10, line 28: I am not sure of the phrase 'ramp simulation'. If you want to use the word 'ramp', perhaps you could use

used in the 'ramp-up' simulation 1pctCO2

with 'ramp-up' inside inverted commas, as the word 'ramp' does not seem to be commonly used with these  $CO_2$ -increasing experiments.

page 10, line 34: an equilibrium PI climate state, conforming to CMIP6 standards, as a

page 11, line 3: This setting has already been

page 11, lines 8, 14: initialisation (use consistent British spelling, as with the word 'initialise' on line 12)

- page 11, line 16: advantage of shortening spin-up
- page 11, line 19: abruptly and steadily increased concentrations
- page 11, line 23: states are not affected by drifts in the atmosphere (??)

page 12, line 22: Where did D come from? Should 'letter D' be 'letters GW'?

page 12, line 30: the Bering Strait

page 12, line 33: 'enables an identification of synergies' or 'enables us to identify synergies'

page 13, line 3: initialise, analysed (if using British spelling)

page 13, line 3: hemispheric

page 13, line 24: Characterisation of the (if using British spelling)

page 14, line 1: a comparably mild -48.8%

page 14, line 5: characterised (if using British spelling)

page 14, line 8: is, over the analysis period, slightly (reads better with commas)

page 14, line 16: focus heavily, is in quasi-equilibrium in Eoi400.

page 14, line 18: cools slightly

page 14, line 21: demonstrate the expediency of simulation Eoi400 and other COSMOS PlioMIP2 simulations for the study of climate anomalies, despite incomplete model equilibration.

page 14, line 32: over (or across) Hudson Bay and the Japanese islands (islands with small 'i' or use 'archipelago').

page 15, line 12: Reduced precipitation is also present

page 15, line 29: In our mid-Pliocene simulations, the warmer hemisphere is the northern one, where warmth is more widespread than in the Southern Hemisphere across all seasons.

page 15, line 35: centre (if using British spelling)

page 15, line 35: ice-free

page 16, line 5: characterises (if using British spelling)

page 16, line 17: mid-Pliocene in COSMOS is largely

page 16, line 20: less intense clockwise circulation, in other words, negative AMOC anomaly,

page 16, line 34: in the south, and in the north

page 17, line 8: Arctic

page 17, line 11: causing warming

page 17, line 18: 'with modern geography' or 'using modern geography'

page 17, line 31: opposes

page 18, line 10: 'Yet, for an interpretation of mid-Pliocene modelling results in terms of near-term future climate,' *or something to that effect*

page 18, line 21: A one percent annual increase in carbon dioxide levels increases global average SST

page 18, line 22: The response of SAT to increased radiative forcing outpaces that of SST, although global mean values of both reach similar values by the end of the simulation.

page 18, line 25: increase of 10°C.

page 18, line 26: 250 years only has a moderate impact on deeper parts

page 18, line 27: also warms

page 18, lines 30,31: 0.8psu, 33.7psu. psu also needs to be put in Figure 22 or in its caption.

page 18, lines 31: *Insert commas:*, for both simulations, *or rewrite sentence as* 'For both simulations, reduction of SSS is related to'

page 19, line 7: energy imbalance page 19, line 9: by the increased slope page 19, line 10: concentration

page 20, line 4: We consider these

page 20, line 17: propose a simulation Eoi560 to consider all nonlinearities

page 20, line 24: this hints at negative feedbacks

page 20, line 28: As in the case of the real world,

page 20, line 31: ice sheets in Greenland and Antarctica

page 21, line 2: both of which are

page 21, line 31: Only for large changes in carbon dioxide does a clear signal of increased AMOC emerge from

page 21, line 32: In comparison to the modern setup, for simulations with the same carbon dioxide forcing, mid-Pliocene geography causes an increase of between 0.48Sv and 1.85Sv in the maximum strength of the AMOC.

page 22, line 1: geographical setup, we find a similar trend towards increased

page 22, line 2: There is a dependency on geography, with a gain of between 0.16 Sv and 1.57 Sv for individual simulations with modern geography

page 22, line 6: no linear relationship between AMOC strength and carbon dioxide.

page 22, line 12: 'overprint' doesn't make sense. Instead, perhaps you can say something like 'although their ranges of values including internal variability overlap one another'?

page 22, line 14: insert commas: we find, for modern geography,

page 22, line 16: in which case the AMOC becomes rather shallow.

page 22, line 17: On the other hand, for both modern and mid-Pliocene geography, carbon dioxide induced strengthening of the AMOC in the upper cell is at the expense of the strength of the lower cell that imports

page 22, line 20: Note also that the mixed layer depth (no comma)

page 22, line 21: This implies that characteristics of the North Atlantic Deep Water (NADW) also change with

carbon dioxide, although the impact is not evidently represented in the structure ('albeit' sounds grammatically wrong here)

page 22, line 33: while the Atlantic

page 22, line 35: multiple ??

page 23, line 32: monotonic relationship

page 24, line 3: Our results, however, do not hint that COSMOS might support

page 24, line 9: 200,000 years

page 24, line 9: A need for progression.....has been suggested

page 24, line 22: Note that (no comma, otherwise remove the word 'that')

page 24, line 26: change 'does have' to 'has'

page 24, line 29: The whole sentence sounds awkward, especially with MyClymont et al. (2020) cited twice so closely together, and the word 'respectively' is used in the wrong way again.

While simulated SST fits similarly well with both UK37'-based and Mg/Ca-based data sets by McClymont et al. (2020), and UK37'-based data sets by Foley and Dowsett (2019), the choice of calibration and time window, respectively, has an impact on the agreement between simulation and reconstruction.

page 25, line 6: Note that (no comma, otherwise remove the word 'that')

page 25, line 6: agreement between model and new data

page 25, line 15: Note that (no comma, otherwise remove the word 'that')

page 25, line 28: The possibility of a cold bias in Mg/Ca records, which requires further attention,

page 26, line 1: Note that (no comma, otherwise remove the word 'that')

page 26, line 1: I don't understand the use of the word 'where'. These simulations were forced?

page 26, line 11: The middle section of the sentence doesn't flow well with the rest of the sentence.

Where high latitude land surface conditions are appropriate for vegetation growth, vegetation establishes in the model, provided the region is not located on modern or mid-Pliocene ice sheets.

page 26, line 12: The dashes make the sentence quite difficult to follow, especially the last two.

The obvious result is that some regions cannot produce vegetation - for a modern geography this refers to Greenland, and for mid-Pliocene geography, this refers to parts of Greenland and, in the Southern Hemisphere, to East Antarctica, due to ice sheets prescribed in the model.

page 26, line 19: with modern geography

page 26, line 21: and the dependency

page 26, line 28: monotonic

page 26, line 29: tree cover does not reach

page 27, line 2: forests in some regions

page 27, line 4: whereby the extent of tree cover returns from the northeast towards

page 27, lines 6, 11, 14: Factorisation (if using British spelling)

page 27, line 14: information necessary for

page 27, line 21: albeit a small one.

page 27, line 26: characterised (if using British spelling)

page 28, line 3: 'As these gateways' mid-Pliocene configuration' or, better still, 'As the mid-Pliocene configuration of these gateways'

page 28, line 4: model-data mismatch

page 28, line 11: on a global scale

page 28, line 21: towards the Canadian Arctic Archipelago

page 28, line 22: the North Atlantic Ocean realm is, for both summer and winter, cooler

page 28, line 24: a cooling by modern gateways extends across all longitudes

page 28, line 25: is weaker for precipitation than for temperature.

page 28, line 31: extent of the EqWP vanishes

page 29, line 2: the Bering Strait

page 29, line 3: is indeed

page 29, line 17: Hence, COSMOS allows us to examine the extent to which changes in modelling methodology

page 30, lines 2, 4: in order from highest to lowest

page 30, line 9: its ECS in PlioMIP2 is larger than the CS reported for PlioMIP1.

page 30, line 15: among which

page 30, line 24: leading towards consistency

page 30, line 27: Because of the length, structure and word order, this sentence is very difficult to follow.

The relationship between simulated mid-Pliocene SAT anomaly and ECS in COSMOS may change from PlioMIP1 to PlioMIP2 due to changes in modelling methodology, including the addition of dynamic vegetation. Below, using reasonable assumptions, we investigate how much the change in this relationship impacts on the PlioMIP2 ensemble relationship between model sensitivity to carbon dioxide and modelled temperature anomaly.

page 31, line 1: First, we exclude two and three models with high ECS from the analysis (Fig. S2c and Fig. S3c, respectively).

page 31, lines 1-5: Correct the usage of the word, 'respectively', as above.

page 31, line 2: we change COSMOS' ECS to its PlioMIP1 value

page 31, line 3: Third, we carried out a test to investigate the impact that COSMOS would have on the model ensemble if the simulated mid-Pliocene PlioMIP2 SAT were as high as in PlioMIP1

page 31, line 4: we assume COSMOS provides

page 31, line 11: We find that the linear relationship

page 31, line 12: the regression for PlioMIP1 and for the 15 member PlioMIP2 ensemble is so large

page 31, line 16: Note that (no comma)

page 31, line 21: Adopting the lower CS of PlioMIP1 for COSMOS

page 31, line 27: Only when we remove models (excluding COSMOS) which do not show a significant relationship between ECS and SAT, do we find a significant relation between model ECS and simulated SAT anomaly with increased slope in the 15-member ensemble.

But isn't this sentence obvious? You are removing the 'undesirable' models', so you will obviously be left with the 'desirable' models

page 31, line 30: causes COSMOS to exhibit opposing differences in ECS and SAT anomaly

page 31, line 34: In PlioMIP2, model characteristics other than ECS, and the potential impact of PRISM4

boundary conditions, may be more important for the simulated amplitude

I don't think 'relevant' is the right word

page 32, line 4: larger than it is in the case for PRISM3

page 32, line 6: have been shown

page 32, line 7: is rather small in COSMOS. Whether these inferences are also robust for

page 32, line 11: of particular relevance to the question of the extent to which the mid-Pliocene may serve as an analogue of future climate. *or*

of particular relevance to the question: "To what extent may the mid-Pliocene serve as an analogue of future climate?"

page 32, line 17: Based on the PlioMIP2 simulation ensemble, which is greatly extended in comparison to PlioMIP1,

'that' suggests that there are other PlioMIP2 simulations.

page 32, line 23: Thanks to this effort

page 32, line 32: most likely

page 33, line 3: and, according to our results, certainly

page 33, line 7: analyse (if using British spelling)

page 33, line 8: I don't understand this phrase 'agreement, respectively discord,'. Probably better to just say 'agreement'

page 33, line 14: Consequently, and also with respect to the updated reconstructions, discord between model and reconstructions

page 33, line 15: the discord that results from low model resolution

page 33, line 18: newly emerging

page 33, line 26: was unlikely

page 33, line 32: state that, based on proxy-data, mid-Pliocene ESS

page 34, line 1: The ratio of ESS to ECS in the mid-Pliocene for our model is 1.2, which is less than the values in the PlioMIP1 simulation - 1.5 for the model ensemble and 1.7 for COSMOS.

page 34, line 4: in COSMOS, carbon dioxide is less effective in reducing the mid-Pliocene meridional temperature gradient, and in increasing global average SAT, when raising the concentration

page 34, line 6: in comparison to the initial change in carbon dioxide, from 280 ppmv to 400 ppmv.

page 34, line 22: the phrases before and after the dash do not flow together.

quite different from conditions observed and felt today, when levels of greenhouse gas concentrations in the atmosphere are likely to be higher than those during the mid-Pliocene.

page 34, line 25: characterised (if using British spelling)

page 34, line 31: climate patterns and mechanisms which may be representative of, or at least similar to,

page 35, line 2: I don't understand the use of the word 'respectively' here. 'Imperfect' is a strange

choice of word here.

On the other hand, there are marked differences between the paleogeography of the mid-Pliocene and of modern day, or rather the near-future, in particular......the latter will be different

page 35, line 10: with a potential change of about 50 ppmv

page 35, line 17: It is relevant to test

page 35, line 18: no need to state Pliocene twice

to the overall mid-Pliocene climate state to yield a climate estimate which may be of relevance to the near or more distant future.

page 35, line 23: have already

page 35, line 30: albedo effects (no hyphen needed)

page 35, line 32: in the Northern

page 35, line 32: These results are, in general, also reproduced by COSMOS within PlioMIP2.

page 35, line 34: over the last few decades

page 35, line 35: hinting at increasing similarity between

page 36, line 2: realised (*if using British spelling*)

page 36, line 5: commas in the wrong position

the warming components, carbon dioxide, ice sheets and topography,

page 36, line 11: this leads

page 36, line 13: useful for correcting the modelled mid-Pliocene climate state for gateway effects in order to obtain a simpler interpretation

page 36, line 15: For future phases of the PlioMIP, an additional

page 36, line 21: we stress that

page 37, line 4: corrected for an interpretation in terms of the near to distant future climate.

page 37, line 7: no need for the word 'strongly'

page 37, line 8: need commas

if we consider that, for modern geography, vegetation

page 37, line 11: Similar findings in COSMOS hold for

page 37, line 17: Based on proxy data, it

page 38, line 5: model representations

page 38, line 7: not all dynamical aspects of the mid-Pliocene climate geared towards understanding the future

page 38, line 14: due to a large additional spin-up time

page 38, line 15: Instead, we implemented

page 38, line 20: is, in principal, possible

page 38, line 23: parameterised (if using British spelling)

page 38, line 24: Dolan et al. (2018) highlight the uncertainty

page 39, line 9: carbon dioxide (no hyphen)

page 39, line 14: may, in the future, also become a test bed

page 39, line 15: currently projected

page 39, line 16: food chain

- page 39, line 19: to simulate
- page 39, line 20: at a smaller scale
- page 39, line 24: the effect of further increased carbon dioxide
- page 39, line 25: characterised (if using British spelling)
- page 39, line 29: the question of how far

page 39, line 34: There is an increase in the AMOC of the equilibrium state, a characteristic observed in only some models of PlioMIP1.

page 40, line 4: two-thirds page 40, line 18: We have also page 40, line 20: *Use commas* we have considered, in our simulations, page 40, line 21: in the North Atlantic

Figs. 3, 8, 9, 10, 26, 28: two metres (if using British spelling)

Fig. 11: Surface air temperature (SAT) here refers to the surface skin temperature

Fig. 13: for all simulations considered

Fig. 14: by also showing results.....depending on the volume mixing ratio

Fig. 15: depending on the volume mixing ratio

- Fig. 17, 18: clockwise circulation from the viewpoint
- Fig. 19: The ZMLD is defined by an increase of 0.125 kgm-3 in seawater density

Table 3: which are given for reference

Table 8: depending on carbon dioxide concentration

that is arbitrarily defined here as the isoline

We specify the northward shift for WC, EC, and ES, and the eastward shift for G.

kilometres (if using British spelling)

Note that the (no comma)

In simulations with high concentrations of carbon dioxide, the tree line defined here is absent

---

## Author Response (AR2)

**CP-2020-10**
*Contribution of the coupled atmosphere–ocean–sea ice–vegetation model COSMOS to the PlioMIP2*
**Reply to the Editor after suggested non-technical modifications**

Dear Wing-Le,

indeed I am very grateful for your thorough check with regard to non-technical modifications that were still present in our revised manuscript. In my opinion, implementing your suggestions has greatly improved the quality of our manuscript. Below I provide a list of your comments and our response to each of these. An updated version of the manuscript, with changes formatted in red, is attached to the end of the author comment. Note that we updated some more inconsistencies beyond what you remarked. All these should be easily visible via red formatting in the updated manuscript version. Once more, thank you very much for the exceptional work that you have invested in helping us to improve the manuscript. This is greatly appreciated.

Christian Stepanek

1. Despite COSMOS being an acronym for a plural word (Models), I think it may be better to refer to COSMOS as a singular noun, as is usually the case with other models. As it is a name, it would also sound better without the definite article (the) preceding it. In fact, I think both of these were done in Stepanek and Lohmann (2012). In the present manuscript, there are inconsistencies – sometimes there is indeed no definite article, and sometimes the following verb refers to a singular noun. Likewise, it would be better to not use the definite article with ECHAM5 and MPIOM, but I will leave that to you.

   I have searched through the whole document and have updated the text as suggested for occurrences of COSMOS, JSBACH, MPIOM, ECHAM5, and OASIS3; in some cases I applied reformulations to avoid sentences starting with an abbreviation.

2. Throughout the manuscript, there are words written according to both British and American spelling. Sometimes, the same words appear with both spellings. I have listed British spellings, but you are free to choose American instead. Please do an automatic search for typical words, eg analyse, characterise, initialise, modelled, centre, etc.

   Indeed, differences between American and British English are not my strong suit. I have done my best to find all occurences of American English terms and replaced these with British English terms. Your detailed list was of great help in this endeavour. I have also located and fixed some further occurrences.

3. The word order of many sentences is slightly awkward, in particular, adverbial phrases which should be moved to another position, or, in some cases, inserted between commas to make reading much easier.

   Thanks for pointing this out. Indeed, one of my language problems is that I stick too close to German phrases when writing in English, and this leads often to cumbersome sentences. Where you indicated suggestions for reformulation, I have considered these.

4. Please check the list of recommended revisions below and make changes as you see fit. The page and line numbers below refer to the file cp-2020-10-author_response-version1.pdf.

   I really appreciate your work and have consequently implemente the vast majority of your suggestions "as is". Only in very few cases have I adjusted your suggestion or rejected it. In such cases, I provide an explanation for my decision.

5. page 1, line 17: comprises of a pronounced

   Actually, the first suggestion is one of the few (or maybe the only) cases where I would like to stick to the original formulation. While the formulation "comprises of" seems to be more and more accepted, the correct expression still appears to be "comprises" (i.e. without an "of"); I base this judgment on two internet sources, these are:

   https://whatis.techtarget.com/feature/Comprises-or-is-comprised-of

   https://english.stackexchange.com/questions/107869/comprise-or-comprise-of.

   Consequently, in this case we propose to keep the original wording.

6. page 1, line 19: at Northern Hemisphere high latitudes

   done

7. page 2, line 15: provides us with a

   done

8. page 3, line 16: Note that our (remove comma if using the word 'that')

   done – we have also fixed further occurrences of this issue

9. page 3, line 17: In order to bridge the gap

   done

10. page 3, line 26: high carbon dioxide levels

    done

11. page 4, line 12: applied to

    done

12. page 4, line 23: is not state–of–the–art anymore.

    done

13. page 5, line 5: at all other latitude bands.

    done

14. page 6, line 21: dependency on

    done

15. page 6, line 34: once a day

    done

16. page 7, line 23: difference in our

    done

17. page 8, line 22: the Hudson Bay, Canadian Arctic Archipelago, as well as the Bering Strait

    done

18. page 9, lines 3, 18: I don't understand the phrase 'modern, respectively PI' (as on line 15, page 11). 'modern, namely PI'? 'modern, in other words PI'?

    We changed to: "modern, namely PI"

19. page 9, line 24: described above

    done

20. page 9, line 32: gateways, the Bering Strait

    done

21. page 10, line 28: I am not sure of the phrase 'ramp simulation'. If you want to use the word 'ramp', perhaps you could use "used in the 'ramp-up' simulation 1pctCO2" with 'ramp-up' inside inverted commas, as the word 'ramp' does not seem to be commonly used with these CO2-increasing experiments.

    We have reformulated to: "forcing used in 'ramp-up' simulation"

22. page 10, line 34: an equilibrium PI climate state, conforming to CMIP6 standards, as a

    done

23. page 11, line 3: This setting has already been

    done

24. page 11, lines 8, 14: initialisation (use consistent British spelling, as with the word 'initialise' on line 12)

    done

25. page 11, line 16: advantage of shortening spin-up

    done

26. page 11, line 19: abruptly and steadily increased concentrations

    done

27. page 11, line 23: states are not affected by drifts in the atmosphere (??)

    We have reformulated to: "to ensure that the analysed model states are not affected by imperfect model equilibration, and that drifts in the atmosphere and at least in upper ocean layers are as low as practically feasible"

28. page 12, line 22: Where did D come from? Should 'letter D' be 'letters GW'?

    Indeed, I have no idea where "D" did come from, and why I did not notice that typo earlier. It has been fixed. The new text reads "difference in gateway configuration is highlighted by the letters GW, that follow the carbon dioxide concentration after an underscore."

29. page 12, line 30: the Bering Strait

    done

30. page 12, line 33: 'enables an identification of synergies' or 'enables us to identify synergies'

    We have changed this to: "it enables us to identify synergies"

31. page 13, line 3: initialise, analysed (if using British spelling)

    done

32. page 13, line 3: hemispheric

    done

33. page 13, line 24: Characterisation of the (if using British spelling)

    done

34. page 14, line 1: a comparably mild -48.8%

    done

35. page 14, line 5: characterised (if using British spelling)

    done

36. page 14, line 8: is, over the analysis period, slightly (reads better with commas)

    done

37. page 14, line 16: focus heavily, is in quasi-equilibrium in Eoi400.

    done

38. page 14, line 18: cools slightly

    done

39. page 14, line 21: demonstrate the expediency of simulation Eoi400 and other COSMOS PlioMIP2 simulations for the study of climate anomalies, despite incomplete model equilibration.

    done

40. page 14, line 32: over (or across) Hudson Bay and the Japanese islands (islands with small 'i' or use 'archipelago').

    We have reformulated to: "winter cooling across Hudson Bay and Japanese archipelago."

41. page 15, line 12: Reduced precipitation is also present

    done

42. page 15, line 29: In our mid-Pliocene simulations, the warmer hemisphere is the northern one, where warmth is more widespread than in the Southern Hemisphere across all seasons.

done

43. page 15, line 35: centre (if using British spelling)

done

44. page 15, line 35: ice-free

done

45. page 16, line 5: characterises (if using British spelling)

done

46. page 16, line 17: mid-Pliocene in COSMOS is largely

done

47. page 16, line 20: less intense clockwise circulation, in other words, negative AMOC anomaly,

done

48. page 16, line 34: in the south, and in the north

done

49. page 17, line 8: Arctic

done

50. page 17, line 11: causing warming

done

51. page 17, line 18: 'with modern geography' or 'using modern geography'

We have reformulated to: "is increased using modern geography."

52. page 17, line 31: opposes

done

53. page 18, line 10: 'Yet, for an interpretation of mid-Pliocene modelling results in terms of near–term future climate,' or something to that effect

We have reformulated to: "Yet, for an interpretation of mid-Pliocene modelling results in terms of near–term future climate,"

54. page 18, line 21: A one percent annual increase in carbon dioxide levels increases global average SST

done

55. page 18, line 22: The response of SAT to increased radiative forcing outpaces that of SST, although global mean values of both reach similar values by the end of the simulation.

   done

56. page 18, line 25: increase of 10°C.

   done

57. page 18, line 26: 250 years only has a moderate impact on deeper parts

   We have reformulated to: "short time period of 250 years has only a moderate impact on deeper parts of the global ocean"

58. page 18, line 27: also warms

   done

59. page 18, lines 30,31: 0.8psu, 33.7psu. psu also needs to be put in Figure 22 or in its caption.

   While there is a long-standing debate on whether practical salinity units are actually a physical unit, or whether they are not (e.g.: http://www.teos-10.org/pubs/Millero_History_EOS.pdf; personally I tend to the view that salinity is a unitless skalar), I have added the unit "PSU" at all suggested locations.

60. page 18, lines 31: Insert commas: ,for both simulations, or rewrite sentence as 'For both simulations, reduction of SSS is related to'

   We reformulated to: "For both simulations, reduction of SSS is related to"

61. page 19, line 7: energy imbalance

   done

62. page 19, line 9: by the increased slope

   done

63. page 19, line 10: concentration

   done

64. page 20, line 4: We consider these

   done

65. page 20, line 17: propose a simulation Eoi560 to consider all nonlinearities

   done

66. page 20, line 24: this hints at negative feedbacks

   done

67. page 20, line 28: As in the case of the real world,

   done

68. page 20, line 31: ice sheets in Greenland and Antarctica

    done

69. page 21, line 2: both of which are

    done

70. page 21, line 31: Only for large changes in carbon dioxide does a clear signal of increased AMOC emerge from

    done

71. page 21, line 32: In comparison to the modern setup, for simulations with the same carbon dioxide forcing, mid-Pliocene geography causes an increase of between 0.48Sv and 1.85Sv in the maximum strength of the AMOC.

    done

72. page 22, line 1: geographical setup, we find a similar trend towards increased

    done

73. page 22, line 2: There is a dependency on geography, with a gain of between 0.16 Sv and 1.57 Sv for individual simulations with modern geography

    done

74. page 22, line 6: no linear relationship between AMOC strength and carbon dioxide.

    done

75. page 22, line 12: 'overprint' doesn't make sense. Instead, perhaps you can say something like 'although their ranges of values including internal variability overlap one another'?

    done

76. page 22, line 14: insert commas: we find, for modern geography,

    done

77. page 22, line 16: in which case the AMOC becomes rather shallow.

    done

78. page 22, line 17: On the other hand, for both modern and mid-Pliocene geography, carbon dioxide induced strengthening of the AMOC in the upper cell is at the expense of the strength of the lower cell that imports

    done

79. page 22, line 20: Note also that the mixed layer depth (no comma)

    done

80. page 22, line 21: This implies that characteristics of the North Atlantic Deep Water (NADW) also change with carbon dioxide, although the impact is not evidently represented in the structure ('albeit' sounds grammatically wrong here)

done

81. page 22, line 33: while the Atlantic

done

82. page 22, line 35: multiple ??

Here, we wanted to indeed speak of a multipole in contrast to a dipole. Yet, as this formulation appears ambiguous, we have reformulated to: "We find a complex pattern of latitudinal distribution of precipitation over the Atlantic Ocean ..."

83. page 23, line 32: monotonic relationship

done

84. page 24, line 3: Our results, however, do not hint that COSMOS might support

done

85. page 24, line 9: 200,000 years

I was not aware that the unit "yrs" should be written out, but the text has been adjusted accordingly

86. page 24, line 9: A need for progression.......has been suggested

done

87. page 24, line 22: Note that (no comma, otherwise remove the word 'that')

done

88. page 24, line 26: change 'does have' to 'has'

done

89. page 24, line 29: The whole sentence sounds awkward, especially with MyClymont et al. (2020) cited twice so closely together, and the word 'respectively' is used in the wrong way again. While simulated SST fits similarly well with both UK37'-based and Mg/Ca-based data sets by McClymont et al. (2020), and UK37'-based data sets by Foley and Dowsett (2019), the choice of calibration and time window, respectively, has an impact on the agreement between simulation and reconstruction.

Thanks for pointing out that problem. Note, though, that your formulation changes our statement (that we have not clearly enough expressed in the first place). We have reformulated the sentence as follows, sticking with the original meaning of the statement: "Simulated SST fits similarly well with both U K 37' -based data sets presented by McClymont et al. (2020). For U K 37' -based data sets by Foley and Dowsett (2019), and as well for the Mg/Ca-based reconstructions by McClymont et al. (2020), choice of calibration and time window have an impact on agreement between simulation and reconstruction."

90. page 24, line 32: SST agree better with...than with

done

91. page 25, line 6: Note that (no comma, otherwise remove the word 'that')

done

92. page 25, line 6: agreement between model and new data

done

93. page 25, line 15: Note that (no comma, otherwise remove the word 'that')

done

94. page 25, line 28: The possibility of a cold bias in Mg/Ca records, which requires further attention,

done

95. page 26, line 1: Note that (no comma, otherwise remove the word 'that')

done

96. page 26, line 1: I don't understand the use of the word 'where'. These simulations were forced?

Indeed, there was a typo changing the meaning of our formulation. This has been fixed to "Note that these simulations were forced with a reconstruction"

97. page 26, line 11: The middle section of the sentence doesn't flow well with the rest of the sentence. Where high latitude land surface conditions are appropriate for vegetation growth, vegetation establishes in the model, provided the region is not located on modern or mid-Pliocene ice sheets.

We have reformulated the sentence to: "Vegetation establishes in the model in regions where high latitude land surface conditions are appropriate for vegetation growth. This is the case for regions outside modern and mid-Pliocene ice sheets."

98. page 26, line 12: The dashes make the sentence quite difficult to follow, especially the last two. The obvious result is that some regions cannot produce vegetation – for a modern geography this refers to Greenland, and for mid-Pliocene geography, this refers to parts of Greenland and, in the Southern Hemisphere, to East Antarctica, due to ice sheets prescribed in the model.

We have reformulated to: "The obvious result is that some regions cannot produce vegetation. For a modern geography this refers to Greenland. For mid-Pliocene geography this refers to parts of Greenland and, in the Southern Hemisphere, to East Antarctica, due to ice sheets prescribed in the model."

99. page 26, line 19: with modern geography

done

100. page 26, line 21: and the dependency

done

101. page 26, line 28: monotonic

done

102. page 26, line 29: tree cover does not reach

    done

103. page 27, line 2: forests in some regions

    done

104. page 27, line 4: whereby the extent of tree cover returns from the northeast towards

    done

105. page 27, lines 6, 11, 14: Factorisation (if using British spelling)

    done

106. page 27, line 14: information necessary for

    done

107. page 27, line 21: albeit a small one.

    done

108. page 27, line 26: characterised (if using British spelling)

    done

109. page 28, line 3: 'As these gateways' mid-Pliocene configuration' or, better still, 'As the mid-Pliocene configuration of these gateways'

    We have reformulated to: "As the mid-Pliocene configuration of these gateways"

110. page 28, line 4: model-data mismatch

    done

111. page 28, line 11: on a global scale

    done

112. page 28, line 21: towards the Canadian Arctic Archipelago

    done

113. page 28, line 22: the North Atlantic Ocean realm is, for both summer and winter, cooler

    done

114. page 28, line 24: a cooling by modern gateways extends across all longitudes

    done

115. page 28, line 25: is weaker for precipitation than for temperature.

    done

116.  page 28, line 31: extent of the EqWP vanishes

      done

117.  page 28, line 32: the Bering Strait

      done

118.  page 29, line 2: the Bering Strait

      done

119.  page 29, line 3: is indeed

      done

120.  page 29, line 17: Hence, COSMOS allows us to examine the extent to which changes in modelling methodology

      done

121.  page 30, lines 2, 4: in order from highest to lowest

      done, also in the second next line for the following bracket

122.  page 30, line 9: its ECS in PlioMIP2 is larger than the CS reported for PlioMIP1.

      done

123.  page 30, line 15: among which

      done

124.  page 30, line 24: leading towards consistency

      done

125.  page 30, line 27: Because of the length, structure and word order, this sentence is very difficult to follow. The relationship between simulated mid-Pliocene SAT anomaly and ECS in COSMOS may change from PlioMIP1 to PlioMIP2 due to changes in modelling methodology, including the addition of dynamic vegetation. Below, using reasonable assumptions, we investigate how much the change in this relationship impacts on the PlioMIP2 ensemble relationship between model sensitivity to carbon dioxide and modelled temperature anomaly.

      done

126.  page 31, line 1: First, we exclude two and three models with high ECS from the analysis (Fig. S2c and Fig. S3c, respectively).

      We have reformulated to: "First, we exclude two and three models with high ECS from the analysis (Fig. S2c and Fig. S3c)."

127.    page 31, lines 1-5: Correct the usage of the word, 'respectively', as above.

We have reformulated to: "Second, we change COSMOS' ECS to its PlioMIP1 value (Fig. S2d and Fig. S3d). Third, we carried out a test to investigate the impact that COSMOS would have on the model ensemble if the simulated mid-Pliocene PlioMIP2 SAT were as high as in PlioMIP1 (Fig. S2e and Fig. S3e). Fourth, we compute a regression where we assume COSMOS provides a larger temperature anomaly at a reduced CS as in PlioMIP1 (Fig. S2f and Fig. S3f). Fifth (Fig. S2g and Fig. S3g), we compare the impact, that all these assumptions with regard to COSMOS would have in comparison to an exclusion of models from our ensemble analysis that are known to reduce significance of the correlation between modelled SAT anomaly and model sensitivity to carbon dioxide (Haywood et al., 2020)."

128.    page 31, line 2: we change COSMOS' ECS to its PlioMIP1 value

done, see above

129.    page 31, line 3: Third, we carried out a test to investigate the impact that COSMOS would have on the model ensemble if the simulated mid-Pliocene PlioMIP2 SAT were as high as in PlioMIP1

done, see above

130.    page 31, line 4: we assume COSMOS provides

done, see above

131.    page 31, line 11: We find that the linear relationship

done

132.    page 31, line 12: the regression for PlioMIP1 and for the 15 member PlioMIP2 ensemble is so large

done

133.    page 31, line 16: Note that (no comma)

done

134.    page 31, line 21: Adopting the lower CS of PlioMIP1 for COSMOS

done

135.    page 31, line 27: Only when we remove models (excluding COSMOS) which do not show a significant relationship between ECS and SAT, do we find a significant relation between model ECS and simulated SAT anomaly with increased slope in the 15-member ensemble. But isn't this sentence obvious? You are removing the 'undesirable' models', so you will obviously be left with the 'desirable' models

You are right, but we kept COSMOS in the analysis. Figure S2g should provide a conclusion of previous assumptions. We hope that applied reformulation of the sentence makes this clear: "To conclude previous assumptions: only when we remove models (excluding COSMOS) which do not show a significant relationship between ECS and SAT, do we find a relation between model ECS and simulated SAT anomaly with increased slope in the 15-member ensemble that is significant (Fig. S2g), and an increased slope with vanishing p-value in the 16 member ensemble (Fig. S3g)."

136.    page 31, line 30: causes COSMOS to exhibit opposing differences in ECS and SAT anomaly

   done

137.    page 31, line 34: In PlioMIP2, model characteristics other than ECS, and the potential impact of PRISM4 boundary conditions, may be more important for the simulated amplitude I don't think 'relevant' is the right word

   you are right; done

138.    page 32, line 4: larger than it is in the case for PRISM3

   done

139.    page 32, line 6: have been shown

   done

140.    page 32, line 7: is rather small in COSMOS. Whether these inferences are also robust for

   done

141.    page 32, line 11: of particular relevance to the question of the extent to which the mid-Pliocene may serve as an analogue of future climate. Or of particular relevance to the question: "To what extent may the mid-Pliocene serve as an analogue of future climate?"

   I particular like the second suggestion - I only adapted it slightly: I think it should read "as an analogue *to* future climate", am I right? The reformulated text is consequently: "We suggest that the inference of potentially reduced ties between simulated mid-Pliocene temperature and model ECS in some PlioMIP2 models is of particular relevance to the question: 'To what extent may the mid-Pliocene serve as an analogue to future climate?'"

142.    page 32, line 17: Based on the PlioMIP2 simulation ensemble, which is greatly extended in comparison to PlioMIP1, 'that' suggests that there are other PlioMIP2 simulations.

   done

143.    page 32, line 23: Thanks to this effort

   done

144.    page 32, line 32: most likely

   done

145.    page 33, line 3: and, according to our results, certainly

   done

146.    page 33, line 7: analyse (if using British spelling)

   done

147.    page 33, line 8: I don't understand this phrase 'agreement, respectively discord,'. Probably better to just say 'agreement'

We have reformulated to: "on agreement between model and inference from the geologic record."

148.    page 33, line 14: Consequently, and also with respect to the updated reconstructions, discord between model and reconstructions

done

149.    page 33, line 15: the discord that results from low model resolution

done

150.    page 33, line 18: newly emerging

done

151.    page 33, line 26: was unlikely

done

152.    page 33, line 32: state that, based on proxy-data, mid-Pliocene ESS

done

153.    page 34, line 1: The ratio of ESS to ECS in the mid-Pliocene for our model is 1.2, which is less than the values in the PlioMIP1 simulation - 1.5 for the model ensemble and 1.7 for COSMOS.

done

154.    page 34, line 4: in COSMOS, carbon dioxide is less effective in reducing the mid-Pliocene meridional temperature gradient, and in increasing global average SAT, when raising the concentration

done

155.    page 34, line 6: in comparison to the initial change in carbon dioxide, from 280 ppmv to 400 ppmv.

done

156.    page 34, line 22: the phrases before and after the dash do not flow together. quite different from conditions observed and felt today, when levels of greenhouse gas concentrations in the atmosphere are likely to be higher than those during the mid-Pliocene.

We have reformulated to: "may look quite different from conditions observed and felt today, when levels of greenhouse gas concentrations in the atmosphere are likely to be already higher than those during the mid-Pliocene."

157.    page 34, line 25: characterised (if using British spelling)

done

158. page 34, line 31: climate patterns and mechanisms which may be representative of, or at least similar to,

    done

159. page 35, line 2: I don't understand the use of the word 'respectively' here. 'Imperfect' is a strange choice of word here. On the other hand, there are marked differences between the paleogeography of the mid-Pliocene and of modern day, or rather the near-future, in particular........the latter will be different

We have taken your suggestion into account and applied further reformulation for clarity: "On the other hand, there are marked differences between the paleogeography of the mid-Pliocene and of modern day, or rather the near-future. In particular, the latter will be different with regard to details in coastlines, state of ocean gateways, and height and extent of ice sheets (e.g. Dowsett et al., 2016; Haywood et al., 2016). Difference in ice sheets will last at least until the modern cryosphere has reached a new equilibrium under the influence of elevated carbon dioxide."

160. page 35, line 10: with a potential change of about 50 ppmv

    done

161. page 35, line 17: It is relevant to test

    done

162. page 35, line 18: no need to state Pliocene twice to the overall mid-Pliocene climate state to yield a climate estimate which may be of relevance to the near or more distant future.

We noticed that the sentence was actually quite long. We reformulated and split for clarity: "It is relevant to test individual contributions of various drivers, that act at different time scales, to the overall mid-Pliocene climate state. Such analysis may contribute to an estimate, which aspects of mid-Pliocene warmth may become of relevance in the near or more distant future."

163. page 35, line 23: have already

    done

164. page 35, line 30: albedo effects (no hyphen needed)

    done

165. page 35, line 32: in the Northern

    done

166. page 35, line 32: These results are, in general, also reproduced by COSMOS within PlioMIP2.

    done

167. page 35, line 34: over the last few decades

    done

168.    page 35, line 35: hinting at increasing similarity between

    done

169.    page 36, line 2: realised (if using British spelling)

    done

170.    page 36, line 5: commas in the wrong position the warming components, carbon dioxide, ice sheets and topography,

    done

171.    page 36, line 11: this leads

    done

172.    page 36, line 13: useful for correcting the modelled mid-Pliocene climate state for gateway effects in order to obtain a simpler interpretation

    done

173.    page 36, line 15: For future phases of the PlioMIP, an additional

    done

174.    page 36, line 21: we stress that

    done

175.    page 37, line 4: corrected for an interpretation in terms of the near to distant future climate.

    done

176.    page 37, line 7: no need for the word 'strongly'

    We have reformulated to: "We also show that simulated vegetation patterns change more in the mid-Pliocene"

177.    page 37, line 8: need commas if we consider that, for modern geography, vegetation

    We have reformulated to: "This is also confirmed if we consider that, for modern geography, vegetation cannot grow on Greenland and Antarctica, as a result of ice sheet cover."

178.    page 37, line 11: Similar findings in COSMOS hold for

    done

179.    page 37, line 17: Based on proxy data, it

    done

180.    page 38, line 5: model representations

    done

181. page 38, line 7: not all dynamical aspects of the mid-Pliocene climate geared towards understanding the future

We have reformulated to: "Consequently, not all dynamical aspects of the mid-Pliocene climate, geared towards understanding the future, are covered by COSMOS in particular, and by PlioMIP2 in general."

182. page 38, line 14: due to a large additional spin-up time

done

183. page 38, line 15: Instead, we implemented

done

184. page 38, line 20: is, in principal, possible

We have reformulated to: "Dynamic treatment of the land cryosphere is, in principal, possible in COSMOS (Barbi et al., 2014) via a coupled ice sheet model but would prevent the implementation of the PRISM4 reconstruction of ice sheet extent and height (Dowsett et al., 2016) as per modelling protocol (Haywood et al., 2016)."

185. page 38, line 23: parameterised (if using British spelling)

done

186. page 38, line 24: Dolan et al. (2018) highlight the uncertainty

done

187. page 39, line 9: carbon dioxide (no hyphen)

done

188. page 39, line 14: may, in the future, also become a test bed

done

189. page 39, line 15: currently projected

done

190. page 39, line 16: food chain

done

191. page 39, line 19: to simulate

done

192. page 39, line 20: at a smaller scale

done

193. page 39, line 24: the effect of further increased carbon dioxide

done

194.    page 39, line 25: characterised (if using British spelling)

    done

195.    page 39, line 29: the question of how far

    done

196.    page 39, line 34: There is an increase in the AMOC of the equilibrium state, a characteristic observed in only some models of PlioMIP1.

    done

197.    page 40, line 4: two-thirds

    done

198.    page 40, line 18: We have also

    done

199.    page 40, line 20: Use commas we have considered, in our simulations,

    done

200.    page 40, line 21: in the North Atlantic

    done

201.    Figs. 3, 8, 9, 10, 26, 28: two metres (if using British spelling)

    done

202.    Fig. 11: Surface air temperature (SAT) here refers to the surface skin temperature

    done

203.    Fig. 13: for all simulations considered

    done

204.    Fig. 14: by also showing results............depending on the volume mixing ratio

Note, that the first part of the text was only an editorial note and is remvoed. We have reformulated to: "Sea ice coverage (SIC) in the Northern Hemisphere for boreal spring (MAM, left) and boreal autumn (SON, right) depending on the volume mixing ratio of carbon dioxide for modern geography. Shown are: a), b) E280; c), d) E400; e), f) E560; g), h) E600. Grey shading illustrates the land sea mask. Red contours illustrate SIC-isolines of 15% (dotted), 75% (dashed), and 95% (solid)."

205.    Fig. 15: depending on the volume mixing ratio

We have reformulated to: "Sea ice coverage (SIC) in the Northern Hemisphere for boreal spring (MAM, left) and boreal autumn (SON, right) depending on the volume mixing ratio of carbon dioxide for mid-Pliocene geography. Shown are: a), b) Eoi280; c), d) Eoi350; e), f) Eoi450; g), h) Eoi560. Grey shading illustrates the land sea mask. Red contours illustrate SIC-isolines of 15% (dotted), 75% (dashed), and 95% (solid)."

206. Fig. 17, 18: clockwise circulation from the viewpoint

done

207. Fig. 19: The ZMLD is defined by an increase of 0.125 kgm-3 in seawater density

done

208. Table 3: which are given for reference

done

[revised manuscript text omitted]